# Reasoning-Aligned Perception Decoupling for Scalable Multi-modal Reasoning

**Yunhao Gou**[1,2*]   **Kai Chen**[2*]   **Zhili Liu**[2,3*]   **Lanqing Hong**[3]   **Xin Jin**[4],
**Zhenguo Li**[3]   **James T. Kwok**[2]   **Yu Zhang**[1†]

[1]Southern University of Science and Technology
[2]The Hong Kong University of Science and Technology
[3] Huawei Noah's Ark Lab    [4] Huawei Cloud

Project Page: `https://github.com/gyhdog99/RAPID/`

## Abstract

Recent breakthroughs in reasoning language models have significantly advanced text-based reasoning. On the other hand, Multi-modal Large Language Models (MLLMs) still lag behind, hindered by their outdated internal LLMs. Upgrading these LLMs is often prohibitively expensive, as it requires costly vision-language alignment retraining. To address this issue, we introduce **Perception-Reasoning Decoupling**, which modularizes the MLLM's reasoning component and makes it easily replaceable. This approach redefines the MLLM's role to convert multi-modal inputs into detailed textual outputs that can be processed by any powerful, external, text-only LLM reasoners. To align the MLLM's perceptual output with the final reasoning task, we propose a novel reinforcement learning algorithm called **Visual Perception Optimization** (VPO). VPO rewards the MLLM based on the correctness of answers generated by the external reasoner to produce faithful and query-relevant captions. Together, this decoupling pipeline and VPO form our **R**easoning-**A**ligned **P**ercept**I**on **D**ecoupling (RAPID) approach. Empirical results show that RAPID achieves significant performance gains on multi-modal reasoning benchmarks. Crucially, RAPID enables a **novel inference-time scaling paradigm**: Once trained with VPO, the MLLM can be paired with any state-of-the-art LLM reasoner for consistent performance improvement without retraining.

## 1 Introduction

Recent reasoning language models, such as OpenAI-o1 (Jaech et al., 2024) and Qwen3 (Yang et al., 2025a), have driven significant gains in complex math and science tasks. By emulating a deliberate, step-by-step reasoning process akin to human reflection, these models avoid superficial shortcuts. As a result, they substantially outperform previous models like GPT-4o (Hurst et al., 2024), with improvements exceeding 30% on math benchmarks like AIME24 (AIME, 2024) and around 10% on science benchmarks like GPQA (Rein et al., 2024).

Translating breakthroughs from the uni-modal text to the multi-modal domain remains a significant challenge. Existing multi-modal large language models (MLLMs), like Qwen2.5-VL (Bai et al., 2025), Gemma3 (Team et al., 2025a), and InternVL3 (Zhu et al., 2025), still struggle with reasoning and math-intensive tasks because their underlying LLMs are outdated or lack slow-thinking capabilities. While approaches like VL-Rethinker (Wang et al., 2025a) and MM-EUREKA (Meng et al., 2025) try to improve performance with reinforcement learning, their success is fundamentally restricted by the reasoning capability of the base LLM. The ideal solution, namely, switching the LLM with the most state-of-the-art one, is often prohibitive, as it requires repeating the entire, costly vision-language alignment process. This raises the critical question: *Can we replace the LLM within an MLLM to unlock advanced reasoning[1] efficiently, without undertaking redundant vision-language retraining?*

To address that, we propose the **Perception-Reasoning Decoupling** pipeline, where we re-focus the MLLM's primary role on *perception*. It first translates the multi-modal inputs into a comprehensive

---

[*]Equal contribution
[†]Correspondence to yu.zhang.ust@gmail.com

[1]In this paper, we focus on multi-modal math and science reasoning tasks.

textual representation, which is then processed by a separate, powerful, external LLM for *reasoning*. This decoupling allows flexible alteration of the LLM reasoner, offering a path to circumvent the costly retraining cycle. Our key distinction from similar two-stage pipelines (Tiong et al., 2022; Guo et al., 2022; Hu et al., 2022; Gou et al., 2024; Lu et al., 2025) lies in the textual representation, which includes both *a query-relevant caption* and *a tentative solution* to ensure all essential visual information is captured for subsequent reasoning. However, the critical challenge in this new pipeline is that *the generated textual outputs are not optimized for correct reasoning*.

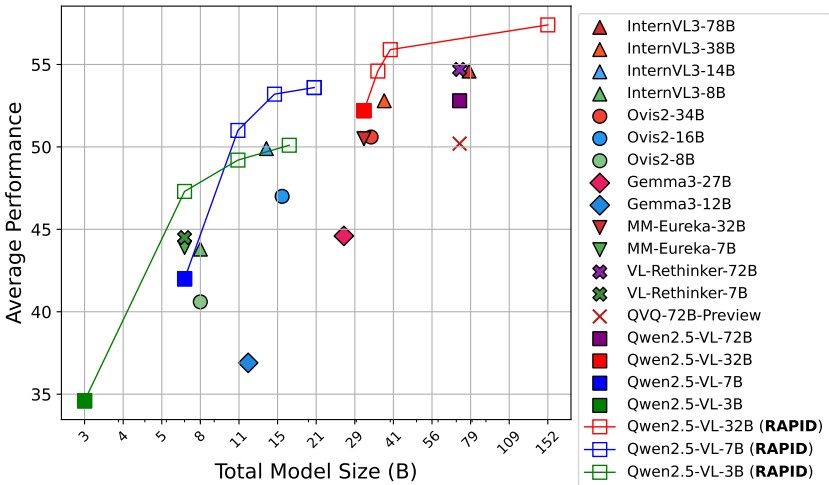

Figure 1: **Comparisons on multi-modal reasoning benchmarks** on average performance and total model size between RAPID-enhanced Qwen2.5-VL series of models and the other existing MLLMs. Check the detailed numerical results in Appendix B and experimental settings in Sec. 4.1.

To overcome that, we introduce **Visual Perception Optimization** (VPO), a novel policy gradient algorithm operating through a reinforcement learning feedback loop where, given a user query, the MLLM first generates a group of query-relevant captions, which are then considered as contexts for the external LLM reasoner to generate final answers. With a correctness-based reward function, the perceptual MLLM is aligned with the reasoning objective, and guided to generate faithful and query-relevant captions optimized for the correctness of downstream reasoning. The combination of the *Perception-Reasoning Decoupling* pipeline with the *VPO algorithm* forms our overall approach, named **R**easoning-**A**ligned **P**ercept**I**on **D**ecoupling (RAPID)

Empirically, RAPID achieves notable performance gains on challenging benchmarks such as Math-Verse (Zhang et al., 2024), MathVision (Wang et al., 2024b) and LogicVista (Xiao et al., 2024). Moreover, as perception and reasoning are decoupled, the MLLMs trained with VPO generate textual outputs that can be directly fed to any LLM for reasoning. This eliminates the necessity for retraining, and enables RAPID to be a practical solution for the rapid evolution of MLLMs and reasoning LLMs. Figure 1 compares various MLLMs against the RAPID-enhanced Qwen2.5-VL series. For each RAPID-enhanced group (*e.g.*, Qwen2.5-VL-3B), we optimize the MLLM with VPO using minimal data (39K). The resulting performance curves are generated simply by pairing the optimized MLLM with increasingly powerful external LLMs (see Appendix B for the choice of configurations), demonstrating a novel inference-time scaling paradigm.

Our contributions can be summarized as follows:

- We introduce the **Perception-Reasoning Decoupling** pipeline, which redefines MLLMs' focus to multi-modal perception, allowing the reasoning component to be flexibly replaced by any advanced external LLM without burdensome retraining.

- We propose **Visual Perception Optimization** (VPO), a novel policy gradient algorithm that aligns the MLLM's perceptual outputs by using the correctness of the external LLM's final answers with the perceptual outputs as contexts for reward signals.

- Combining both, RAPID achieves significant performance gains and introduces an efficient, novel "plug-and-play" inference-time scaling approach. By eliminating the costly retraining required by traditional methods, an one-time optimized MLLM can be paired with any stronger LLM for continual performance improvements, as demonstrated in Figure 1.

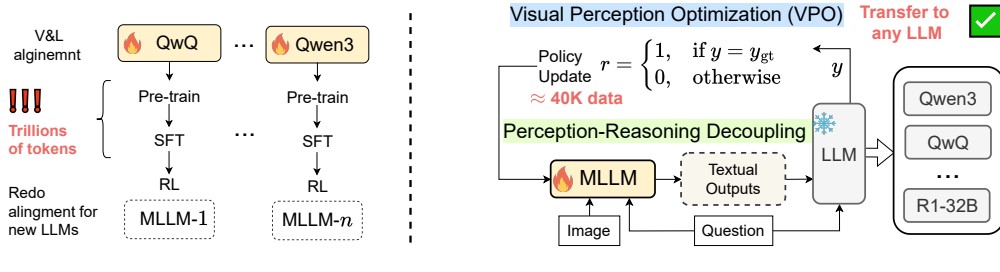

(a) **Existing MLLM alignment methods**.          (b) **RAPID (ours)**.

Figure 2: **Comparisons between RAPID and existing alignment methods for reasoning MLLMs.** For novel LLMs, existing methods (a) repeatedly conduct the compute-intensive alignment procedure, while (b) RAPID decouples the *visual perception* from *text-only reasoning* (Sec. 3.1) by learning to extract reasoning-aligned visual contexts with the proposed VPO algorithm (Sec. 3.2). Note that the caption penalty, as in Eq. 3, is omitted here for simplicity. The flame and snowflake icons indicate the models are trainable and frozen, respectively, during the process.

## 2 RELATED WORK

**Improving the Reasoning Ability of MLLMs.**    Start with an existing MLLM (*e.g.*, Qwen2.5-VL (Bai et al., 2025)), a widely adopted approach is to perform reinforcement learning or knowledge distillation. For example, VL-Rethinker (Wang et al., 2025a), MM-EUREKA (Meng et al., 2025) and NoisyRollout (Liu et al., 2025a) apply GRPO (Shao et al., 2024) (or its variant (Liu et al., 2025c)) to MLLMs to learn deliberate reasoning patterns. Distillation-based methods, such as R1-OneVision (Yang et al., 2025b), Vision-R1 (Huang et al., 2025) and ReVisual-R1 (Chen et al., 2025b) perform supervised fine-tuning (SFT) on reasoning data. However, both methods are restricted by the base LLMs (*e.g.*, Qwen2.5 (Yang et al., 2024)), which lags behind state-of-the-art reasoning models (*e.g.*, Qwen3-8B (Yang et al., 2025a)). While adopting a stronger LLM is an intuitive solution, re-aligning vision and language through full retraining on trillions of tokens is prohibitively costly.

**Caption-then-Reason Pipelines.**    To leverage LLM reasoning without intensive retraining, prior work explores similar "caption-then-reason" pipelines that decouple perception from reasoning. These approaches use Vision-Language Models (VLMs) (Radford et al., 2021; Li et al., 2022; Yu et al., 2022) or MLLMs for the perception task, while a separate LLM handles reasoning. Efforts in this area primarily focus on improving caption generation, for instance by selecting query-relevant image patches for captioning (Tiong et al., 2022; Guo et al., 2022), prompting MLLMs for query-aware captions (Gou et al., 2024), or enhancing captioning datasets (Hu et al., 2022; Lu et al., 2025). RAPID differs from these works in two key aspects. First, it includes a tentative solution in its generated output to better capture critical visual information. Second, while existing methods do not optimize captions for the final outcome, RAPID rewards the captioning process based on the correctness of the final answer produced by the reasoning LLM.

## 3 METHODOLOGY

This section describes the two main components of RAPID: *perception-reasoning decoupling* (Sec. 3.1) and *visual perception optimization* (Sec. 3.2). Figure 2 gives an overview of the approach.

### 3.1 PERCEPTION-REASONING DECOUPLING

Given an image $I$ and a relevant query $q$, our perception-reasoning decoupling pipeline involves two consecutive stages: (i) **Perception**, where an MLLM (*e.g.*, Qwen2.5-VL (Bai et al., 2025)) acts as a perception module to generate a group of textual outputs $O_p$ (detailed below) with respect to the image $I$ and a perception prompt. (ii) **Reasoning**, where a powerful text-only LLM reaonser (*e.g.*, R1-Distilled-7B (Guo et al., 2025) or Qwen3-8B (Yang et al., 2025a)) receives the original query $q$ and a consolidated set of perceptual outputs, $O_p$, which are structured by a reasoning prompt $P_r$ (shown in Fig. 12): $y = \text{LLM}(P_r(q, O_p))$.

A key advantage of this decoupling pipeline is that the textural outputs form a universal interface between perception (MLLMs) and reasoning (LLMs). This allows the reasoning LLMs to be upgraded

independently, boosting performance without the necessity to retrain the MLLMs or alignment. A detailed empirical analysis is provided in Sec. 4.3.

**Strategies for Visual Perception** $O_p$**.**   We explore strategies to generate precise perceptual outputs for reasoning:

- `none`: An empty set of perceptual outputs $O_p$, which serves as the control reference group.

- `cap` (Lu et al., 2025): A standard image caption $o_{\text{cap}} = \text{MLLM}(I, P_{\text{cap}})$ with the template $P_{\text{cap}}$ in Fig. 13.

- `qcap` (Gou et al., 2024):  A query-relevant caption $o_{\text{qcap}} = \text{MLLM}(I, P_{\text{qcap}}(q))$ with $P_{\text{qcap}}$ in Fig. 14.

- `sol`: A tentative solution $o_{\text{sol}} = \text{MLLM}(I, P_{\text{sol}}(q))$ with the template $P_{\text{sol}}$ in Fig. 15, which is "tentative" as it acts as contexts for LLMs instead of final answers.

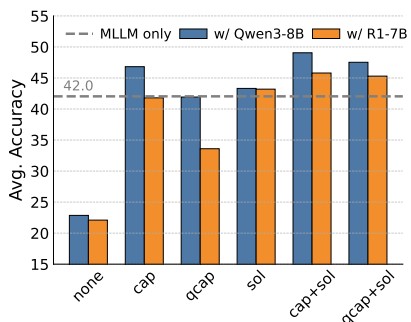

Figure 3: **Comparison of the strategies for visual perception** $O_p$**.**

To evaluate how different compositions of the perceptual output set $O_p$ affect performance, we perform an experiment on seven multi-modal reasoning benchmarks (details in Sec. 4.1). In particular, we set $O_p$ to: (i) none; (ii) $\{o_{\text{cap}}\}$; (iii) $\{o_{\text{qcap}}\}$; (iv) $\{o_{\text{sol}}\}$; (v) $\{o_{\text{cap}}, o_{\text{sol}}\}$; and (vi) $O_p = \{o_{\text{qcap}}, o_{\text{sol}}\}$. The reasoning prompt $P_r$ (shown in Fig. 12) is designed to provide a unified structure for all the above cases. We use Qwen2.5-VL-7B for perception and adopt Qwen3-8B and R1-Distilled-7B (refered to as R1-7B) for reasoning. Figure 3 shows the average accuracies obtained. As can be seen,

- **Standard captions outperform query-relevant counterparts.** This can be attributed to the MLLM's extensive training on standard image captioning tasks (Bai et al., 2025), whereas query-relevant captioning remains less optimized. However, with proper optimization (Sec. 3.2), query-relevant captioning can offer an advantage by extracting contextually relevant visual details.

- **Combining tentative responses and standard captions achieves best results**, delivering significant improvement (+7% w/ Qwen3-8B) over the original MLLM. This success is due to the complementary roles played by the caption and tentative response in reasoning.  The caption provides the LLM with essential contexts for problem-solving, while the tentative response serves as a reference.

While Figure 3 shows that `cap+sol` performs best, we will adopt `qcap+sol` as the default in the sequel.  The reason is empirical.  Our findings in Section 4.2 reveal that `qcap+sol` outperforms `cap+sol` once VPO (to be introduced in Section 3.2) is applied.  This indicates that the query-relevant approach, while less optimized initially, possesses greater potential.

## 3.2   VISUAL PERCEPTION OPTIMIZATION (VPO)

Although the combination of caption and tentative solution (*i.e.*, both `cap+sol` and `qcap+sol`) demonstrates superior results in Figure 3, they are not optimized for the correctness of the final reasoning outcome. In other words, the MLLM generates its perception outputs without any feedback on whether these outputs actually guide the reasoning LLM to the correct answer. To address this limitation, we introduce **Visual Perception Optimization** (VPO). As illustrated in Figure 4, VPO establishes a reinforcement learning feedback loop that fine-tunes the MLLM for better captioning, explicitly rewarding it based on the correctness of the final answer produced by the reasoning LLM.

**Objective Design.**   Without loss of generality, we describe VPO using the query-relevant caption (`qcap`) setting. VPO is inspired by Group Relative Policy Optimization (GRPO) (Shao et al., 2024), a policy optimization algorithm originally developed for text-only LLMs. In our setting, the policy $\pi_\theta$ to optimize is the MLLM performing visual captioning. For a given input pair $(I, q)$ from the training set $p_{\mathcal{D}}$, the old policy generates $G$ caption rollouts $\{o \sim \pi_{\theta_{\text{old}}}(\cdot|I, P_{\text{qcap}}(q))\}$. Let $R_i$ be the reward for the $i$th rollout. The normalized advantage is $\hat{A}_i = \frac{R_i - \bar{R}}{\sigma(R)}$, where $\sigma(R)$ is the standard deviation of rewards within the group $R = \{R_i\}$ and $\bar{R} = \frac{1}{G}\sum_{i=1}^{G} R_i$ is the baseline reward. Thus,

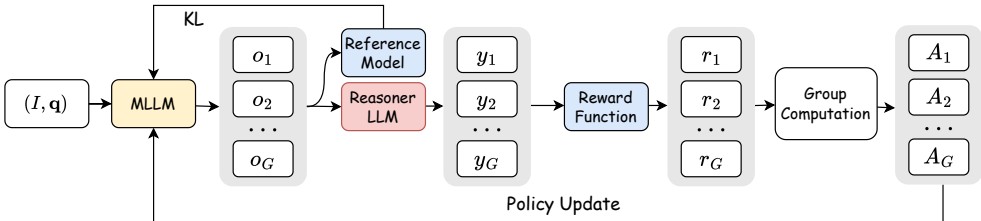

Figure 4: **Visual Perception Optimization (VPO)** reinforces *captions* that induce correct *reasoning* results via reinforcement learning with verifiable rewards. Here we omit caption penalty for simplicity.

the objective of VPO, following the formulation of GRPO, can be represented as:

$$L(\theta) = \mathbb{E}_{(I,q)\sim p_{\mathcal{D}}, o\sim \pi_{\theta_{\text{old}}}(\cdot|I, P_{\text{qcap}}(q))}$$
$$\left[ \frac{1}{G} \sum_{i=1}^{G} \min \left( \frac{\pi_\theta(o_i \mid I, P_{\text{qcap}}(q))}{\pi_{\theta_{\text{old}}}(o_i \mid I, P_{\text{qcap}}(q))} \hat{A}_i, \text{clip}\left( \frac{\pi_\theta(o_i \mid I, P_{\text{qcap}}(q))}{\pi_{\theta_{\text{old}}}(o_i \mid I, P_{\text{qcap}}(q))}, 1 - \epsilon_{\text{l}}, 1 + \epsilon_{\text{h}} \right) \hat{A}_i \right) \right],$$

(1)

which incorporates a surrogate loss clipped to $[1 - \epsilon_{\text{l}}, 1 + \epsilon_{\text{h}}](\epsilon_{\text{l}} > 0, \epsilon_{\text{h}} > 0))$ and a KL-penalty $D_{\text{KL}}[\pi_\theta | \pi_{\theta_{\text{ref}}}]$ weighted by $\beta$ (not shown) to stabilize optimization.

**Reward Design.** GRPO is often used in math reasoning problems (Shao et al., 2024; Guo et al., 2025), in which the reward is determined by a simple rule because the model's output is the final solution itself. However, in our setting, the MLLM generates an intermediate caption, from which the reasoning solution cannot be directly extracted. To address this, for each caption rollout $o_i$, we prompt the reasoning LLM to generate a final answer, and the reward is determined by whether this answer matches the ground-truth. This is formalized as follows:

$$\hat{R}_i = r(y_{\text{gt}}, y_i) = \mathbb{1}(y_{\text{gt}} = \text{parse}(y_i)), \quad \text{where } y_i = \text{LLM}(P'_{\text{reason}}(q, o_i)),$$

(2)

where $y_i$ is the answer produced by the LLM from caption $o_i$, $\mathbb{1}(\cdot)$ is the indicator function, and $P'_{\text{reason}}(q, o_i)$ (shown in Figure 16) is the reasoning prompt different from that used for inference (template in Figure 12) as it omits the tentative solution. The reward function $r(\cdot, \cdot)$ compares the predicted answer with the ground-truth $y_{\text{gt}}$.

During training, we observe reward hacking (details in Sec. 4.2), where the MLLM directly solves the problem instead of performing captioning. Consequently, the model's captioning ability does not improve. To address that, we impose a penalty on reward $\hat{R}_i$ if (i) $o_i$ leads to a correct answer; and (ii) $o_i$ does not contain a caption (determined by the policy MLLM $\pi_\theta$ via few-shot prompting):

$$R_i = \hat{R}_i - \lambda \mathbb{1}\big(\hat{R}_i = 1 \wedge \neg\text{hasCap}(o_i)\big),$$

(3)

where $\lambda$ is a penalizing factor, and the function[2] hasCap$(\cdot)$ checks if $o_i$ contains a caption (template in Figure 17).

In summary, VPO offers two primary advantages:

- **Generation of Reasoning-Aligned Captions:** VPO uses the final reasoning outcome as a reward signal to optimize MLLMs, ensuring the captions are not merely descriptive but also functionally aligned for further reasoning. The performance improvement will be shown in Sec. 4.4.

- **LLM-Agnostic and Generalizable Improvement:** VPO is an LLM-agnostic optimization, *i.e.*, the optimized MLLM communicates with the LLM reasoner via natural language, and thus, the performance gain is generalizable across any LLMs. This enables a **one-time alignment**, which can be paired with any LLM in a plug-and-play way without repeating VPO, as shown in Sec. 4.3.

In addition to the caption, we further improve the quality of the tentative solution generated by the MLLM. As the tentative solution can be easily verified by a rule-based reward function, we apply GRPO on the MLLM. Details can be found in Appendix D. In the experiments, we optimize the MLLM with GRPO and VPO in a sequential manner, with VPO followed by GRPO.

---

[2]Other variants of the check function are considered in Appendix E.3.

Table 1: **Comparison on multi-modal reasoning benchmarks** with respect to average accuracies. The best results are **bold**, while the second best are underlined. *: short for GPT-OSS-120B-A5B. ‡: undergone GRPO training.

| Model | MathVista | MathVision | MathVerse | MMMU | WeMath | DynaMath | LogicVista | AVG |
|---|---|---|---|---|---|---|---|---|
| **Proprietary Models** | | | | | | | | |
| Claude-3.7-Sonnet | 66.8 | 41.9 | 46.7 | **75.0** | 49.3 | 39.7 | 58.2 | 53.9 |
| Gemini-2.0-Flash | 70.4 | 43.6 | 47.7 | 72.6 | 47.4 | **42.1** | 52.3 | 53.7 |
| GPT-4o-20241120 | 60.0 | 31.2 | 40.6 | 70.7 | 45.8 | 34.5 | 52.8 | 47.9 |
| **Open-Source Models** | | | | | | | | |
| MM-Eureka-7B | 73.0 | 27.9 | 46.1 | 54.9 | 34.7 | 22.6 | 48.3 | 43.9 |
| InternVL3-8B | 73.6 | 29.3 | 39.8 | 62.7 | 37.1 | 25.5 | 44.1 | 44.6 |
| VL-Rethinker-7B | 74.9 | 30.0 | 47.5 | 56.9 | 37.3 | 21.4 | 43.6 | 44.5 |
| ReVisual-R1-7B | 70.8 | 43.0 | 52.7 | 55.7 | 40.7 | 30.5 | 51.2 | 49.2 |
| Ovis2-8B | 71.8 | 25.9 | 42.3 | 57.4 | 27.2 | 20.4 | 39.4 | 40.6 |
| InternVL3-14B | 75.1 | 37.2 | 44.4 | 67.1 | 43.0 | 31.3 | 51.2 | 49.9 |
| Ovis2-16B | 73.7 | 30.1 | 45.8 | 60.7 | 45.0 | 26.3 | 47.4 | 47.0 |
| Gemma-3-12B | 56.1 | 30.3 | 21.1 | 55.2 | 33.6 | 20.8 | 41.2 | 36.9 |
| MM-Eureka-32B | 74.7 | 36.6 | 51.5 | 62.0 | 37.1 | 33.5 | 58.2 | 50.5 |
| InternVL3-38B | 75.1 | 34.2 | 48.2 | 70.1 | 48.6 | 35.3 | 58.4 | 52.8 |
| Ovis2-34B | 76.1 | 31.9 | 50.1 | 66.7 | 51.9 | 27.5 | 49.9 | 50.6 |
| Gemma-3-27B | 59.8 | 39.8 | 34.0 | 64.9 | 37.9 | 28.5 | 47.3 | 44.6 |
| QVQ-72B-Preview | 70.3 | 34.9 | 48.2 | 70.3 | 39.0 | 30.7 | 58.2 | 50.2 |
| Qwen2.5-VL-72B | 74.2 | 39.3 | 47.3 | 68.2 | 49.1 | 35.9 | 55.7 | 52.8 |
| VL-Rethinker-72B | 78.2 | 42.2 | 54.6 | 67.2 | 49.2 | 34.9 | 56.6 | 54.7 |
| InternVL3-78B | **79.0** | 43.1 | 51.0 | 72.2 | 46.0 | 35.1 | 55.9 | 54.6 |
| **Verification-augmented and Tool-enabled MLLMs** | | | | | | | | |
| Qwen2.5-VL-7B‡ (Bo8) | 76.8 | 31.6 | 46.8 | 58.1 | 43.1 | 29.7 | 48.6 | 47.8 |
| SRPO-7B | 75.8 | 32.9 | - | 57.1 | - | - | - | - |
| ReVPT-7B | 66.0 | - | - | - | - | - | - | - |
| DeepEyes-7B | 70.1 | 26.6 | 47.3 | - | 38.9 | - | 47.7 | - |
| **Prior Caption-then-Reason Methods** | | | | | | | | |
| ECSO | 64.6 | 42.7 | 42.8 | 61.4 | 38.4 | 25.0 | 39.4 | 44.9 |
| OmniCaptioner | 67.5 | 43.3 | 48.0 | 62.2 | 38.7 | 30.5 | 56.2 | 49.5 |
| **Qwen2.5-VL series and our RAPID-enhanced counterparts** | | | | | | | | |
| Qwen2.5-VL-3B | 64.5 | 21.9 | 28.8 | 50.1 | 24.2 | 13.4 | 39.6 | 34.6 |
| w/ RAPID (Qwen3-8B) | 69.6 | 40.8 | 48.6 | 60.9 | 39.1 | 29.3 | 56.4 | 49.2 |
| Qwen2.5-VL-7B | 70.3 | 25.8 | 41.0 | 57.3 | 34.4 | 19.4 | 46.1 | 42.0 |
| w/ RAPID (Qwen3-8B) | 76.1 | 43.7 | 52.2 | 64.7 | 45.4 | 32.7 | 57.7 | 53.2 |
| Qwen2.5-VL-32B | 76.8 | 37.8 | 50.1 | 69.0 | 43.1 | 33.3 | 55.0 | 52.2 |
| w/ RAPID (Qwen3-8B) | 76.8 | 47.0 | 54.4 | 67.8 | 48.5 | 36.5 | **60.4** | 55.9 |
| w/ RAPID (GPT-A5B)* | 75.9 | 52.1 | 54.3 | 69.8 | 50.8 | 38.3 | **60.4** | 57.4 |
| Qwen2.5-VL-72B | 74.2 | 39.3 | 47.3 | 68.2 | 49.1 | 35.9 | 55.7 | 52.8 |
| w/ RAPID (GPT-A5B)* | 75.1 | **53.4** | **56.2** | 72.4 | **52.1** | 37.9 | 59.1 | **58.0** |

## 4 EXPERIMENTS

### 4.1 MAIN RESULTS

**Baselines.** We compare our method with the following baselines: (i) Proprietary models, including Claude-3.7-Sonnet (Anthropic, 2025), Gemini-2.0-Flash (DeepMind, 2025) and GPT-4o (Hurst et al., 2024); (ii) Open-source general-purpose MLLMs, including Qwen2.5-VL (3B/7B/32B/72B) (Bai et al., 2025), InternVL3 (8B/14B/38B/78B) (Zhu et al., 2025), Gemma-3 (12B/27B) (Team et al., 2025a) and Ovis2 (8B/16B/34B) (Lu et al., 2024b); (iii) Open-source MLLMs specialized for reasoning, including MM-Eureka (7B/32B) (Meng et al., 2025), VL-Rethinker (7B/72B) (Wang et al., 2025a), QVQ-72B-Preview (Qwen, 2024) and ReVisual-R1-7B (Chen et al., 2025b); (iv) Latest Caption-then-Reason pipelines, such as ECSO (Gou et al., 2024) and OmniCaptioner (Lu et al., 2025). We use the GRPO-optimized Qwen2.5-VL-7B as captioner and Qwen3-8B as the reasoner; (v) Verification-based methods such as Qwen2.5-VL-7B‡ (Bo8) that performs best-of-8 sampling with VisualPRM-8B-v1.1 (Wang et al., 2025b), SRPO-7B (Wan et al., 2025) that is instructed to perform reflection after reasoning , and Tool-enabled MLLMs such as ReVPT-7B (Zhou et al., 2025) and DeepEyes-7B (Zheng et al., 2025) that call tools for better perception.

Table 2: **Ablation study of different components of RAPID** (with Qwen2.5-VL-7B by default). VPO$^\dagger$: VPO without the caption penalty; $^\ddagger$: using `cap+sol` for reasoning-perception decoupling.

| | Decouple | GRPO | VPO$^\dagger$ | Cap. penalty | Math Vista | Math Vision | Math Verse | MMMU | We Math | Dyna Math | Logic Vista | AVG |
|---|---|---|---|---|---|---|---|---|---|---|---|---|
| Ⓐ | | | | | 70.3 | 25.8 | 41.0 | 57.3 | 34.4 | 19.4 | 46.1 | 42.0 |
| Ⓑ | ✓ | | | | 70.0 | 40.4 | 45.2 | 62.0 | 39.1 | 26.3 | 49.7 | 47.5 |
| Ⓒ | ✓ | ✓ | | | 72.7 | 43.2 | 50.0 | 63.3 | 41.1 | 28.7 | 54.1 | 50.5 |
| Ⓓ | ✓ | ✓ | ✓ | | 76.0 | 41.5 | 50.6 | 62.9 | 43.1 | **33.1** | 57.7 | 52.2 |
| Ⓔ | ✓ | ✓ | ✓ | ✓ | **76.1** | 43.7 | **52.2** | **64.7** | **45.4** | 32.7 | 57.7 | **53.2** |
| Ⓕ | ✓$^\ddagger$ | ✓ | ✓ | ✓ | 71.2 | **43.8** | 48.1 | 64.6 | 39.9 | 32.3 | **57.9** | 51.1 |
| Ⓖ | ✓ | | ✓ | ✓ | 74.2 | 42.5 | 50.8 | 62.0 | 39.4 | 31.9 | 56.6 | 51.1 |
| Ⓗ | | ✓ | | | 74.2 | 29.7 | 44.8 | 55.9 | 41.0 | 27.7 | 48.1 | 45.9 |
| Ⓘ | | ✓ | ✓ | ✓ | 75.0 | 29.8 | 42.0 | 55.8 | 40.8 | 23.0 | 46.3 | 44.7 |

**Evaluation** is conducted on a diverse set of multi-modal reasoning benchmarks, *e.g.*, MathVista (testmini) (Lu et al., 2024a), MathVision (test) (Wang et al., 2024b), MathVerse (vision-only) (Zhang et al., 2024), MMMU (val) (Yue et al., 2024), WeMath (Qiao et al., 2024), DynaMath (Zou et al., 2024), and LogicVista (Xiao et al., 2024). As in recent works (Wang et al., 2025c; Zhu et al., 2025), we use VLMEvalKit (Duan et al., 2024) for evaluation, and report the worst-case accuracy for DynaMath, the strict accuracy for WeMath, and overall accuracy for the other benchmarks.

**Implementation Details of RAPID.** We perform RAPID upon the Qwen2.5-VL series (3B, 7B, 32B, and 72B) MLLMs. During training, we use R1-Distilled-7B (R1-7B) as the reasoner to compute reward signals for all MLLMs. During evaluation, we adopt Qwen3-8B [3] (Yang et al., 2025a) and GPT-OSS-120B (Agarwal et al., 2025) as the LLM reasoners. For training data, we adopt ViRL39K (Wang et al., 2025a), a curated dataset of 38,870 verifiable multi-modal question-answer pairs tailored for multi-modal reasoning. We implement GRPO and VPO with verl (Sheng et al., 2025) with a global batch size of 256, a rollout temperature of 1.0, and a learning rate of $1e^{-6}$.

**Implementation Details of GRPO.** We set the number of rollouts to 8 for the 3B/7B MLLMs and 4 for the 32B/72B MLLMs. Following Yu et al. (2025), we remove KL regularization and use the "Clip-Higher" strategy, setting $\epsilon_l$ to 0.2 and $\epsilon_h$ to 0.25. When reporting performance with GPRO but without VPO (*e.g.*, Ⓒ and Ⓗ in Table 2, and the baselines in Table 3), we select the best-performing checkpoints (with perception-reasoning decouple applied) at 400, 300, and 100 steps for the 3B, 7B, and 72B MLLMs, respectively, based on the average accuracies across the seven reasoning datasets (evaluated every 50 steps). GRPO is not applied to the 32B variant, as it has already been RL-tuned.

**Implementation Details of VPO.** We set the number of rollouts to 4, KL-penalty coefficient $\beta$ to $1e^{-3}$, and penalizing constant $\lambda$ in Eq. (3) to 0.1. VPO is applied to the MLLM following 200 steps of GRPO (except for the 32B model, which we directly use the original model).[4] Similar to GRPO, we select the best checkpoints at 200, 150, 100, and 100 for the 3B, 7B 32B, and 72B models, respectively, according to their average accuracies on the reasoning datasets.

**Results.** Table 1 compares the performance of RAPID and baseline MLLMs on seven multi-modal reasoning datasets. It highlights two merits of RAPID: (i) **It achieves significant performance gains on the reasoning tasks compared to the original MLLMs.** For example, applying RAPID to Qwen2.5-VL-7B with a similar-sized LLM (Qwen3-8B) yields an average accuracy of 53.2% (+11.2% compared to the original MLLM). Notably, when applying RAPID to Qwen2.5-VL-72B with GPT-OSS-120B as the LLM, we achieve the best average accuracy of 58% across all the models compared, even surpassing proprietary MLLMs. (ii) **RAPID achieves better**

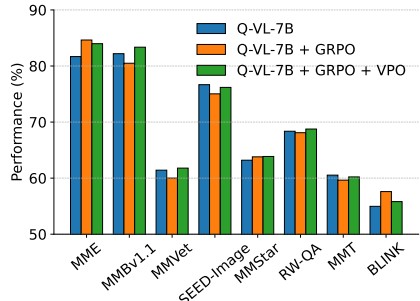

Figure 5: **General benchmark Results.**

**performance-size trade-off.** For example, Qwen2.5-VL-7B with RAPID (Qwen3-8B) with a total size of 15B outperforms larger models such as MM-Eureka-32B, InternVL3-38B and Ovis2-34B. Similarly, Qwen2.5-VL-32B with RAPID (Qwen3-8B) surpasses VL-Rethinker-72B and InternVL3-78B. Check Figure 1 for a better visualization of the performance-size trade-off. More analysis on the **training and inference compute efficiencies** is provided in Appendix E. (iii) **RAPID surpasses**

---

[3]We do not use R1-7B as we found the similar-sized Qwen3-8B performs better in Sec. 4.3.

[4]For the 3B model, we observe that training with VPO after GRPO results in slight forgetting of reasoning. To mitigate this, we switch back to GRPO optimization for an additional 100 steps after VPO.

**latest caption-then-reason methods** (Gou et al., 2024; Lu et al., 2025), mainly due to the usage of tentative responses and VPO.

**Evaluation on General Benchmarks** Although RAPID is specifically designed for multi-modal math and science reasoning, we verify that it does not hurt general abilities. We evaluated the VPO/GRPO-optimized Qwen2.5-VL-7B on general benchmarks in a "non-thinking" mode, per common protocols (Yang et al., 2025a; Wang et al., 2025d). As shown in Figure 5, its performance remains on par with the original model. This confirms that our method is a targeted enhancement for reasoning that preserves the model's general abilities. (Benchmark details and Qwen2.5-VL-3B results are in Appendix F.1).

## 4.2 ABLATION STUDIES

In this section, we first investigate the effectiveness of the proposed components, *i.e.*, reasoning-perception decoupling (denoted "Decouple") and VPO. For VPO, we ablate the choices on reward computation during training. Next, we assess the generalization and scalability of RAPID to different LLMs. We use the same training configurations as in Sec. 4.1. Qwen2.5-VL-3B/7B are adopted for ablations due to resource constraints. Unless otherwise specified, we use R1-7B as the default LLM for training (reward computation) and Qwen3-8B for inference.

**Reasoning-Perception Decoupling & VPO.** Table 2 presents a detailed ablation study of RAPID's with the 7B MLLM (see Appendix F.2 for the 3B MLLM), which we analyze by incrementally adding each one to the baseline. Starting from the baseline MLLM (Ⓐ), we first apply **perception-reasoning decoupling**. This step alone (Ⓑ) yields a significant 5.5% average improvement, demonstrating the immediate benefit of leveraging a stronger external LLM (Qwen3-8B) for reasoning. Building on this, we apply **GRPO** to enhance the MLLM's perception by optimizing its tentative solutions, which adds another 3.0% to the average score (Ⓒ). We then apply **VPO** without the caption penalty (Ⓓ), and achieves a further 1.7% gain. Finally, incorporating the **caption penalty** leads to our full model (Ⓔ), adding another 1.0%. This brings the total improvement from our full VPO method to 2.7% over the model with only GRPO (Ⓒ). This caption penalty is crucial for VPO's effectiveness, as it prevents reward-hacking where the model might generate solutions instead of the intended captions. Figure 7 confirms this: without the penalty, the ratio of rollouts containing valid captions diminishes rapidly, whereas with the penalty, it remains stable above 95% before 150 steps.

The results also highlight that GRPO and VPO are complementary, whose synergy is evident in two ways: 1) Removing either method from the full decoupled setup (Ⓖ vs. Ⓔ, and Ⓒ vs. Ⓔ) results in suboptimal performance, confirming both are necessary. 2) Figure 6 further visualizes this dynamics: after initial gains from GRPO begin to plateau, VPO provides a distinct secondary performance boost. Despite these gains, the decoupling strategy remains the most critical element. An MLLM improved by VPO and GRPO alone (Ⓘ) still lags far behind the decoupled version (Ⓔ), underscoring its importance. We also note that VPO does not improve MLLM's reasoning capabilities on its own (Ⓗ vs. Ⓘ); the slight performance drop suggests minor forgetting during sequential training. How-

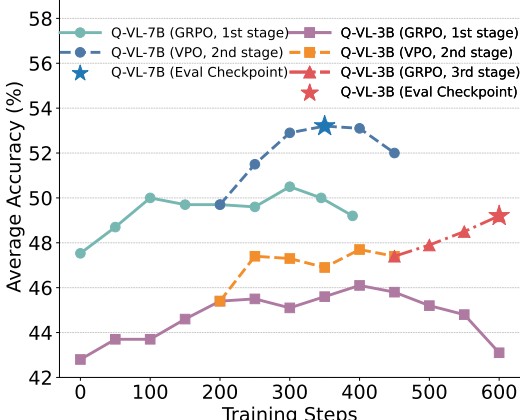

Figure 6: **Training dynamics of GRPO and VPO.** Performance is evaluated under the perception-reasoning decouple pipeline.

ever, we demonstrate in Appendix E.4 that this issue can be addressed by simple GPRO training without impacting the overall performance of the 7B model.

While standard captions (cap+sol) initially outperform query-relevant ones (qcap+sol) as discussed in Section 3.1, this trend reverses after applying VPO (see Ⓔ and Ⓕ). We hypothesize that this is because qcap is easier to optimize during VPO, as the query guides the MLLM to focus on relevant visual details. Instead, without such guidance, the MLLM struggles to identify pertinent information for cap. This is confirmed by Figure 8, which shows that training rewards for qcap increase steadily, while rewards for cap oscillate without consistent growth.

**Choices on Reward Computation.** We study VPO reward computation by varying the reasoning LLM (during training) and its input content. We take the best-performing checkpoint (optimized

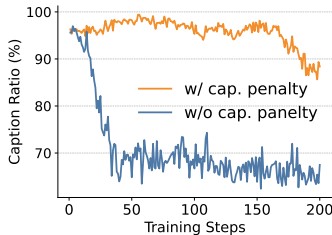 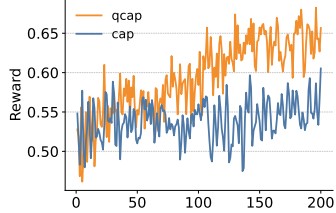 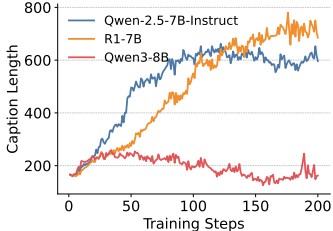

Figure 7: **Reward hacking without the caption penalty.**

Figure 8: **Reward dynamics of adopting `qcap` and `cap`.**

Figure 9: **Caption length trend with various LLMs.**

with GRPO) as the baseline and evaluate under the perception-reasoning decoupled paradigm. First, we keep the input content as `qcap+sol` and test three LLMs with increasing reasoning capacity: Qwen2.5-7B-Instruct (weak), R1-7B (intermediate), and Qwen3-8B (strong), respectively.

As shown in Table 3, the R1-7B LLM performs best. We hypothesize this is due to a trade-off in reasoning capacity, reflected in caption lengths. Figure 9 shows the caption lengths during training for various LLMs. We speculate that the stronger Qwen3-8B can succeed with succinct captions, thus inadvertently rewarding short captions that miss details[5], while the weaker Qwen2.5-Instruct-7B incentivizes over-long captions that even include inaccurate solutions. R1-7B strikes an effective balance, making it our default choice.

Table 3: **Ablation studies** on (a) LLM types and (b) input to LLM for reward computation.

| Configuration | Q-VL-3B | Q-VL-7B |
|---|---|---|
| w/o VPO (baseline) | 46.1 | 50.5 |
| *(a) LLM Types* | | |
| Qwen2.5-7B-Instruct | 47.5 | 51.9 |
| R1-Distilled-7B | **47.9** | **53.2** |
| Qwen3-8B | 47.1 | 51.3 |
| *(b) LLM Input Contents* | | |
| `qcap+sol` | 47.4 | 49.5 |
| `qcap` | **47.9** | **53.2** |

Next, we examine the choice of input content: using caption alone (`qcap`) versus using the caption plus a tentative solution (`qcap+sol`). As in Table 3, using only the caption is superior. Including tentative solutions allows the LLM to take a shortcut during training—relying on the solutions while ignoring captions—which generates a noisy reward signal ineffective for optimizing caption quality.[6] Additionally, we explore fine-tuning the LLM for better reasoning ability in Appendix E.6.

## 4.3 GENERALIZATION AND SCALING WITH DIFFERENT LLMs.

A practical requirement is that our MLLM, once optimized, should generalize to new, unseen LLMs at inference time without retraining. To test this, we perform VPO on the GRPO-trained MLLM using only R1-7B as the LLM for this optimization step. We then evaluate its performance against the baseline (the MLLM without VPO) by pairing both MLLMs with a diverse range of different LLMs at inference time, as shown in Figure 10.

We have three observations. *First, the performance gain from VPO generalizes effectively.* The gap between the VPO-trained model (solid curves) and the baseline (dashed curves) is maintained or widened when using stronger LLMs, revealing that the benefit is not confined to the specific LLM used for training. *Second, the RAPID's scalability is evident as absolute performance trends upward when using more capable LLMs, although this improvement is not strictly monotonic with model size.* Additionally, among the LLMs tested, Qwen3-8B strikes the best balance between performance and model size, establishing it to be our default choice for the inference stage. Note that the *optimal LLMs for training and inference might differ* (*c.f.*, Table 3 and Figure 10).

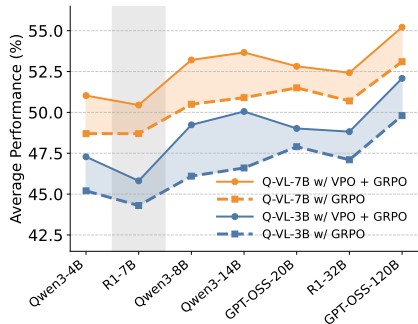

Figure 10: **Performance with different LLMs.** (Only R1-7B is used in training)

## 4.4 EVALUATION ON CAPTION QUALITY

---

[5]We further validate this in Appendix E.5.

[6]Notably, this optimal input format differs from that used in the perception-reasoning decoupling stage.

Table 4: **Effectiveness of different components of RAPID** (with InternVL3-8B by default). VPO[†]: VPO without the caption penalty.

| Decouple | GRPO | VPO[†] | Cap. penalty | Math Vista | Math Vision | Math Verse | MMMU | We Math | Dyna Math | Logic Vista | AVG |
|---|---|---|---|---|---|---|---|---|---|---|---|
| | | | | 73.6 | 29.3 | 39.8 | 62.7 | 37.1 | 25.5 | 44.1 | 44.6 |
| ✓ | | | | 71.3 | 42.4 | 39.3 | 64.4 | 38.8 | 29.1 | 50.1 | 47.9 |
| ✓ | ✓ | | | 73.2 | 42.6 | 44.3 | 64.4 | 40.2 | 31.1 | 52.1 | 49.7 |
| ✓ | ✓ | ✓ | ✓ | **75.4** | **43.2** | **48.5** | **64.6** | **43.2** | **33.2** | **55.5** | **51.9** |

We validate the improved quality of the captions generated by VPO-optimized MLLMs via a pairwise comparison (Chen et al., 2023; Liu et al., 2024b) using GPT-4o (OpenAI, 2024) as a judge. With 1000 random samples per dataset, GPT-4o compares captions from Qwen2.5-VL-3B trained with and without VPO. The judge is instructed to prefer captions with more comprehensive and accurate details required to answer the question, while excluding any solving process (the prompt is in Appendix F.3). We alternate the caption order to mitigate position bias (Zheng et al., 2023). As shown in Figure 11, the VPO-optimized MLLM's captions shows a clear advantage across all datasets, highlighting the VPO's effectiveness (Check the case study on caption quality in Appendix I). Moreover, we extend this comparison to other MLLMs and validate these findings with human assessment in Appendix F.3.

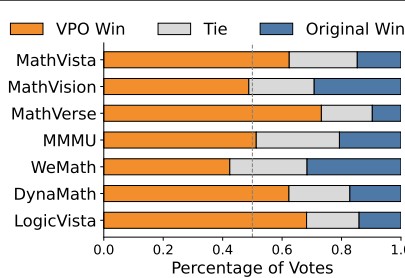

Figure 11: **Pairwise comparison** on the caption quality with and without VPO.

### 4.5 GENERALIZATION ACROSS MLLMS

We confirm RAPID's generalizability across various MLLMs. Applying decoupling (with Qwen3-8B) and VPO to InternVL3-8B yields significant gains (Table 4), mirroring our main results (Table 2). This suggests that RAPID does generalize across different MLLMs. Moreover, applying the decoupling pipeline alone to more MLLMs (*i.e.*, InternVL3-8B, VL-Rethinker-7B and MM-Eureka-7B) with different LLMs (*i.e.*, Qwen3-8B and GPT-OSS-120B) also shows consistent improvements (Table 17), indicating that decoupling is a broadly effective strategy for enhancing MLLM performance.

### 4.6 COST ANALYSIS

This section analyzes the cost for GRPO and VPO training, based on NVIDIA H20 GPU rentals from LuchenTech[7] at approximately $0.56 per GPU/hour. Table 5 details the resource requirements. Our method is remarkably cost-effective, requiring less than $300 for significant performance gains (see Table 1). This low cost ensures accessibility for small teams and individual researchers. Furthermore, as detailed in Appendix E.1, these expenses are less than 0.1% of the cost to pre-train or fine-tune an MLLM with a new LLM, making RAPID a highly practical alternative.

Table 5: **Training time and cost analysis** for different model sizes and training phases in RAPID.

| Model Size | Stage | Wall-Clock Time (hours) | Training Steps | Hardware (GPUs) | GPU Hours | Estimated Total Cost (USD) |
|---|---|---|---|---|---|---|
| 3B | GRPO | 16.7 | 200 | 8 × H20 | 133.6 | $74.8 |
| | VPO | 23.9 | 200 | 16 × H20 | 382.4 | $214.1 |
| 7B | GRPO | 24.0 | 200 | 8 × H20 | 192.0 | $107.5 |
| | VPO | 16.3 | 150 | 16 × H20 | 260.0 | $145.6 |
| 32B | VPO | 13.6 | 100 | 32 × H20 | 435.2 | $243.7 |

### 5 CONCLUSION

This paper proposes RAPID, an efficient method for constructing multi-modal reasoning models. By decoupling visual perception (MLLM) from text-only reasoning (LLM), RAPID leverages the advanced reasoning of frontier LLMs while avoiding burdensome visual re-alignment. Enhanced with Visual Perception Optimization, this method reinforces precise captions to provide rich visual context, improving reasoning and effectively scaling to more advanced LLMs at inference time. Our approach achieves significant accuracy gains on multiple multi-modal reasoning benchmarks while remaining computationally efficient.

---

[7]https://www.luchentech.com/

## ETHICS STATEMENT

We affirm that this work adheres to the ICLR Code of Ethics.[8] Our research does not involve human subjects, personal data, or sensitive attributes, and it does not pose foreseeable risks of physical, psychological, or social harm. All datasets used in our experiments are publicly available and widely adopted in the research community. We have carefully considered potential issues related to bias, fairness, and misuse, and we believe that the scope of this study does not introduce additional ethical concerns. Furthermore, our work complies with all relevant legal, institutional, and research integrity requirements, and we declare that there are no conflicts of interest or competing financial relationships that could have influenced this research.

## REPRODUCIBILITY STATEMENT

We have taken extensive steps to ensure the reproducibility of our work. All model architectures, training procedures, and hyperparameters are described in detail in the main text (Section 3 and Section 4) and the Appendix. For empirical results, we specify dataset sources and preprocessing steps in Section 4.1, and we provide implementation details and experimental settings in Section 4.1. Anonymous source code and scripts for reproducing the main experiments will be made available in the link provided in the abstract. We believe these efforts enable full reproducibility of our reported results.

## ACKNOWLEDGMENT

This work was supported by National Natural Science Foundation of China under Grant no. 62136005 and Shenzhen fundamental research program JCYJ20250604144724032. This research was also supported in part by the Research Grants Council of the Hong Kong Special Administrative Region (Grants 16202523 and HKU C7004-22G).

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

APPENDIX

CONTENTS

## A  LIMITATION

**Auto-thinking.**  RAPID is specifically designed for multi-modal reasoning, and it is appealing to explore how to flexibly switch between fast and slow thinking dependent on the complexity of input queries, without human prior and prompt engineering.

**Domain-Specific Design.**  The RAPID architecture, in its current form, is optimized for multi-modal math and science reasoning. Its extension to other domains, such as spatial reasoning (Wang et al., 2024a), would require more than re-evaluation; it would demand specific adaptations to the model itself. The experiments on general benchmarks reported in Sec. 4.1 do not test the full reasoning

Table 6: **Model Configurations for each RAPID-enhanced model group in Figure 1**. "-" denotes the corresponding item (*e.g.*, LLM reasoner, GRPO/VPO training) is not applied.

| Size (B) | LLM | GRPO | VPO | Avg. Performance |
|---|---|---|---|---|
| *Qwen2.5-VL-3B* | | | | |
| 7 | Qwen3-4B | ViRL39K | ViRL39K | 47.3 |
| 11 | Qwen3-8B | ViRL39K | ViRL39K | 49.2 |
| 17 | Qwen3-14B | ViRL39K | ViRL39K | 50.1 |
| *Qwen2.5-VL-7B* | | | | |
| 11 | Qwen3-4B | ViRL39K | ViRL39K | 51.0 |
| 15 | Qwen3-8B | ViRL39K | ViRL39K | 53.2 |
| 21 | Qwen3-14B | ViRL39K | ViRL39K | 53.6 |
| *Qwen2.5-VL-32B* | | | | |
| 36 | Qwen3-B | - | ViRL39K | 54.6 |
| 40 | Qwen3-8B | - | ViRL39K | 55.9 |
| 152 | GPT-OSS-120B | - | ViRL39K | 57.4 |

Table 7: **Index of Prompt templates for RAPID.**

| Component | Purpose | Notation | Prompt Template |
|---|---|---|---|
| MLLM | Standard captions | $P_{cap}$ | Figure 13 |
| MLLM | Query-relevant captions | $P_{qcap}$ | Figure 14 |
| MLLM | Tentative response | $P_{sol}$ | Figure 15 |
| MLLM | Caption Penalty | - | Figure 17 |
| LLM Reasoner | Inference | $P_{reason}$ | Figure 12 |
| LLM Reasoner | Reward computation | $P'_{reason}$ | Figure 16 |

pipeline, as the LLM was not activated. Therefore, investigating the adaptations required to generalize RAPID remains a key open direction for future research.

**Adapting the LLM.** In the current implementation of RAPID, the LLM functions as a static reasoning module with its parameters kept frozen. A valuable direction for future work is to explore adapting the LLM to our perception-reasoning framework, potentially through methods like supervised fine-tuning or reinforcement learning. While this would introduce additional computational overhead for training, it remains an open empirical question whether the potential performance gains would justify the increased cost.

## B  MODEL CONFIGURATION FOR FIGURE 1

For each RAPID-enhanced model group (*e.g.*, Qwen2.5-VL-3B (**RAPID**)) in Figure 1, we train the original model (*e.g.*, Qwen2.5-VL-3B) using both the proposed VPO and GRPO objectives, where the former encourages the MLLM to generate query-relevant captions with higher quality while the later optimizes it to give better reasoning traces. The details for VPO and GRPO can be found in Sec. 4.1 and Appendix D. We then pair the trained MLLM with different LLMs under our RAPID. Table 6 shows the configuration for each RAPID-enhanced model group. Specifically, for each model in the group, we report the total model size (B), paired LLMs, data used to conduct GRPO and VPO and the average performance across 7 tasks. Note that the results for Qwen2.5-VL-32B group does not involve GRPO training because it has already undergone an RL stage before release (Bai et al., 2025). Results for other MLLMs are consistent with Table 1.

## C  PROMPT TEMPLATES

In Table 7, we provide an index of the prompt templates used in RAPID.

**System Prompt:**
You are a helpful assistant.

**User Prompt:**
In the following text, you will receive a detailed caption of an image and a relevant question. In addition, you will be provided with a tentative model response. You goal is to answer the question using these information.

### The detailed caption of the provided image: {}

### A problem to be solved: {}

### A tentative model response. {}

### Note that the above tentative response might be inaccurate (due to calculation errors, incorrect logic/reasoning and so on), under such a case, please ignore it and give your own solutions. However, if you do not have enough evidence to show it is wrong, please output the tentative response.

Figure 12: **Prompt templates used by the reasoner LLM for inference.**

**System Prompt:**
You are a helpful assistant.

**User Prompt:**
Describe this image in detail.

Figure 13: **Prompt templates used by the MLLM to obtain the standard captions.**

**System Prompt:**
You are given an image and a relevant question. Based on the query, please describe the image in detail. Do not try to answer the question.

**User Prompt:**
Question: {}
Please describe the image. DO NOT try to answer the question!

Figure 14: **Prompt templates used by the MLLM to obtain the query-relevant captions.**

**System Prompt:**
You are a helpful assistant.

**User Prompt:**
{}

Figure 15: **Prompt templates used by the MLLM to obtain the tentative response.** The placeholder is for the question.

> **System Prompt:**
> You are a helpful assistant.
>
> **User Prompt:**
> ### The detailed caption of the provided image: {}
>
> ### Question: {}
>
> Please think step by step. The final answer MUST BE put in \\boxed{}.

Figure 16: **Prompt templates used by the reasoner LLM for training.**

## D    FORMULATIONS OF GRPO

GRPO (Shao et al., 2024) is a policy optimization algorithm originally developed to enhance the reasoning capability of text-only LLMs. In our setting, the policy $\pi_\theta$ to optimize becomes the MLLM. For a given input pair $(I, q)$ of image and text question from the training set $p_{\mathcal{D}}$, the old policy generates $G$ rollouts, *i.e.*, $o \sim \pi_{\theta_{\mathrm{old}}}(I, P_{\mathrm{sol}}(q))$. Denoting $R_i$ as the reward for the $i$-th rollout, the normalized advantage is $\hat{A}_i = \frac{R_i - \bar{R}}{\sigma(R)}$, where $\sigma(R)$ denotes the standard deviation of rewards within the group and the baseline reward is $\bar{R} = \frac{1}{G} \sum_{i=1}^{G} R_i$. The objective incorporates a surrogate loss clipped within $[1 - \epsilon, 1 + \epsilon](\epsilon > 0)$ and a KL-penalty $D_{\mathrm{KL}}[\pi_\theta | \pi_{\theta_{\mathrm{ref}}}]$ weighted by $\beta$ (not shown here) to stabilize optimization:

$$L(\theta) = \mathbb{E}_{(I,q) \sim p_{\mathcal{D}}, o \sim \pi_{\theta_{\mathrm{old}}}(\cdot | I, P_{\mathrm{sol}}(q))}$$
$$\left[ \frac{1}{G} \sum_{i=1}^{G} \min \left( \frac{\pi_\theta(o_i \mid I, P_{\mathrm{sol}}(q))}{\pi_{\theta_{\mathrm{old}}}(o_i \mid I, P_{\mathrm{sol}}(q))} \hat{A}_i, \mathrm{clip} \left( \frac{\pi_\theta(o_i \mid I, P_{\mathrm{sol}}(q))}{\pi_{\theta_{\mathrm{old}}}(o_i \mid I, P_{\mathrm{sol}}(q))}, 1 - \epsilon, 1 + \epsilon \right) \hat{A}_i \right) \right]. \tag{4}$$

The reward $R_i$ for the $i$-th rollout is expressed as: $R_i = r(y_{\mathrm{gt}}, o_i) = \mathbb{1}(y_{\mathrm{gt}} = \mathrm{parse}(o_i))$, where $y_{\mathrm{gt}}$ denotes the ground-truth answer of a reasoning question and $\mathbb{1}(\cdot)$ is an indicator function that outputs 1 if the final parsed prediction matches the ground-truth and 0 otherwise.

## E    MORE ANALYSIS

### E.1    ANALYSIS ON THE TRAINING COMPUTE EFFICIENCY OF RAPID

RAPID's decoupled design enables the flexible adoption of recent LLM reasoners, such as Qwen3 (Yang et al., 2025a), without retraining. This raises a critical question: *how does our efficient approach compare against models that require full and costly retraining of their visual-language alignment to integrate the latest LLMs?*

To investigate this trade-off between performance and computational cost, we compare RAPID with two leading MLLMs also built on the Qwen3-8B LLM: Keye-VL (Team et al., 2025b) and InternVL3.5 (Wang et al., 2025d). Table 8 presents a comparative analysis, reporting average accuracy across seven multi-modal reasoning tasks alongside the training tokens and estimated training FLOPs (calculated as model size × training tokens). The number of training tokens for Keye-VL-8B and InternVL3.5-8B are sourced from their respective technical reports.

As can be seen, although still inferior to the end-to-end methods, RAPID with Qwen2.5-VL-7B can achieve 90.8% of Keye-VL-8B performance with $1250\times$ less training FLOPs, and 88.2% of InternVL3.5-8B performance with $864.2\times$ training cost reduction. Thanks to the remarkable training efficiency, we can adopt larger MLLMs such as Qwen2.5-VL-32B, we achieve comparable (92.7%) performance with InternVL3.5-8B but with $1025\times$ less training FLOPs

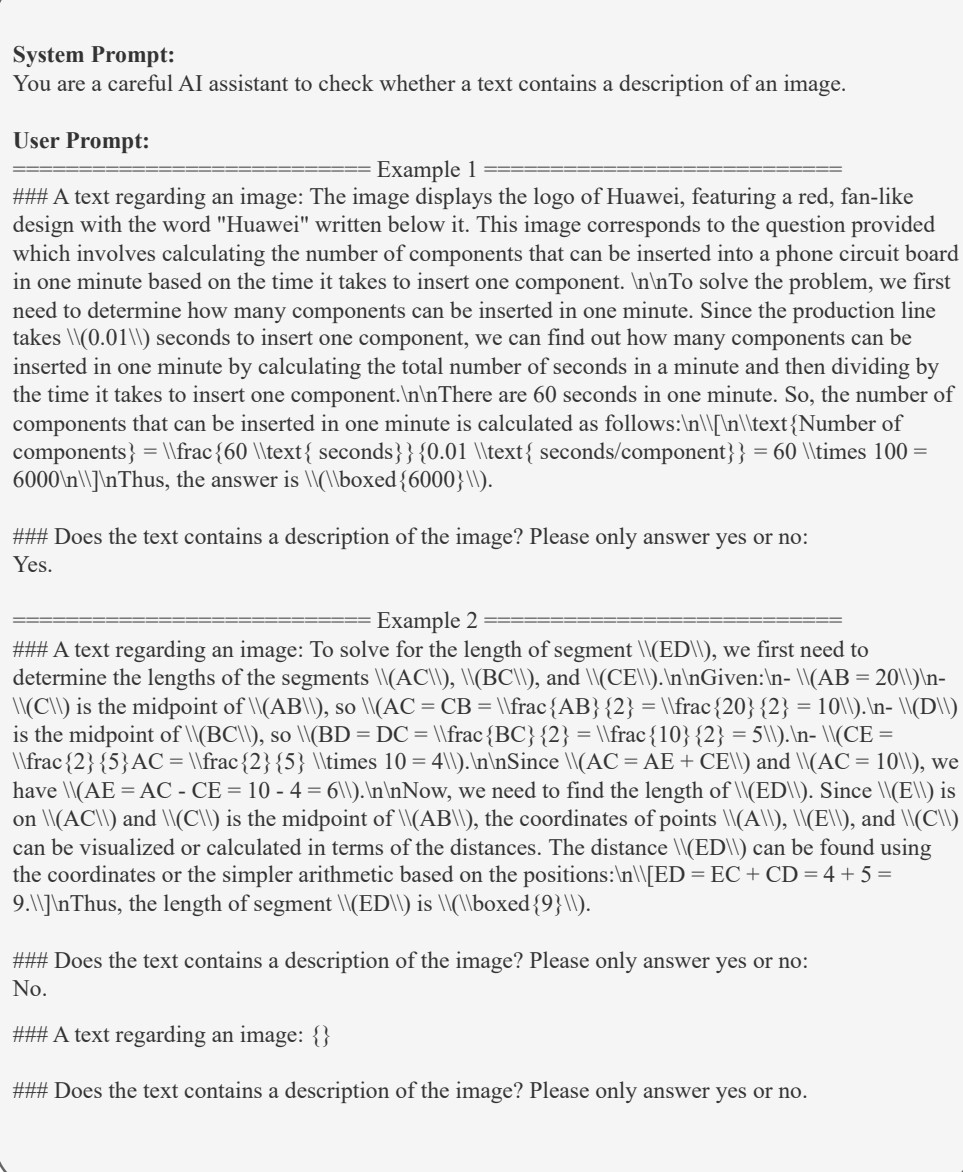

**System Prompt:**
You are a careful AI assistant to check whether a text contains a description of an image.

**User Prompt:**
========================= Example 1 =========================
### A text regarding an image: The image displays the logo of Huawei, featuring a red, fan-like design with the word "Huawei" written below it. This image corresponds to the question provided which involves calculating the number of components that can be inserted into a phone circuit board in one minute based on the time it takes to insert one component. \n\nTo solve the problem, we first need to determine how many components can be inserted in one minute. Since the production line takes \\(0.01\\) seconds to insert one component, we can find out how many components can be inserted in one minute by calculating the total number of seconds in a minute and then dividing by the time it takes to insert one component.\n\nThere are 60 seconds in one minute. So, the number of components that can be inserted in one minute is calculated as follows:\n\\[\n\\text{Number of components} = \\frac{60 \\text{ seconds}}{0.01 \\text{ seconds/component}} = 60 \\times 100 = 6000\n\\]\nThus, the answer is \\(\\boxed{6000}\\).

### Does the text contains a description of the image? Please only answer yes or no:
Yes.

========================= Example 2 =========================
### A text regarding an image: To solve for the length of segment \\(ED\\), we first need to determine the lengths of the segments \\(AC\\), \\(BC\\), and \\(CE\\).\n\nGiven:\n- \\(AB = 20\\)\n- \\(C\\) is the midpoint of \\(AB\\), so \\(AC = CB = \\frac{AB}{2} = \\frac{20}{2} = 10\\).\n- \\(D\\) is the midpoint of \\(BC\\), so \\(BD = DC = \\frac{BC}{2} = \\frac{10}{2} = 5\\).\n- \\(CE = \\frac{2}{5}AC = \\frac{2}{5} \\times 10 = 4\\).\n\nSince \\(AC = AE + CE\\) and \\(AC = 10\\), we have \\(AE = AC - CE = 10 - 4 = 6\\).\n\nNow, we need to find the length of \\(ED\\). Since \\(E\\) is on \\(AC\\) and \\(C\\) is the midpoint of \\(AB\\), the coordinates of points \\(A\\), \\(E\\), and \\(C\\) can be visualized or calculated in terms of the distances. The distance \\(ED\\) can be found using the coordinates or the simpler arithmetic based on the positions:\n\\[ED = EC + CD = 4 + 5 = 9.\\]\nThus, the length of segment \\(ED\\) is \\(\\boxed{9}\\).

### Does the text contains a description of the image? Please only answer yes or no:
No.

### A text regarding an image: {}

### Does the text contains a description of the image? Please only answer yes or no.

Figure 17: **Prompt templates used by the MLLM for caption panelty.**

Table 8: **Training cost comparison**. *As we did not apply GRPO to Qwen2.5-VL-32B, it consumes less training tokens (100M) than Qwen2.5-VL-7B (550M).

| Method | AVG Accuracies | # Tokens | FLOPs (Ratio) |
|---|---|---|---|
| Keye-VL-8B (Team et al., 2025b) | 58.6 | 600B | 1500× |
| InternVL3.5-8B (Wang et al., 2025d) | **60.3** | 410B | 1025× |
| Qwen2.5-VL-7B w/ RAPID (Qwen3-8B) | 53.2 | 550M | 1.2× |
| Qwen2.5-VL-32B w/ RAPID (Qwen3-8B) | 55.9 | 100M* | 1× |

**System Prompt:**
Please act as an impartial judge and evaluate the quality of the captions provided by two multi-modal AI assistants regarding an image and a question. You will receive an image and a question regarding it. You should choose the caption that (i) more accurately and comprehensively reflect the content in the image and (ii) contain more details/facts required to solve the question. (iii) contain less visual hallucination (describing objects not shown in the image). Note that the solution to the problem is not considered as facts or details! Note that if the caption contains any solution process, you should ignore it (completely delete it) and only consider the remaining when conducting your evaluation. Begin your evaluation by comparing the two captions and provide a short explanation. Avoid any position biases and ensure that the order in which the captions were presented does not influence your decision. Do not allow the length of the captions to influence your evaluation. Do not favor certain names of the assistants. Be as objective as possible. After providing your explanation, output your final verdict by strictly following this format: \"[[A]]\" if assistant A is better, \"[[B]]\" if assistant B is better, and \"[[C]]\" for a tie.

**User Prompt:**
[User Question]
{}

[The Start of Assistant A's Caption]
{}
[The End of Assistant A's Caption]

[The Start of Assistant B's Caption]
{}
[The End of Assistant B's Caption]

Figure 18: **Prompt templates used for GPT evaluations on caption qualities.**

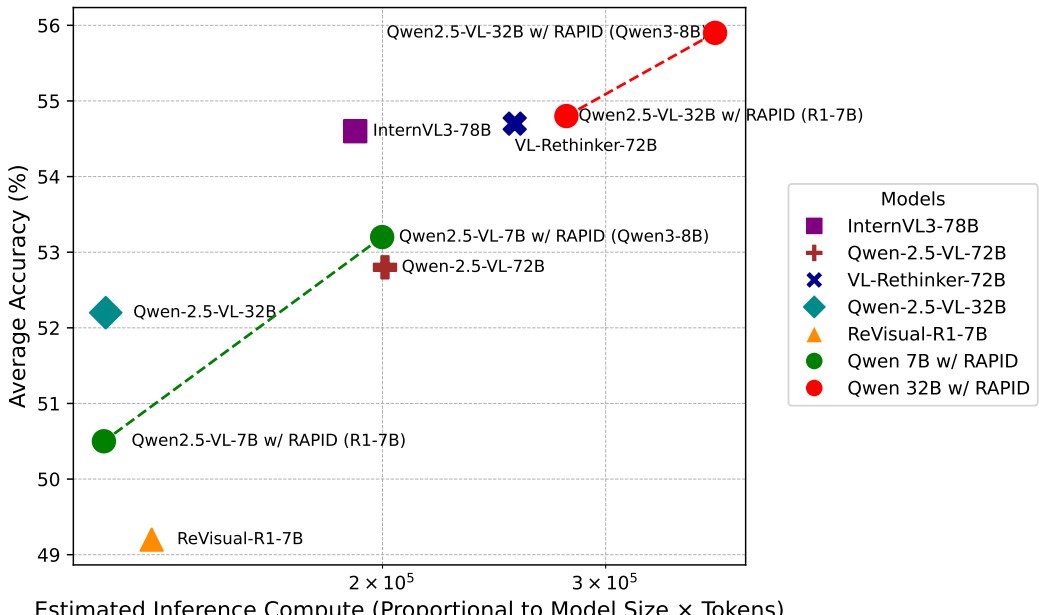

Figure 19: **Inference compute versus average accuracy.**

### E.2 ANALYSIS ON THE INFERENCE COMPUTE EFFICIENCY OF RAPID

To evaluate the inference compute efficiency of RAPID, we estimate the computational cost across the seven evaluation datasets from Table 1. For each dataset, we perform inference on a sample of 100 examples and approximate the compute as **model size × number of generated tokens**. For our staged RAPID approach, the total compute is the sum from the perception and reasoning stages, with the compute for each stage calculated using its respective model size. We benchmark these results against top-performing MLLMs from Table 1. Figure 19 plots the resulting average accuracy against inference compute for both RAPID-enhanced models and their baselines.

The analysis reveals two key findings:

- **RAPID achieves a favorable trade-off between accuracy and inference compute.** This efficiency is highlighted by specific configurations that demonstrate Pareto-optimality. For instance, Qwen2.5-VL-7B w/ RAPID (R1-7B) provides a better accuracy-to-compute ratio than ReVisual-R1-7B. In another example, Qwen2.5-VL-7B w/ RAPID (Qwen3-8B) outperforms the much larger Qwen2.5-VL-72B while being more computationally efficient.

- **RAPID demonstrates strong scalability with inference compute.** The architecture is designed such that allocating more computational resources at inference—for example, by swapping in a more powerful LLM reasoner—consistently yields higher accuracy.

### E.3 ANALYSIS ON THE HASCAP(·) CHECK AND OTHER VARIANTS

**Rationale for the use of hasCap(·).** Our initial goal was to discourage the captioning MLLM from simply outputting a final solution instead of a descriptive caption. A key empirical finding, however, was that strictly penalizing any text containing solution-like elements was suboptimal. For many visual reasoning tasks, a descriptive caption naturally and concisely includes the answer. For instance, if asked for the time on a clock face, a good caption might be "The image shows a clock with the hands pointing to 3:15," which contains both description and solution.

To test this, we ran an ablation study using a stricter checker that penalized the model whenever any part of the solution was detected in the output. As shown in Table 9, this variant hasSol(·) performed even worse in the decoupled pipeline than no check ("None"), confirming our hypothesis that forcing

Table 9: Comparisons of different checking strategies during VPO (Qwen2.5-VL-7B).

| (Penalty) Check | MathVista | MathVision | MathVerse | MMMU | WeMath | DynaMath | LogicVista | AVG |
|---|---|---|---|---|---|---|---|---|
| None | 76.0 | 41.5 | 50.6 | 62.9 | 43.1 | 33.1 | **57.7** | 52.1 |
| hasCap($\cdot$) | **76.1** | **43.7** | **52.2** | **64.7** | **45.4** | 32.7 | **57.7** | **53.2** |
| hasSol($\cdot$) | 75.8 | 40.8 | 48.7 | 61.3 | 42.4 | **33.0** | 55.5 | 51.1 |

a strict separation can harm performance by preventing the model from generating natural and direct descriptions.

Therefore, hasCap($\cdot$) was designed as a balanced heuristic: it ensures that a caption is present but does not have to strictly forbid the co-existence of a solution.

**Quantitative audit of the hasCap($\cdot$) heuristic.** We manually audited a random sample of 400 generations from our MLLM trained with VPO. We classified each generation into two ground-truth categories:

- **Positive Class**: The generation is a valid caption, which may or may not contain solution elements. (381 samples)
- **Negative Class**: The generation could be simply a solution or an invalid caption, such as a pure solution disguised with minimal boilerplate text (*e.g.*, "Here is a description... [solution]"). (19 samples)

The hasCap($\cdot$) prompt-based check produced the following results on this set:

- **True Positives (TP)**: 379 (Correctly identified a valid caption)
- **False Negatives (FN)**: 2 (Incorrectly flagged a valid caption as invalid)
- **True Negatives (TN)**: 14 (Correctly identified a gamed/invalid caption or a solution)
- **False Positives (FP)**: 5 (Incorrectly identified a gamed/invalid caption or a solution as valid)

From these numbers, we can derive the following metrics for the hasCap($\cdot$) detector:

- **False Negative Rate**: $2/(379 + 2) = 0.525\%$
- **False Positive Rate**: $5/(5 + 14) = 26.32\%$

This audit demonstrates that the hasCap($\cdot$) check is highly accurate and reliable.

**Analysis of Failure Modes** The failure cases mostly consist of false positives (or the invalid caption ignored by the 'hasCap($\cdot$)' check). We found they exhibit a similar pattern (hiding pure solution in a caption-like text) as shown below:

> *# Example of an invalid output (incorrectly identified as positive by our check):*
>
> Ok, here is a description of the image regarding the query. To find the circumference... [detailed mathematical derivation] ... Therefore, the circumference of the circle is 25.12 cm. ... This description aligns with the mathematical calculation...

Though such cases occur, they are too infrequent (5 out of 400 total samples) to impact the downstream performance.

### E.4 VPO HURTS REASONING ABILITY OF THE MLLM

We note a performance decrease in the MLLM's reasoning ability after VPO training (Table 2, rows ⒣ vs. ⒤) and we have investigated it further.

Our analysis reveals two key findings:

- The decrease in reasoning is not permanent and can be fully recovered with a simple, subsequent fine-tuning step.

Table 10: Comparisons of reasoning ability of Qwen2.5-VL-3B/7B at different stages.

| MLLM | Math Vista | Math Vision | Math Verse | MMMU | We Math | Dyna Math | Logic Vista | AVG |
|---|---|---|---|---|---|---|---|---|
| 3B (GRPO) | **69.1** | **26.9** | 38.2 | **56.9** | **34.0** | 20.0 | 42.5 | 41.1 |
| 3B (GRPO+VPO) | 68.8 | **26.9** | **39.8** | 49.4 | 33.0 | **21.6** | **46.3** | 40.8 |
| 3B (GRPO+VPO+GPRO) | **69.1** | **26.9** | **39.8** | 55.1 | 33.3 | 20.5 | 44.0 | **41.2** |
| 7B (GRPO) | 74.2 | 29.7 | **44.8** | 55.9 | **41.0** | 27.7 | **48.1** | 45.9 |
| 7B (GRPO+VPO) | **75.0** | **29.8** | 42.0 | 55.8 | 40.8 | 23.0 | 46.3 | 44.7 |
| 7B (GRPO+VPO+GPRO) | 74.5 | **29.8** | 44.3 | **55.9** | 40.7 | **28.5** | **48.1** | 46.0 |

Table 11: Decoupling results of Qwen2.5-VL-7B at different stages.

| MLLM | Math Vista | Math Vision | Math Verse | MMMU | We Math | Dyna Math | Logic Vista | AVG |
|---|---|---|---|---|---|---|---|---|
| 7B (GRPO+VPO) + Qwen3-8B | 76.1 | **43.7** | 52.2 | **64.7** | **45.4** | 32.7 | **57.7** | **53.2** |
| 7B (GRPO+VPO+GPRO)+ Qwen3-8B | **76.5** | 43.6 | **52.4** | 63.9 | 44.8 | **33.3** | 57.3 | 53.1 |

- The impact of this temporary decrease on the final, decoupled system's performance is primarily significant for smaller models, while larger models are more robust to this effect.

Below, we elaborate on these two points.

**Recovering reasoning performance with additional GRPO.** To counteract this, we performed a brief, additional 100-step GRPO training stage after the VPO stage. As demonstrated in Table 10, this additional GRPO stage successfully restores the reasoning performance for both the 3B and 7B MLLMs, bringing them back to the levels seen before VPO was applied.

**The impact on the decoupled framework is model-scale dependent.** Interestingly, we discovered that the necessity of this recovery step depends on the scale of the MLLM backbone.

- **For the 3B Model**: We observed that the drop in reasoning after VPO did negatively impact the performance of the full decoupled pipeline. Our hypothesis is that for a smaller model, this degradation leads to lower-quality "tentative solutions" being passed to the reasoner, thereby creating a bottleneck. For this reason, we had already incorporated this additional GRPO stage for the 3B model in our original paper, as illustrated in Figure 6. This step was crucial for achieving the strong performance gains reported for the 3B model.

- **For the 7B Model**: Although applying the extra GRPO stage to the 7B model restored its own reasoning ability (Table 10), it yielded no significant improvement for the final decoupled system (Table 11). Therefore, to maintain the methodological simplicity of RAPID, we omitted this non-essential step for the 7B model in our paper.

E.5 ANALYSIS ON THE EFFECT OF CAPTION LENGTH

In our original analysis (Figure 9 in Section 4.2), the final performance correlates with both the choice of LLM for reward computation and the length of the generated captions. This raises the question of whether caption length is the primary causal factor for the performance difference.

To isolate the effect of caption length, we conducted a controlled experiment. Our goal was to decouple the reward model's identity from the resulting caption length. We started with the setup that uses, Qwen3-8B, the stronger reasoner, to compute the VPO reward, which normally results in shorter captions (average 153 tokens) and worse performance. We then introduced an explicit length-controlled reward to encourage the perception model (Qwen2.5-VL-7B) to generate longer captions, matching the average length produced when using R1-7B for rewards (approx. 654 tokens).

To achieve this, we added a length penalty term to the reward function, as formulated in Aggarwal & Welleck (2025): $r_{len}(y, n_{\text{target}}) = -\alpha|n_{\text{target}} - n_y|$. Here, $n_{\text{target}}$ was set to 650, $n_y$ is the token count of the generated caption $y$, and the weight $\alpha$ was set to 0.0003 as per Aggarwal & Welleck (2025).

This approach successfully controlled the output length; the average caption length for the model trained with Qwen3-8B rewards increased from around 153 to around 627 tokens, as intended. We then evaluated this MLLM on our benchmark suite, with the results presented in Table 12. The model

Table 12: Correlation between the choices of LLM, the length of the captions and final performance.

| LLM | Length | MathVista | MathVision | MathVerse | MMMU | WeMath | DynaMath | LogicVista | AVG |
|---|---|---|---|---|---|---|---|---|---|
| R1-7B | 653.7 | **76.1** | **43.7** | **52.2** | **64.7** | **45.4** | 32.7 | **57.7** | **53.2** |
| Qwen3-8B | 153.1 | 75.8 | 40.8 | 48.7 | 61.3 | 42.4 | **33.0** | 55.5 | 51.1 |
| Qwen3-8B (lengh-controlled) | 627.1 | 75.8 | 40.8 | 48.7 | 61.3 | 42.4 | **33.0** | 55.5 | 51.1 |

Table 13: Adapting the LLM to reason over captions via GPRO training.

| Models | MathVista | MathVision | MathVerse | MMMU | WeMath | DynaMath | LogicVista | AVG |
|---|---|---|---|---|---|---|---|---|
| RAPID | 76.1 | **43.7** | 52.2 | **64.7** | **45.4** | 32.7 | **57.7** | 53.2 |
| RAPID (LLM trained) | **77.1** | 43.4 | **53.2** | 63.3 | 45.1 | **33.3** | 57.5 | **53.3** |

trained with length-controlled rewards showed a minor performance improvement over the baseline model trained with standard Qwen3-8B rewards. However, its performance still significantly lags behind the model trained with R1-7B as the reward source.

This outcome leads to a clear conclusion. Forcing the MLLM to generate longer text does not guarantee more comprehensive or useful descriptions. Instead, the model may produce verbose but less informative content to satisfy the length constraint. This result allows us to eliminate caption length as a confounding variable, confirming that the performance gap is attributable to the reasoning ability of the LLM that generates the reward signal.

### E.6 ADAPTING THE LLM TO REASON OVER CAPTIONS VIA FINE-TUNING

We conducted an experiment where we fine-tuned the LLM reasoner (Qwen3-8B) in a separate stage after VPO on the ViRL-39K dataset. The training data for the LLM consisted of the captions generated by our VPO-trained MLLM (Qwen2.5-VL-7B). We then applied the same GRPO objective (with a group-size of 4) to optimize the reasoner.

However, the experiment did not yield significant improvements. During training, we observed that the reward signal was highly unstable, fluctuating without a consistent upward trend. The final evaluation results, as shown in Table 13, confirmed this observation, revealing only marginal gains.

Our hypothesis is that a powerful, pre-trained LLM like Qwen3-8B already possesses robust reasoning capabilities that generalize effectively to understanding captions. Consequently, further fine-tuning provides diminishing returns, especially when the base model's reasoning is already strong.

### E.7 USING THE SAME MLLM FOR REASONING

We conducted a new experiment using the same trained MLLM, Qwen2.5-VL-7B (GRPO+VPO), for both the perception (captioning) and reasoning stages of our decoupled pipeline. Note that for this case, it is not actually a "decoupling" result as the image is visible to the reasoner. In Table 14, we compare this "decouple with self" approach against the standard end-to-end usage of the same MLLM, where it processes the image and question simultaneously.

As the table shows, applying our decoupled pipeline even with the same model for both stages yields a tangible performance improvement (46.1% vs. 44.7% average). This demonstrates that the structured two-stage process of first externalizing perception into text and then performing reasoning is beneficial in itself. However, it still lags behind when using Qwen3-8B, the default setting of our main paper, as the reasoner, which could be possibly attributed to their gap in reasoning capacity.

## F MORE EXPERIMENTAL DETAILS

### F.1 EVALUATION ON GENERAL BENCHMARKS

We select MME (Fu et al., 2024a), MMBench-v1.1 (Liu et al., 2024a), MM-Vet (Yu et al., 2023), SEED-Image[9] (Li et al., 2023), MMstar (Chen et al., 2024), RealWorld-QA (RW-QA) (xAI, 2024),

---

[9]We evaluated on the "Image" split.

Table 14: Performance comparison between decoupling (the same MLLM performing both captioning and reasoning) and an end-to-end MLLM. Qwen2.5-VL-7B (GRPO+VPO) is adopted.

| LLM | MathVista | MathVision | MathVerse | MMMU | WeMath | DynaMath | LogicVista | AVG |
|---|---|---|---|---|---|---|---|---|
| Decouple | 73.7 | **30.0** | **44.0** | **57.3** | **41.0** | **27.3** | **49.2** | **46.1** |
| End-to-end | **75.0** | 29.8 | 42.0 | 55.8 | 40.8 | 23.0 | 46.3 | 44.7 |

Table 15: **Ablation study of different components of RAPID** (with Qwen2.5-VL-3B). VPO†: VPO without the caption penalty; ‡: using `cap+sol` for reasoning-perception decoupling. *: After VPO, we additionally conduct GRPO to recover its reasoning ability.

| | Decouple | GRPO | VPO† | Cap. penalty | Math Vista | Math Vision | Math Verse | MMMU | We Math | Dyna Math | Logic Vista | AVG |
|---|---|---|---|---|---|---|---|---|---|---|---|---|
| Ⓐ | | | | | 64.5 | 21.9 | 28.8 | 50.1 | 24.2 | 13.4 | 39.6 | 34.6 |
| Ⓑ | ✓ | | | | 65.5 | 39.1 | 31.9 | 59.0 | 31.1 | 24.8 | 48.3 | 42.8 |
| Ⓒ | ✓ | ✓ | | | 68.5 | 40.0 | 39.2 | **61.0** | 35.1 | 26.9 | 51.7 | 46.1 |
| Ⓓ | ✓ | ✓ | ✓ | | 68.5 | 39.4 | 43.4 | 59.9 | 37.4 | 27.9 | 55.3 | 47.4 |
| Ⓔ | ✓ | ✓ | ✓ | ✓ | 69.0 | 39.7 | 44.3 | 60.9 | 38.6 | 27.3 | 55.3 | 47.9 |
| Ⓕ | ✓ | ✓* | ✓ | ✓ | **69.6** | 40.8 | **48.6** | 60.9 | **39.1** | **29.3** | 56.4 | **49.2** |
| Ⓖ | ✓‡ | ✓ | ✓ | ✓ | 67.0 | 40.9 | 44.3 | 58.0 | 33.2 | 28.9 | 54.4 | 46.7 |
| Ⓗ | ✓ | | ✓ | ✓ | 68.8 | **41.0** | 43.8 | 59.8 | 34.8 | 28.7 | 54.4 | 47.3 |
| Ⓘ | | ✓ | | | 69.1 | 26.9 | 38.2 | 56.9 | 34.0 | 20.0 | 42.5 | 41.1 |
| Ⓙ | | ✓ | ✓ | ✓ | 68.8 | 26.9 | 39.8 | 49.4 | 33.0 | 21.6 | 46.3 | 40.8 |

MMT-Bench (MMT) (Ying et al., 2024), and BLINK (Fu et al., 2024b) to assess foundational vision-language capabilities, which cover tasks such as object recognition, text recognition (OCR), spatial awareness and so on. Figure 20 presents the results for Qwen2.5-VL-3B. Similar to the observations for the 7B model (Figure 5), the VPO/GRPO-optimized model performs comparably to the original MLLM (Note we do not report results with Qwen2.5-VL-32B/72B as they show the same observations). This confirms that RAPID preserves general-purpose abilities across different model scales.

## F.2 ABLATION STUDY ON QWEN2.5-VL-3B

In this section, we extend our ablation study to the smaller Qwen2.5-VL-3B model, with results presented in Table 15. While the results are largely consistent with those from its 7B counterpart (Table 2), a critical difference emerges. The 3B model necessitates an additional GRPO stage (Ⓕ) following VPO to restore its reasoning capabilities[10], which in turn yields considerable accuracy gains. We attribute this requirement to the limited capacity of the 3B model, where optimizing for the VPO task appears to degrade its inherent reasoning performance—a trade-off that is less pronounced in the larger 7B model.

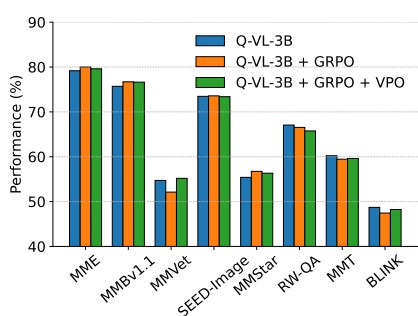

Figure 20: **General benchmark Results.** (Qwen-2.5-VL-3B)

## F.3 DETAILS FOR PAIRWISE COMPARISONS

**Extended comparisons with OmniCaptioner-7B and MM-Eureka-7B.** Following the setting in Sec. 4.4, we conducted a head-to-head comparison of our model, Qwen2.5-VL-7B (GRPO+VPO), against two strong baselines: OmniCaptioner-7B (an MLLM enhanced for standard captioning) and MM-Eureka-7B (an MLLM specially optimized for reasoning)

The win/tie/lose rates for Qwen2.5-VL-7B (GRPO+VPO) are reported in Figure 21 where we observe the following:

---

[10]We hypothesize that VPO degrades the quality of the intermediate reasoning steps passed to the LLM, an effect not always visible in the final accuracy.

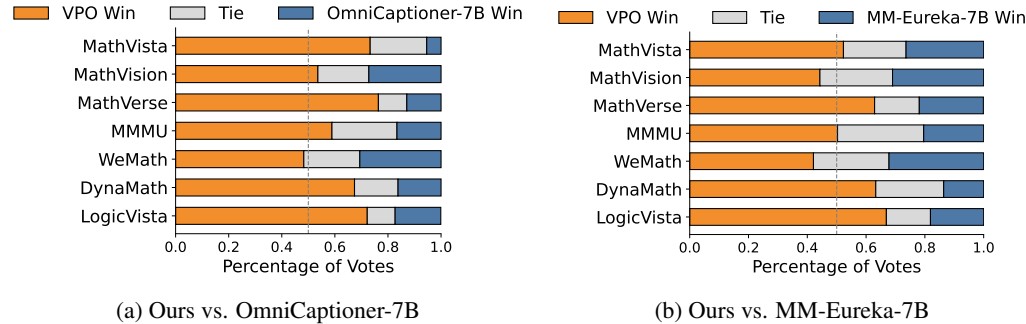

(a) Ours vs. OmniCaptioner-7B         (b) Ours vs. MM-Eureka-7B

Figure 21: Pairwise comparisons on caption quality among ours and OmniCaptioner-7B/MM-Eureka.

- **RAPID vs. OmniCaptioner-7B**: Our model's advantage stems from its focus on generating query-relevant captions, in contrast to OmniCaptioner's standard captions. Our model also benefits from a more advanced base model (Qwen2.5-VL-7B vs. Qwen2-VL-7B).

- **RAPID vs. MM-Eureka-7B**: Our model performs better because VPO directly optimizes for captioning quality, whereas MM-Eureka-7B is optimized for end-to-end reasoning.

This validates that VPO significantly enhances the MLLM's ability to generate high-quality, task-relevant descriptions, outperforming MLLMs specialized for either standard captioning or reasoning.

**Prompt for GPT evaluations.** We provide the prompt for GPT evaluations on the quality of the caption in Figure 18. Consistent with the details in Sec. 4.4, this instructs the GPT to (1) choose captions that include more comprehensive and accurate details required to answer the question and (2) exclude any solving process in the captions.

**Human evaluations.** We present the details of the human evaluation conducted for the pairwise comparison experiment. For this, 100 questions are randomly sampled from the testmini set of MathVista, and captions are generated using Qwen2.5-VL-3B, trained with and without VPO. A total of 4 trained human annotators are recruited, with each annotator comparing all the captions pairs to determine a winner or a tie. For each caption pair, we aggregate the results from different annotators by taking the majority of the decisions. Specifically, there are 4 annotators and only 3 decisions (win, tie and lose), so there is at least one decision that occurs twice. We compute the **inter-annotator consistency** following Zheng et al. (2023) by calculating the ratio of identical decision pairs out of all possible decision pairs and average them across all samples.

In Table 16, we report the win/tie/lose ratio (*i.e.*, "win" means captions generated by MLLMs with VPO is better) and an additional measure of **GPT-human consistency**, calculated by the agreement rate between GPT-4o and human judgments. As can be seen, Qwen2.5-VL-3B trained with VPO demonstrates superior caption quality under human evaluation. This aligns with the result in Figure 11, which is further supported by the high consistency of 87%. This supports the rationale of using GPT-4o as a judge for evaluating caption quality.

Table 16: **Human evaluation on pairwise comparison of the caption quality**.

| Win | Tie | Lose | GPT-human consistency | Inter-annotator consistency |
|-----|-----|------|-----------------------|-----------------------------|
| 62% | 32% | 6%   | 87%                   | 85%                         |

## F.4 EXTENDING DECOUPLING TO OTHER MLLMS.

We apply the decoupling pipeline alone to more MLLMs (*i.e.*, InternVL3-8B, VL-Rethinker-7B and MM-Eureka-7B) with different LLMs (*i.e.*, Qwen3-8B and GPT-OSS-120B) and report the results in Table 17.

Table 17: **Decoupling (using Qwen3-8B and GPT-OSS-120B) performance of different MLLMs.** The best results are **bold**.

| Model | MathVista | MathVision | MathVerse | MMMU | WeMath | DynaMath | LogicVista | AVG |
|---|---|---|---|---|---|---|---|---|
| InternVL3-8B | **73.6** | 29.3 | 39.8 | 62.7 | 37.1 | 25.5 | 44.1 | 44.6 |
| w/ Qwen3-8B | 71.3 | 42.4 | 39.3 | 64.4 | 38.8 | **29.1** | 50.1 | 47.9 |
| w/ GPT-OSS-120B | 70.6 | **47.1** | **41.2** | **68.1** | **41.0** | **29.1** | **51.0** | **49.7** |
| VL-Rethinker-7B | **74.9** | 30.0 | 47.5 | 56.9 | 37.3 | 21.4 | 43.6 | 44.5 |
| w/ Qwen3-8B | 72.8 | 43.0 | **51.9** | 59.7 | 41.1 | **30.9** | 52.3 | 50.2 |
| w/ GPT-OSS-120B | 72.8 | **47.8** | 50.0 | **68.1** | **46.3** | 30.7 | **55.0** | **53.0** |
| MM-Eureka-7B | **73.0** | 27.9 | 46.1 | 54.9 | 34.7 | 22.6 | 48.3 | 43.9 |
| w/ Qwen3-8B | 72.2 | 42.1 | 47.7 | 61.4 | 35.9 | 28.9 | **51.2** | 48.5 |
| w/ GPT-OSS-120B | 70.5 | **47.5** | **51.8** | **68.2** | **43.9** | **33.5** | 50.8 | **52.3** |

## F.5 TRAINING DYNAMICS OF VPO

We show the average reward and caption lengths over training in Figures 22 and 23. We observe that:

- **Rewards increased as training progresses.** This confirms the effectiveness of VPO as it allows the MLLM to generate captions that lead to higher reasoning accuracy.
- **Caption lengths grow as training progresses.** An explanation for this phenomenon is that the MLLM learns to generate more comprehensive captions during training, which is reflected by longer lengths. This is also confirmed in the Appendix I, where we visualize the captions.

## F.6 TRAINING DYNAMICS OF GRPO

We visualize the training dynamics of GRPO in Figure 24. For 3B and 7B MLLMs, the rewards fluctuate and eventually drop after an initial convergence. In contrast, the 72B model exhibits a more stable convergence. In both cases, subsequent VPO stage continues to improve the performance under the perception-reasoning pipelines.

## G MORE DISCUSSIONS

### G.1 ADVANTAGES OF RAPID OVER UNIFIED ARCHITECTURE

We are aware that RAPID is a modular framework rather than a unified architecture that is adopted by most existing MLLMs. Our perspective is that RAPID are not to replace these unified systems, but rather to serve a dual, complementary role. Specifically, it could serve as a data engine to build future powerful unified models. However, under limited budget, it could be a pragmatic and economic solution to address capacity gap of unified models.

**RAPID as a Data Engine for Future Unified Models.** We agree that a powerful, unified MLLM is the ultimate goal. However, training such models is hampered by the scarcity of high-quality, multi-modal reasoning data. Our modular framework directly addresses this bottleneck by serving as a data engine (Gou et al., 2024; 2025). The RAPID pipeline can generate vast amounts of complex reasoning trajectories. This high-quality, model-generated data can then be used to train and significantly improve a future unified MLLM (Chen et al., 2025a), a technique proven effective in prior work (Yang et al., 2025b; Huang et al., 2025).

**RAPID-like Methods Bridge Current Ability Gap.** At present, general-purpose MLLMs still lag behind specialized models in critical perception tasks like object counting (Tamarapalli et al., 2025), fine-grained OCR (Chen et al., 2025c), depth estimation (Fu et al., 2024b) and semantic segmentation (Anonymous, 2025). However, expert-agent-based systems could pragmatically bridge this gap by integrating these "expert" models (Zhou et al., 2025; Su et al., 2025; Liu et al., 2025b; Gou et al., 2023; Liu et al., 2024b). This allows the system to leverage SoTA performance on these sub-tasks immediately, achieving higher overall accuracy.

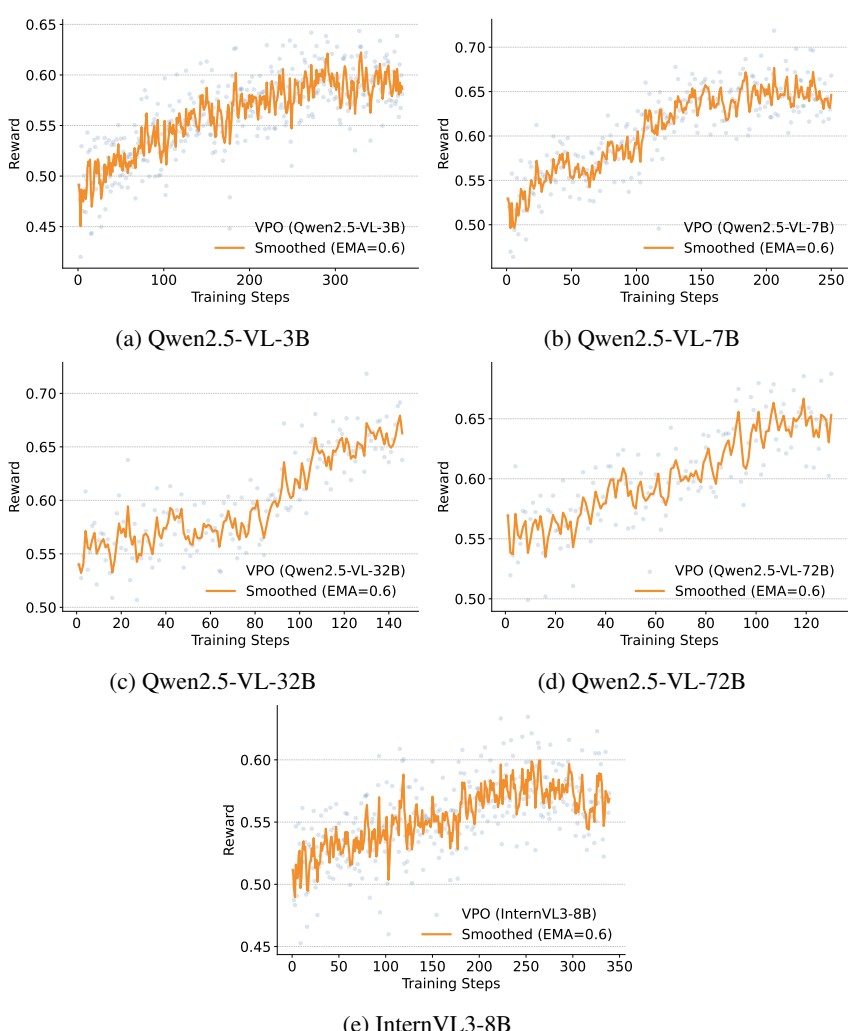

(a) Qwen2.5-VL-3B

(b) Qwen2.5-VL-7B

(c) Qwen2.5-VL-32B

(d) Qwen2.5-VL-72B

(e) InternVL3-8B

Figure 22: Rewards over training steps for VPO.

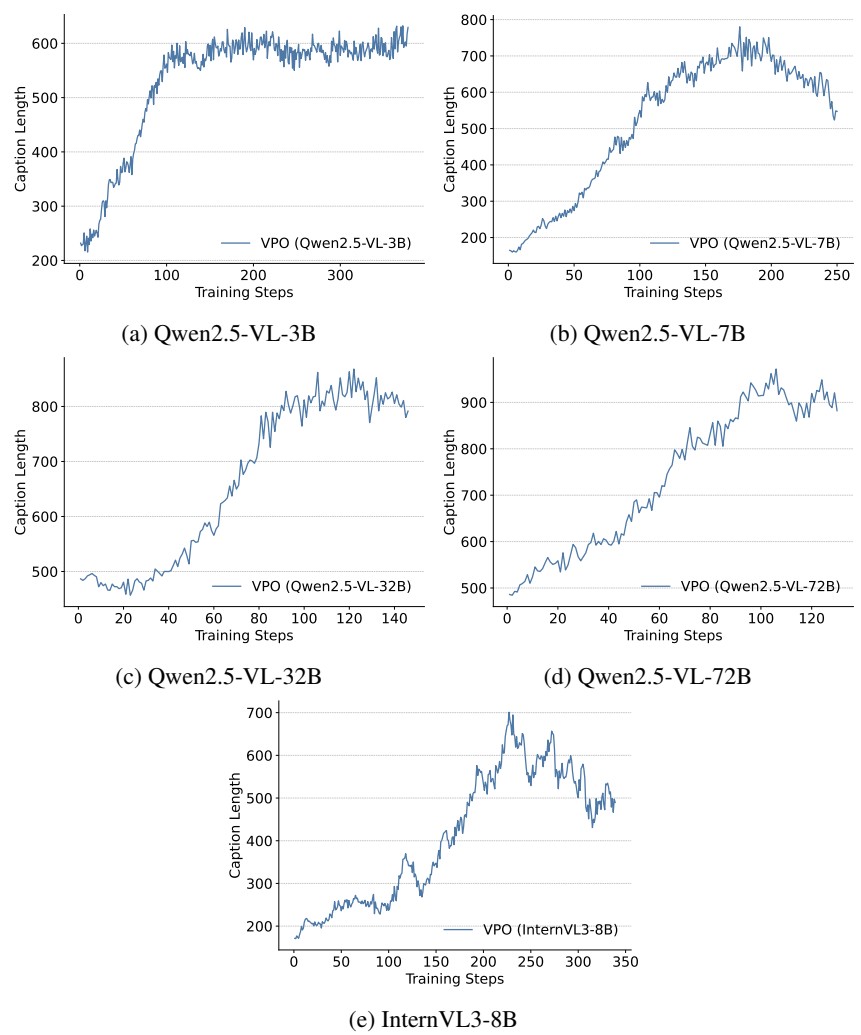

(a) Qwen2.5-VL-3B

(b) Qwen2.5-VL-7B

(c) Qwen2.5-VL-32B

(d) Qwen2.5-VL-72B

(e) InternVL3-8B

Figure 23: Caption length over training steps for VPO.

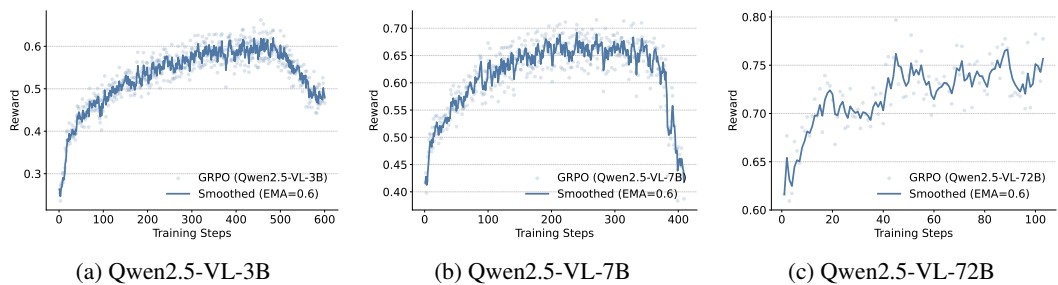

(a) Qwen2.5-VL-3B

(b) Qwen2.5-VL-7B

(c) Qwen2.5-VL-72B

Figure 24: Rewards over training steps for GRPO.

**RAPID as an Economic Solution with Limited Compute.** While a unified architecture is a compelling goal, RAPID is a more pragmatic solution under restricted training budgets. It circumvents the prohibitive cost of training a unified model on massive data. For example, RAPID could enjoy the advanced reasoning ability of new LLMs without training a new MLLM from scratch.

## H USE OF LARGE LANGUAGE MODELS (LLMS)

In this paper, the role of Large Language Models (LLMs) was confined to a minor, supporting capacity for polishing the writing. They were not involved in the core research process, such as ideation or analysis.

## I CASE STUDY

**Caption qualities.** We conduct a case study on the generated query-relevant captions. Specifically, for a multi-modal reasoning question and image, we investigate the quality of the generated captions. For MLLMs, we consider Qwen2.5-VL series (3B/32B) both with and without VPO. We visualize the question, image and captions in Tables 18- 24 for Qwen2.5-VL-3B and Tables 25- 29 for Qwen2.5-VL-32B.

Comparing the captions generated by MLLMs with and without VPO, we discover the following:

- **VPO leads to more visual details.** We highlight these visual details in red in the table. Notably, these details are important clues required to correctly solve the question. This shows that VPO is effective in improving the quality (especially comprehensiveness) of the query-relevant captions.

- **VPO leads to captions with more organized and hierarchical structures.** For example, in Table 23, the MLLM with VPO describes the images at three levels, *i.e.*, Tropic level, Terrestrial food chain and aquatic food chain. This allows the reasoner to quickly locate important information in the captions. However, the original MLLM uses a sequence of sentences that are less clear.

- **Large-sized MLLMs generate more comprehensive captions.** We found the captions generated by Qwen2.5-VL-32B are significantly longer than those generated by the 3B model. This is because larger MLLMs have better reasoning abilities that allow it to describe the image from multiple perspectives and in a more logically coherent way. This leads to longer captions.

**Reasoning accuracies.** In Table 30, we provide a complete comparison of captions (generated by Qwen2.5-VL-3B with and without VPO) and the resulting reasoning results (produced by R1-7B). Similarly to the case study on caption quality, MLLMs trained with VPO generate captions that capture more details. This is critical to the correctness of the reasoning process. As can be seen, the reasoner that receives captions with VPO arrives at the correct answer after several rounds of thinking and reflection. *However, the reasoner that accepts the under-optimized caption experiences multiple contradictions and confusion (highlighted in brown), which leads to responses that exceeds the maximum context length and finally fails this problem.*

Table 18: **Case Study with Qwen2.5-VL-3B.** Additional visual details that are relevant to the question but are only generated by MLLMs with VPO are highlighted in red.

**Image:**

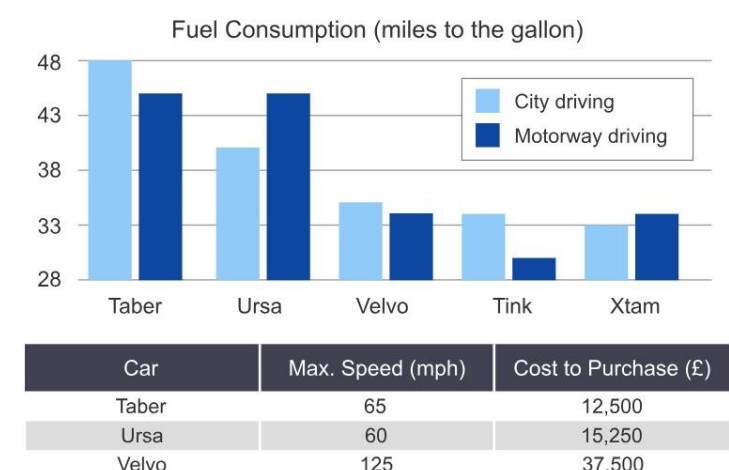

| Car | Max. Speed (mph) | Cost to Purchase (£) |
|---|---|---|
| Taber | 65 | 12,500 |
| Ursa | 60 | 15,250 |
| Velvo | 125 | 37,500 |
| Tink | 95 | 55,250 |
| Xtam | 110 | 62,500 |

**Question:**
If a driver travels an average of 4,250 miles per month driving only along motorways in an Xtam car, what is the predicted annual consumption of fuel (in gallons)? Select from A, B, C, D and E.
(A)Cannot say (B) 1,500 (C) 125 (D) 150 (E) 1,250

**Query-relevant Captions (Qwen2.5-VL-3B w/ VPO):**
The image is a bar graph titled "Fuel Consumption (miles to the gallon)" that compares the fuel efficiency of five different cars, labeled as Taber, Ursa, Velvo, Tink, and Xtam. The graph is divided into two categories: "City driving" and "Motorway driving."
For each car, there are two bars:
- The light blue bar represents fuel consumption for city driving.
- The dark blue bar represents fuel consumption for motorway driving.
The data for each car is as follows:
- **Taber:**
   City driving: 48 miles per gallon
   Motorway driving: 43 miles per gallon
- **Ursa:**
   City driving: 39 miles per gallon
   Motorway driving: 43 miles per gallon
- **Velvo:**
   City driving: 35 miles per gallon
   Motorway driving: 33 miles per gallon
- **Tink:**
   City driving: 33 miles per gallon
   Motorway driving: 28 miles per gallon
- **Xtam:**
   City driving: 33 miles per gallon
   Motorway driving: 33 miles per gallon
Below the bar graph, there is a table that provides additional information about each car:
- **Max. Speed (mph):** Taber (65), Ursa (60), Velvo (125), Tink (95), Xtam (110)
- **Cost to Purchase (£):** Taber (12,500), Ursa (15,250), Velvo (37,500), Tink (55,250), Xtam (62,500)
The image does not contain any information related to the question about fuel consumption for a specific car model.

**Query-relevant Captions (Qwen2.5-VL-3B):**
The image is a bar graph titled "Fuel Consumption (miles to the gallon)" that compares the fuel efficiency of five different cars, Taber, Ursa, Velvo, Tink, and Xtam, under two driving conditions: city driving and motorway driving. The graph uses two colors to differentiate between these conditions: light blue for city driving and dark blue for motorway driving.

Below the graph, there is a table that provides additional information about each car, including:

- **Max. Speed (mph)**: The maximum speed each car can achieve.
- **Cost to Purchase (£)**: The price of each car in pounds.

The table entries are as follows:
- **Taber**:
  Max. Speed: 65 mph
  Cost to Purchase: £12,500
- **Ursa**:
  Max. Speed: 60 mph
  Cost to Purchase: £15,250
- **Velvo**:
  Max. Speed: 125 mph
  Cost to Purchase: £37,500
- **Tink**:
  Max. Speed: 95 mph
  Cost to Purchase: £55,250
- **Xtam**:
  Max. Speed: 110 mph
  Cost to Purchase: £62,500

The graph shows that the Taber and Ursa cars have higher fuel efficiency in city driving compared to motorway driving, while the Velvo, Tink, and Xtam cars have higher fuel efficiency in motorway driving compared to city driving.

Table 19: **Case Study with Qwen2.5-VL-3B.** Additional visual details that are relevant to the question but are only generated by MLLMs with VPO are highlighted in red.

**Image:**

As shown in the figure, AB parallel CD, EG bisects angle BEF,
then angle 2 is equal to ()
Choices: A:50°   B:60°   C:65°   D:90°

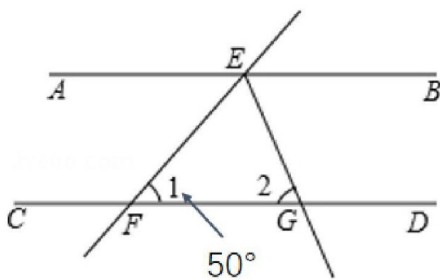

**Question:**
According to the question shown in the image, please directly answer the question and provide the correct option letter, e.g., A, B, C, D.

**Query-relevant Captions (Qwen2.5-VL-3B w/ VPO):**
The image is a geometric diagram with the following elements:
- Two parallel lines labeled $AB$ and $CD$.
- A line segment $EF$.
- A point $G$ where line segment $EG$ intersects line segment $CD$.
- An angle labeled $50°$ at point $F$.

- An angle labeled 2 at point $G$.
- A question asking about the measure of angle 2, with choices:
   A: 50°
   B: 60°
   C: 65°
   D: 90°
The diagram shows a triangle $\triangle EFG$ with angle 1 at point $F$ and angle 2 at point $G$.
The angle at point $F$ is given as 50°.
The question asks for the measure of angle 2, given that $AB \parallel CD$ and $EG$ bisects angle $\angle BEF$.

**Query-relevant Captions (Qwen2.5-VL-3B):**
The image shows a geometric figure with lines AB and CD parallel to each other. Line EG bisects angle BEF. Angle 1 is marked as 50 degrees.

Table 20: **Case Study with Qwen2.5-VL-3B.** Additional visual details that are relevant to the question but are only generated by MLLMs with VPO are highlighted in red.

**Image:**

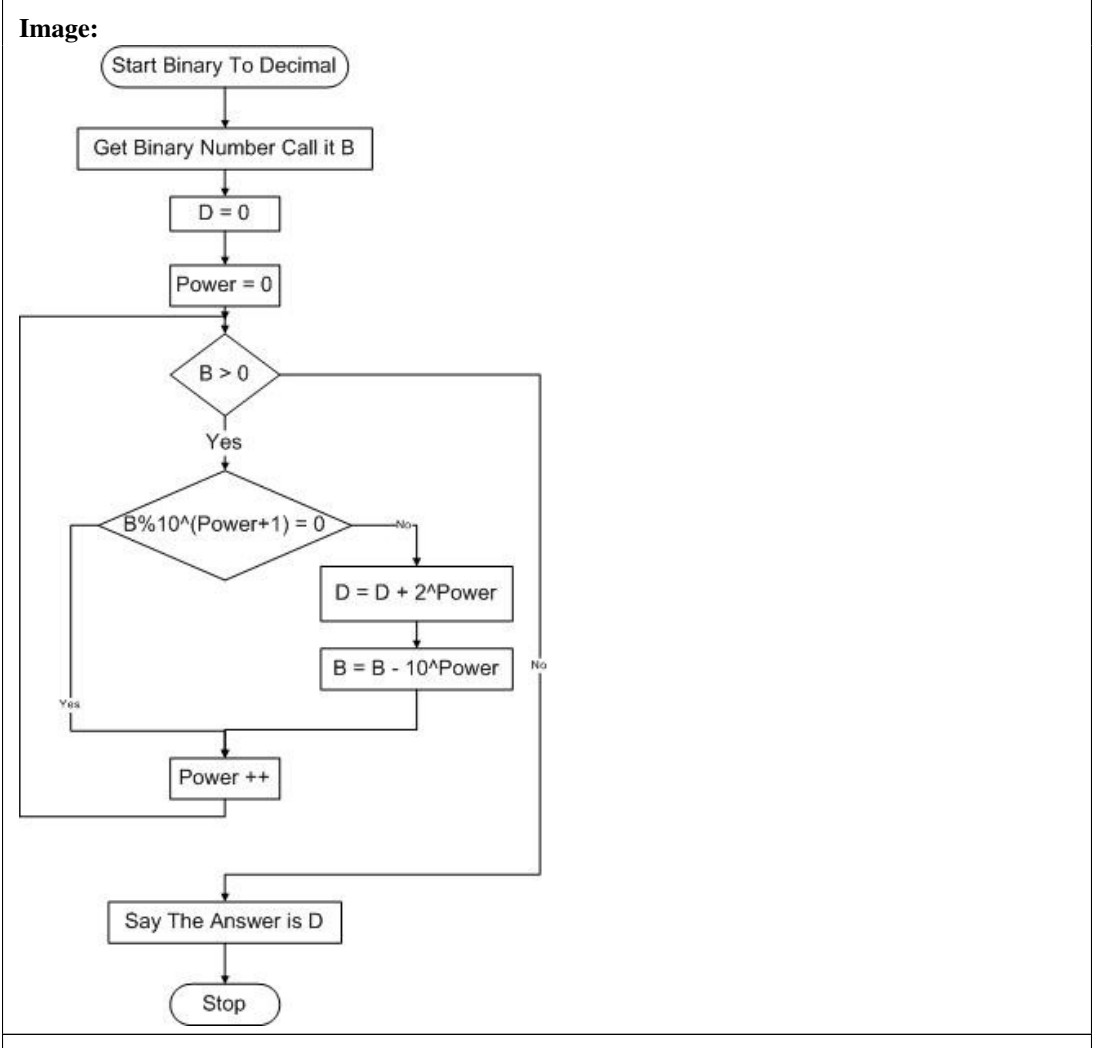

**Question:**
Input binary number B = 11000, find the corresponding decimal value by the algorithm shown in image.

**Query-relevant Captions (Qwen2.5-VL-3B w/ VPO):**

The image is a flowchart titled "Start Binary To Decimal." It outlines a process to convert a binary number to its decimal equivalent. The flowchart starts with the instruction "Get Binary Number Call it B," which is followed by initializing variables D and Power to 0. The flowchart then checks if B is greater than 0. If B is greater than 0, it checks if B modulo 10 raised to the power of (Power + 1) is equal to 0. If this condition is false, it updates D by adding 2 raised to the power of Power, and then updates B by subtracting 10 raised to the power of Power. If the condition is true, it increments Power. This process continues until B is no longer greater than 0. Finally, it states "Say The Answer is D" and "Stop."

The image does not contain any visual elements or additional information beyond the flowchart itself.

**Query-relevant Captions (Qwen2.5-VL-3B):**
Start Binary To Decimal
Get Binary Number Call it B
$D = 0$
Power $= 0$
$B > 0$
$B \% 10^{(}\text{Power} + 1) = 0$
$D = D + 2^{\text{Power}}$
$B = B - 10^{\text{Power}}$
Yes
Power **++**

Table 21: **Case Study with Qwen2.5-VL-3B.** Additional visual details that are relevant to the question but are only generated by MLLMs with VPO are highlighted in red.

**Image:**

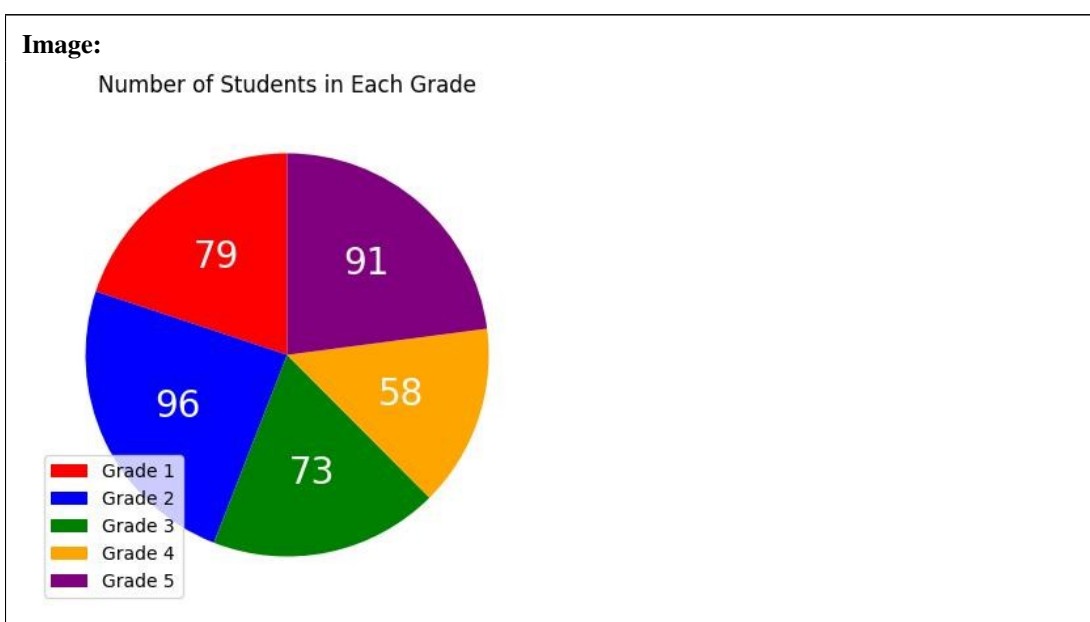

**Question:**
The pie chart below represents the number of each grade at a United States elementary school. What is the percentage of Grade 2? The answer should be a percentage between 0 and 100.

**Query-relevant Captions (Qwen2.5-VL-3B w/ VPO):**
The image is a pie chart titled "Number of Students in Each Grade." The chart is divided into five segments, each representing a different grade level and the number of students in that grade. The grades and their corresponding numbers are as follows:
- Grade 1: 79 students
- Grade 2: 96 students
- Grade 3: 73 students

- Grade 4: 58 students
- Grade 5: 91 students

The pie chart visually represents the distribution of students across these grades. The segments are color-coded for each grade, with Grade 1 in red, Grade 2 in blue, Grade 3 in green, Grade 4 in orange, and Grade 5 in purple. The chart provides a clear visual representation of the number of students in each grade at the elementary school.

**Query-relevant Captions (Qwen2.5-VL-3B):**
The pie chart below represents the number of each grade at a United States elementary school. What is the percentage of Grade 2? The answer should be a percentage between 0 and 100.

Table 22: **Case Study with Qwen2.5-VL-3B.** Additional visual details that are relevant to the question but are only generated by MLLMs with VPO are highlighted in red.

**Image:**

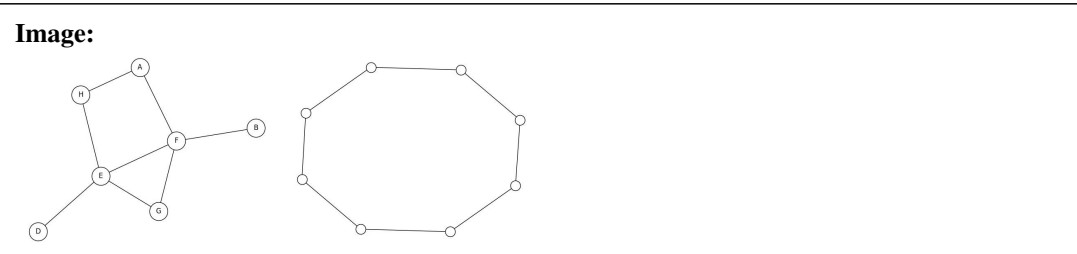

**Question:**
Is the left graph isomorphic with the right graph? choice: (A) Yes (B) No

**Query-relevant Captions (Qwen2.5-VL-3B w/ VPO):**
The image consists of two graphs. The left graph is a connected graph with the following vertices and edges: A, B, C, D, E, F, G, and H. The edges are as follows: A is connected to H, A is connected to F, H is connected to E, B is connected to F, F is connected to E, E is connected to G, F is connected to G, and E is connected to D.

The right graph is a simple cycle graph with 8 vertices, labeled as 1, 2, 3, 4, 5, 6, 7 and 8 and edges connecting each vertex to the next in a circular manner.

Based on the description, the left graph is not isomorphic to the right graph. The left graph has a more complex structure with multiple connections and a central node (E), while the right graph is a simple cycle with no such central node and a different number of vertices.

**Query-relevant Captions (Qwen2.5-VL-3B):**
The left graph is a graph with 8 vertices and 10 edges. The right graph is a graph with 8 vertices and 8 edges. The left graph is not isomorphic to the right graph.

Table 23: **Case Study with Qwen2.5-VL-3B.** Additional visual details that are relevant to the question but are only generated by MLLMs with VPO are highlighted in red.

**Image:**

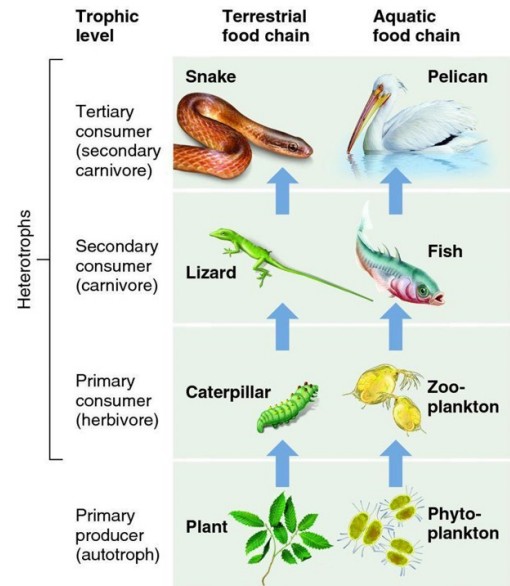

**Question:** Hint: Please answer the question and provide the correct option letter, e.g., A, B, C, D, at the end. Question: What happens to fish if pelicans increase?Choices: (A) decrease (B) nothing (C) increase (D) none of the above

**Query-relevant Captions (Qwen2.5-VL-3B w/ VPO):**
The image depicts a food chain diagram showing the trophic levels and different types of consumers in both terrestrial and aquatic environments. Here is a detailed description of the image:
1. **Trophic Level**:
- **Primary producer (autotroph)**: This level includes plants and phytoplankton, which are the base of the food chain.
- **Primary consumer (herbivore)**: This level includes caterpillars and zooplankton, which feed on the primary producers.
- **Secondary consumer (carnivore)**: This level includes lizards and fish, which feed on the primary consumers.
- **Tertiary consumer (secondary carnivore)**: This level includes snakes and pelicans, which feed on the secondary consumers.
2. **Terrestrial food chain**:
- **Primary producer (autotroph)**: Plant
- **Primary consumer (herbivore)**: Caterpillar
- **Secondary consumer (carnivore)**: Lizard
- **Tertiary consumer (secondary carnivore)**: Snake
3. **Aquatic food chain**:
- **Primary producer (autotroph)**: Phytoplankton
- **Primary consumer (herbivore)**: Zooplankton
- **Secondary consumer (carnivore)**: Fish
- **Tertiary consumer (secondary carnivore)**: Pelican
The image also shows arrows indicating the direction of energy flow from one trophic level to the next, emphasizing the flow of energy through different ecosystems.

**Query-relevant Captions (Qwen2.5-VL-3B):** The image shows a food chain diagram with two types of food chains: terrestrial and aquatic. The terrestrial food chain includes a caterpillar, lizard, fish, and snake, with the caterpillar being the primary producer and the snake being the tertiary consumer. The aquatic food chain includes zooplankton, phytoplankton, fish, and a pelican, with the zooplankton being the primary producer and the pelican being the tertiary consumer.

Table 24: **Case Study with Qwen2.5-VL-3B.** Additional visual details that are relevant to the question but are only generated by MLLMs with VPO are highlighted in red.

---

**Image:**

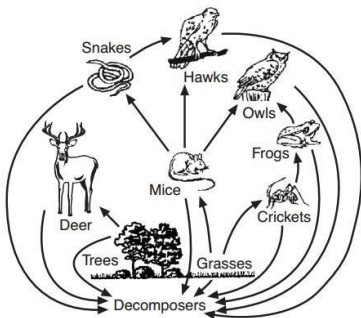

---

**Question:**
Hint: Please answer the question and provide the correct option letter, e.g., A, B, C, D, at the end.
Question: If frogs were removed from this environment what animal would potentially see an increase in its population? Choices:(A) crickets (B) deer (C) snakes (D) hawks

---

**Query-relevant Captions (Qwen2.5-VL-3B w/ VPO):**
The image is a food web diagram depicting various organisms and their interactions within an ecosystem. The diagram includes the following elements:

- **Deer** (at the bottom left) consuming **Trees**.
- **Mice** (in the center) consuming **Grasses**.
- **Crickets** (in the center right) consuming **Grasses**.
- **Snakes** (at the top left) consuming **Mice**.
- **Hawks** (at the top center) consuming **Snakes**.
- **Owls** (at the top right) consuming **Mice**.
- **Frogs** (in the center right) consuming **Crickets**.
- **Decomposers** (at the bottom center) breaking down organic matter.

The arrows indicate the direction of energy flow from one organism to another, showing the relationships between the different species in the ecosystem.

---

**Query-relevant Captions (Qwen2.5-VL-3B):** The image is a food web diagram depicting various organisms and their interactions within an ecosystem. It includes a deer, trees, grasses, mice, crickets, frogs, hawks, and owls. The arrows indicate the direction of energy flow from producers (trees and grasses) to consumers (deer, mice, crickets, frogs, hawks, and owls), and decomposers (decomposers). The diagram shows how energy moves through the ecosystem, with each organism feeding on others, and how the removal of frogs could potentially lead to an increase in the population of crickets, as they are a primary food source for frogs.

---

Table 25: **Case Study with Qwen2.5-VL-32B.** Additional visual details that are relevant to the question but are only generated by MLLMs with VPO are highlighted in red.

---

**Image:**

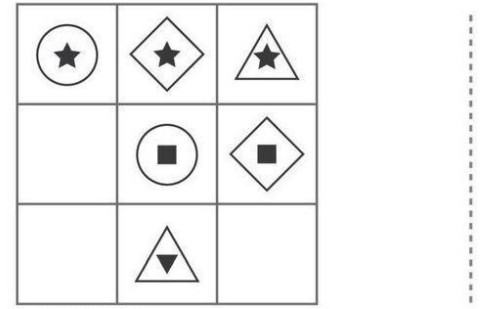 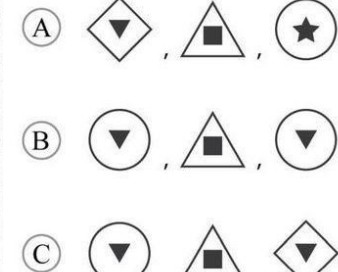

**Question:**
Hint: Please answer the question and provide the correct option letter, e.g., A, B, C, D, at the end.
Question: Which option is correct? Choices: (A) A (B) B (C) C

**Query-relevant Captions (Qwen2.5-VL-32B w/ VPO):**
The image presents a logic puzzle involving patterns and shapes. Here is a detailed description of the image:

**Left Side: Grid with Shapes**

- The grid is a 3x3 matrix.
- Each cell contains a shape with a specific pattern or design.
- The shapes include circles, diamonds, and triangles, each with different internal designs (e.g., stars, squares, etc.).

*Row 1:*

1. Cell 1: A circle with a star inside.
2. Cell 2: A diamond with a star inside.
3. Cell 3: A triangle with a star inside.

*Row 2:*

1. Cell 1: Empty.
2. Cell 2: A circle with a square inside.
3. Cell 3: A diamond with a square inside.

*Row 3:*

1. Cell 1: Empty.
2. Cell 2: A triangle with a downward arrow inside.
3. Cell 3: Empty.

**Right Side: Multiple Choice Options**
There are three options labeled A, B, and C. Each option shows a sequence of three shapes. The task is to determine the correct pattern that fits the logic of the grid.

*Option A:*

1. A diamond with a downward arrow inside.
2. A triangle with a square inside.
3. A circle with a star inside.

*Option B:*

1. A diamond with a downward arrow inside.
2. A triangle with a square inside.
3. A circle with a downward arrow inside.

*Option C:*

1. A diamond with a downward arrow inside.

2. A triangle with a square inside.

3. A diamond with a downward arrow inside.

**Analysis of the Grid Patterns**
*Column-wise Observation:*

- **Column 1:**
    - Row 1: Circle with a star.
    - Row 2: Empty.
    - Row 3: Empty.
- **Column 2:**
    - Row 1: Diamond with a star.
    - Row 2: Circle with a square.
    - Row 3: Triangle with a downward arrow.
- **Column 3:**
    - Row 1: Triangle with a star.
    - Row 2: Diamond with a square.
    - Row 3: Empty (to be determined).

*Row-wise Observation:*

- **Row 1:** Circle (star), Diamond (star), Triangle (star)
- **Row 2:** Empty, Circle (square), Diamond (square)
- **Row 3:** Empty, Triangle (downward arrow), ? (to be determined)

*Shape Progression:*

- The shapes in each row and column follow a specific order: Circle → Diamond → Triangle.
- The internal designs also follow a pattern: Star → Square → Downward Arrow.

---

**Query-relevant Captions (Qwen2.5-VL-32B):**
The image consists of two main sections:
**Left Section:**

- A 3x3 grid with various shapes and patterns.
- The grid contains:
    - A circle with a star inside in the top-left cell.
    - A diamond with a star inside in the top-middle cell.
    - A triangle with a star inside in the top-right cell.
    - A circle with a square inside in the middle-left cell.
    - A diamond with a square inside in the middle-middle cell.
    - A triangle with a square inside in the middle-right cell.
    - A triangle with a downward-pointing arrow inside in the bottom-left cell.
    - The bottom-middle and bottom-right cells are empty.

**Right Section:**

- Three options labeled A, B, and C.
- Each option contains three shapes:
  - **Option A:** A diamond with a downward-pointing arrow inside, a triangle with a square inside, and a circle with a star inside.
  - **Option B:** A circle with a downward-pointing arrow inside, a triangle with a square inside, and another circle with a downward-pointing arrow inside.
  - **Option C:** A circle with a downward-pointing arrow inside, a triangle with a square inside, and a diamond with a downward-pointing arrow inside.

The task appears to involve identifying a pattern or sequence based on the shapes and their internal symbols in the grid and selecting the correct option from the choices provided.

Table 26: **Case Study with Qwen2.5-VL-32B.** Additional visual details that are relevant to the question but are only generated by MLLMs with VPO are highlighted in red.

**Image:**

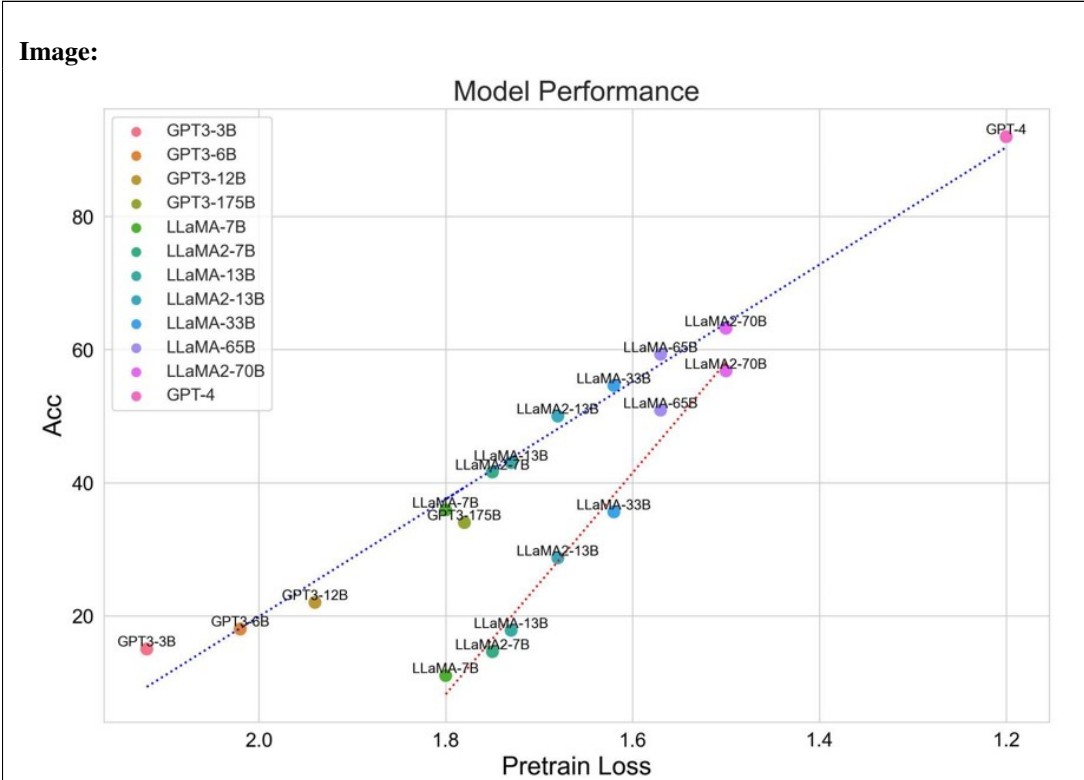

Figure 2: The performance of SFT (blue lines) and ICL (red lines) settings on GSM8K. GPT-4 states they use some part of the GSM8K data in pre-training, and suggest others consider its performance between SFT and ICL.

**Question:**
Hint: Please answer the question requiring an integer answer and provide the final value, e.g., 1, 2, 3, at the end. Question: How many models in the figure achieve an Acc score greater than 60?

**Query-relevant Captions (Qwen2.5-VL-32B w/ VPO):**

The image is a scatter plot titled **"Model Performance,"** which compares the performance of various large language models on a specific task, likely related to the GSM8K dataset. The plot uses different colors and line styles to distinguish between models and their training settings. Here is a detailed description of the image:

**Title and Labels:**

- **Title:** "Model Performance"
- **X-axis:** Labeled "Pretrain Loss," ranging from approximately 1.2 to 2.0.
- **Y-axis:** Labeled "ACC" (presumably short for Accuracy), ranging from 0 to 80.

**Legend:**
The legend on the left side lists various models and their corresponding colors and markers:

- GPT3-3B (pink), GPT3-6B (orange), GPT3-12B (yellow), GPT3-175B (dark green)
- LLaMA-7B (light green), LLaMA2-7B (teal), LLaMA-13B (dark teal), LLaMA2-13B (light blue)
- LLaMA-33B (blue), LLaMA-65B (purple), LLaMA2-70B (pink), GPT-4 (dark pink)

**Line Styles:**

- **Blue Lines:** Represent the SFT (Supervised Fine-Tuning) setting.
- **Red Lines:** Represent the ICL (In-Context Learning) setting.
- Dotted lines connect points for each model, showing trends in performance.

**Data Points:**
Each model is represented by a series of points on the plot, with the position determined by:

- **X-axis (Pretrain Loss):** Lower values indicate better pretraining.
- **Y-axis (ACC):** Higher values indicate better accuracy.

**Key Observations:**

1. **GPT-4 Performance:**
   - GPT-4 (dark pink) achieves the highest accuracy (ACC) score, well above 60, and has a low pretraining loss.
   - It is plotted as a single point at the top-right of the graph.
2. **LLaMA and LLaMA2 Models:**
   - Various LLaMA and LLaMA2 models (e.g., LLaMA-7B, LLaMA2-7B, LLaMA-13B, etc.) are scattered across the plot.
   - Larger models (e.g., LLaMA2-70B) tend to have higher accuracy scores.
3. **GPT3 Models:**
   - GPT3 models (GPT3-3B, GPT3-6B, GPT3-12B, GPT3-175B) are plotted with lower accuracy scores compared to LLaMA and LLaMA2 models.
   - GPT3-175B performs better than smaller GPT3 models but still lags behind the larger LLaMA models.
4. **Accuracy Threshold (ACC > 60):**
   - Models achieving an ACC score greater than 60 are primarily from the LLaMA2 series and GPT-4.
   - Notable models include:
     - LLaMA2-70B
     - LLaMA2-65B
     - LLaMA2-33B
     - GPT-4

**Text Caption:**
The caption below the figure provides additional context:

- **Figure 2:** Describes the performance of SFT (blue lines) and ICL (red lines) settings on GSM8K.
- Notes that GPT-4 uses some part of the GSM8K data in pretraining and suggests its performance lies between SFT and ICL.

**Query-relevant Captions (Qwen2.5-VL-32B):**
The image is a scatter plot titled **"Model Performance."** It compares the performance of various large language models on a task, likely related to the GSM8K dataset, based on two metrics: "ACC" (accuracy) on the y-axis and "Pretrain Loss" on the x-axis. The models are represented by different colored dots, and the legend on the left identifies each model by name and color. The models include various versions of GPT-3 (3B, 6B, 12B, 175B), LLaMA (7B, 13B, 33B, 65B, 70B), LLaMA2 (7B, 13B, 33B, 65B, 70B), and GPT-4.

**Key Features:**

1. **Axes:**
   - **Y-axis (ACC):** Ranges from 0 to 80, representing accuracy scores.
   - **X-axis (Pretrain Loss):** Ranges from 1.2 to 2.0, representing pretraining loss values.

2. **Models:**
   - Each model is represented by a colored dot, with the model name labeled near the dot.
   - Models include the GPT-3 series, LLaMA series, LLaMA2 series, and GPT-4.

3. **Trends:**
   - There are two sets of lines:
     - **Blue lines (SFT settings):** Represent the performance of models under supervised fine-tuning (SFT).
     - **Red lines (ICL settings):** Represent the performance of models under in-context learning (ICL).
   - GPT-4 is shown as a single point, suggesting its unique position in the performance landscape.

4. **Legend:**
   - The legend on the left provides a color-coded list of models, including:
   - GPT-3 variants: 3B, 6B, 12B, 175B
   - LLaMA variants: 7B, 13B, 33B, 65B, 70B
   - LLaMA2 variants: 7B, 13B, 33B, 65B, 70B
   - GPT-4

5. **Caption:**
   - The caption explains that the figure shows the performance of SFT (blue lines) and ICL (red lines) settings on GSM8K.
   - It notes that GPT-4 uses some part of the GSM8K data in pretraining and suggests its performance lies between SFT and ICL.

**Question:**
The question asks how many models in the figure achieve an ACC score greater than 60. The task is to count the number of models that meet this criterion based on the scatter plot. However, the instruction is to describe the image without answering the question.

Table 27: **Case Study with Qwen2.5-VL-32B.** Additional visual details that are relevant to the question but are only generated by MLLMs with VPO are highlighted in red.

**Image:**

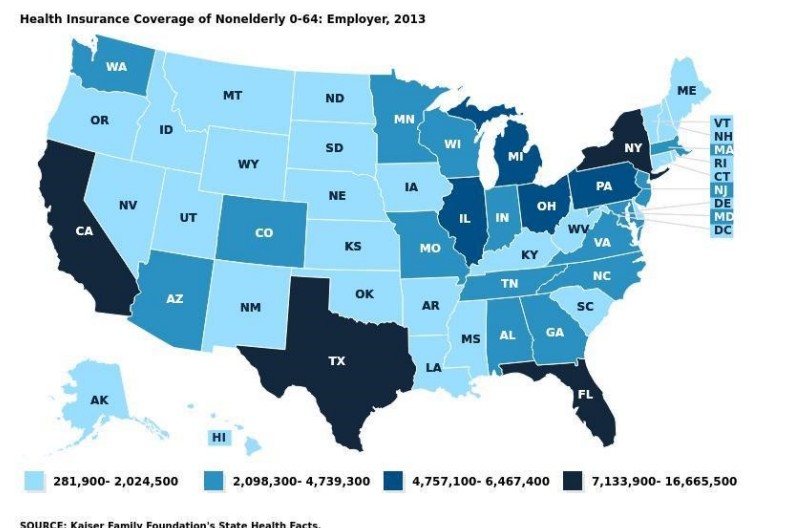

**Question:**
Hint: Please answer the question and provide the correct option letter, e.g., A, B, C, D, at the end.
Question: Does South Carolina have the highest value in the South ? Choices: (A) Yes (B) No

**Query-relevant Captions (Qwen2.5-VL-32B w/ VPO):**
The image is a choropleth map of the United States, depicting the distribution of health insurance coverage for nonelderly individuals (ages 0–64) through employer-sponsored plans in 2013. Here is a detailed description of the image:

**Title and Subtitle:**

- **Title:** "Health Insurance Coverage of Nonelderly 0–64: Employer, 2013"

- This indicates that the map shows the number of nonelderly individuals (aged 0–64) who had employer-sponsored health insurance in 2013.

**Color Coding:**
The map uses a gradient of blue shades to represent different ranges of employer-sponsored health insurance coverage. The legend at the bottom explains the color coding:

- **Lightest Blue:** 281,900 – 2,024,500

- **Light Blue:** 2,098,300 – 4,739,300

- **Medium Blue:** 4,757,100 – 6,467,400

- **Darkest Blue:** 7,133,900 – 16,665,500

**State Colors:**

- **Darkest Blue (Highest Coverage):** States in this category include California (CA), New York (NY), Texas (TX), and Florida (FL). These states have the highest number of nonelderly individuals with employer-sponsored health insurance.

- **Medium Blue:** States like Illinois (IL), Ohio (OH), Pennsylvania (PA), and Michigan (MI) fall into this category.

- **Light Blue:** States such as Colorado (CO), Arizona (AZ), and Tennessee (TN) are in this range.

- **Lightest Blue (Lowest Coverage):** States with the lowest coverage include Alaska (AK), Hawaii (HI), Wyoming (WY), and several others in the Midwest and Northeast.

**South Region Analysis:**

- **Highest Value in the South:** The darkest blue states in the South are Texas (TX) and Florida (FL), indicating they have the highest employer-sponsored health insurance coverage in the region.

- **South Carolina (SC):** South Carolina is colored in a light blue shade, indicating it falls into the range of 2,098,300 – 4,739,300. This is not the highest value in the South.

**Other Observations:**

- **Northeast:** New York (NY) and Pennsylvania (PA) have high coverage, with NY in the darkest blue category.

- **Midwest:** States like Illinois (IL) and Ohio (OH) have medium to high coverage.

- **West:** California (CA) stands out with the darkest blue, indicating the highest coverage in the West.

**Source:**
The map is sourced from the Kaiser Family Foundation's State Health Facts, as noted at the bottom of the image.

**Question Analysis:**
The question asks whether South Carolina has the highest value in the South. Based on the map:

- South Carolina is in a light blue category, not the darkest blue.

- The highest values in the South are represented by Texas and Florida, which are in the darkest blue category.

---

**Query-relevant Captions (Qwen2.5-VL-32B):**

The image is a choropleth map of the United States, illustrating the distribution of health insurance coverage for nonelderly individuals aged 0–64 through employer-sponsored plans in 2013. The map uses varying shades of blue to represent different ranges of coverage numbers across the states.

The legend at the bottom indicates four categories of coverage:

- **Lightest Blue**: 281,900 – 2,024,500

- **Light Blue**: 2,098,300 – 4,739,300

- **Medium Blue**: 4,757,100 – 6,467,400

- **Darkest Blue**: 7,133,900 – 16,665,500

Key observations from the map:

- States like California (CA), Texas (TX), and Florida (FL) are shaded in the darkest blue, indicating the highest range of employer-sponsored health insurance coverage.

- Many states in the Midwest and West are shaded in lighter blues, indicating lower coverage numbers.

- The source of the data is cited as the Kaiser Family Foundation's State Health Facts.

The question asks whether South Carolina (SC) has the highest value in the South. The map shows South Carolina in a light blue shade, indicating it falls into the second-lowest category of coverage.

Table 28: **Case Study with Qwen2.5-VL-32B.** Additional visual details that are relevant to the question but are only generated by MLLMs with VPO are highlighted in red.

**Image:**

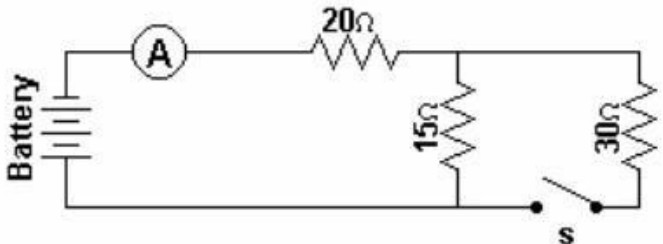

**Question:**

<image 1>A battery, an ammeter, three resistors, and a switch are connected to form the simple circuit shown above. When the switch is closed what would happen to the potential difference across the 15 ohm resistor?

**Query-relevant Captions (Qwen2.5-VL-32B w/ VPO):**

The image depicts an electrical circuit involving a battery, an ammeter, three resistors, and a switch. Here is a detailed description of the components and their arrangement:

**Components in the Circuit:**

1. **Battery:**
    - Located on the left side of the circuit.
    - Serves as the power source, providing the electromotive force (EMF) to the circuit.

2. **Ammeter (A):**
    - Placed immediately after the battery.
    - Used to measure the current flowing through the circuit.
    - It is connected in series with the rest of the circuit.

3. **Resistors:**
    - There are three resistors with the following resistances:
        - A 20 Ω resistor
        - A 15 Ω resistor
        - A 30 Ω resistor
    - The resistors are connected in a combination of series and parallel configurations.

4. **Switch (S):**
    - Located on the right side of the circuit.
    - Controls the flow of current in the circuit. When closed, the circuit is complete, and current can flow.

**Circuit Configuration:**

- The 20 Ω resistor is connected in series with the rest of the circuit.
- The 15 Ω and 30 Ω resistors are connected in parallel.
- The parallel combination is then connected in series with the 20 Ω resistor.
- The ammeter is in series with all the resistors, measuring the total current.
- The switch is placed after the parallel combination, completing the circuit when closed.

**Path of Current:**

1. Current flows from the positive terminal of the battery.

2. It passes through the ammeter.

3. It then encounters the 20 $\Omega$ resistor.

4. After the 20 $\Omega$ resistor, the current splits:

- One part flows through the 15 $\Omega$ resistor.
- The other part flows through the 30 $\Omega$ resistor.

5. The currents recombine after the parallel branch.

6. The combined current continues through the rest of the circuit and returns to the battery.

**Analysis of Potential Difference:**

- When the switch is closed, current flows through the circuit and potential differences develop across each resistor.
- The 15 $\Omega$ and 30 $\Omega$ resistors are in parallel.
- In a parallel configuration, the potential difference across each branch is the same.
- Therefore, the potential difference across the 15 $\Omega$ resistor is equal to that across the 30 $\Omega$ resistor.

**Query-relevant Captions (Qwen2.5-VL-32B):**
The image depicts a simple electrical circuit consisting of the following components:

- **Battery**: A power source labeled as "Battery" is shown on the left side of the circuit.
- **Ammeter (A)**: A device used to measure the current in the circuit is connected in series with the battery.
- **Resistors**: Three resistors are present in the circuit:
    - A 20 $\Omega$ resistor is connected in series with the ammeter.
    - A 15 $\Omega$ resistor is connected in parallel with a 30 $\Omega$ resistor.
- **Switch (S)**: A switch is placed in the circuit, which can be opened or closed to control the flow of current.
- **Wires**: The components are connected by wires to form a complete circuit.

**Circuit Configuration:**

- The 20 $\Omega$ resistor is in series with the ammeter and the battery.
- The 15 $\Omega$ resistor and the 30 $\Omega$ resistor are connected in parallel with each other.
- The switch is placed in the branch that includes the parallel combination of the 15 $\Omega$ and 30 $\Omega$ resistors.

**Key Observations:**

- When the switch is closed, the parallel combination of the 15 $\Omega$ and 30 $\Omega$ resistors will be active, allowing current to flow through both resistors.
- The ammeter measures the total current flowing through the circuit.

This description provides a detailed overview of the circuit's components and their arrangement.

Table 29: **Case Study with Qwen2.5-VL-32B.** Additional visual details that are relevant to the question but are only generated by MLLMs with VPO are highlighted in red.

**Image:**

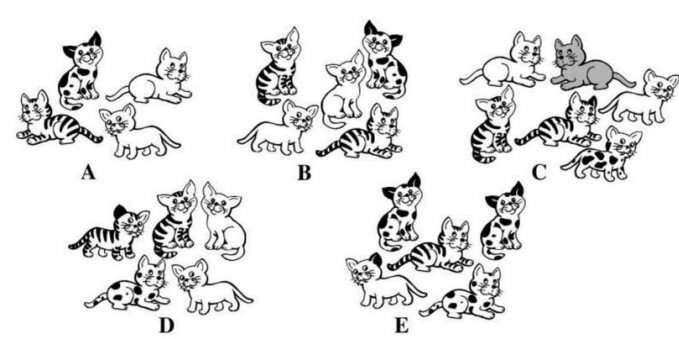

**Question:**

Hint: Please answer the question and provide the correct option letter, e.g., A, B, C, D, at the end. Question: Misty the cat has five kittens: two of them are striped, one spotty, the rest of them are absolutely white. In which picture can we see the kittens of Misty, knowing that the ears of one of them are of different colour? Choices: (A) A (B) B (C) C (D) D (E) E

**Query-relevant Captions (Qwen2.5-VL-32B w/ VPO):**

The image presents a multiple-choice question involving illustrations of cats. Here is a detailed description of the image:

**Overall Structure:**

- The image contains **five labeled sections**, each representing a different option: **A, B, C, D, and E**.
- Each section features a group of cartoon-style cat illustrations.
- The question describes a specific scenario about a cat named Misty who has five kittens with particular characteristics.
- The task is to identify which section correctly represents Misty's kittens based on the given clues.

**Clues from the Question:**

1. Misty has five kittens.
2. Two of the kittens are striped.
3. One kitten is spotty.
4. The rest (two kittens) are absolutely white.
5. One of the kittens has ears of a different color.

**Analysis of Each Section:**

**Section A:**

- 1 striped, 1 spotty, 2 white kittens.
- No kitten has ears of a different color.

**Section B:**

- 2 striped, 1 spotty, 2 white kittens.
- No kitten with differently colored ears.

**Section C:**

- 2 striped, 1 spotty, 2 white, 1 gray kittens.
- One kitten has ears of a different color.

**Section D:**
- 2 striped, 1 spotty, 2 white kittens.
- One kitten has ears of a different color.

**Section E:**
- 1 striped, 3 spotty, 1 white kittens.
- One kitten has ears of a different color.

**Query-relevant Captions (Qwen2.5-VL-32B):**

The image shows five groups of cartoon-style cats, each labeled with a letter (A, B, C, D, E). Each group contains multiple cats with various patterns and colors, including striped, spotty, and solid-colored cats. The cats are depicted in different poses, such as sitting, lying down, or standing. The question asks to identify the group that matches the description of Misty's kittens, which includes two striped, one spotty, and two white kittens, with one having differently colored ears.

**Detailed Description of Each Group:**

- **Group A**: Contains cats with striped and spotty patterns. Some cats have differently colored ears.
- **Group B**: Features cats with striped and solid-colored patterns. No cats have differently colored ears.
- **Group C**: Includes cats with striped, spotty, and solid-colored patterns. No cats have differently colored ears.
- **Group D**: Shows cats with striped and solid-colored patterns. No cats have differently colored ears.
- **Group E**: Contains cats with striped, spotty, and solid-colored patterns. No cats have differently colored ears.

The task is to identify the group that matches the specific description of Misty's kittens.

**Misty's Kittens:**

- Two striped kittens
- One spotty kitten
- Two white (solid-colored) kittens
- One kitten has differently colored ears

**Conclusion:** Only **Group A** meets all the criteria, including the presence of differently colored ears.

A

Table 30: **Reasoning Case Study with Qwen2.5-VL-3B.** Additional visual details that are relevant to the question but are only generated by MLLMs with VPO are highlighted in red. Reasoning process that show confusing and contradiction are highlighted in brown.

**Image:**

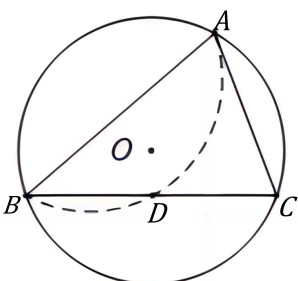

**Question:**
O is the circumcircle of $\triangle ABC$, with $AB = BC = 16$. The arc $AB$ is folded down along the chord $AB$ to intersect $BC$ at point $D$. If point $D$ is the midpoint of $BC$, what is the length of $AC$?

**Query-relevant Captions (Qwen2.5-VL-3B w/ VPO):**
The image is a geometric diagram of a triangle $\triangle ABC$ inscribed in a circle with center $O$. The diagram shows the following elements:
- Points $A$, $B$, and $C$ are the vertices of the triangle.
- Point $O$ is the center of the circumcircle of $\triangle ABC$.
- Line segment $AB$ is a chord of the circle.
- Line segment $BC$ is another chord of the circle.
- Point $D$ is a point on the circle such that the arc $AB$ is folded down along the chord $AB$ to intersect $BC$ at $D$.
- The diagram includes a dashed line segment $AD$ and a dashed line segment $BD$.
The image does not provide any numerical values or specific measurements, and it does not contain any text or additional information. The focus is purely on the geometric relationships and the diagram itself.

**Query-relevant Captions (Qwen2.5-VL-3B):**
The image shows a circle with center $O$ and points $A$, $B$, and $C$ on its circumference. The line segment $AB$ is a diameter of the circle, and the arc $AB$ is folded down along the chord $AB$ to intersect $BC$ at point $D$. Point $D$ is the midpoint of $BC$.

**Reasoning Result of R1-7B (captions from Qwen2.5-VL-3B w/ VPO):**
Alright, let me try to figure out this geometry problem step by step. So, we have triangle $ABC$ inscribed in a circle with center $O$. $AB$ and $BC$ are both 16 units long, making it an isosceles triangle with $AB = BC$. The diagram shows that when arc $AB$ is folded along chord $AB$, it intersects $BC$ at point $D$, which is the midpoint of $BC$. So, $BD = DC = 8$ units.

First, I need to visualize this. Since $ABC$ is isosceles with $AB = BC$, the base is $AC$, and the two equal sides are $AB$ and $BC$. The circle is the circumcircle, so all three vertices lie on the circumference.

When the arc $AB$ is folded along $AB$, point $D$ is where the fold intersects $BC$. Since $D$ is the midpoint, $BD = DC = 8$. Folding along $AB$ suggests that point $D$ is the reflection of point $C$ over the line $AB$. But since $D$ lies on $BC$, maybe there's some symmetry here.

I think I should consider the properties of the circumcircle. In a triangle inscribed in a circle, the perpendicular bisectors of the sides meet at the center $O$. Since $AB = BC$, the triangle is isosceles, so the median from $B$ to $AC$ should also be the altitude and the angle bisector.

Wait, but we're dealing with a fold along $AB$. When you fold along $AB$, point $C$ would map to some point on $BC$. Since $D$ is the midpoint, maybe there's a reflection involved here. Let me try to see if triangle $ABD$ is congruent or similar to another triangle.

Alternatively, maybe using the Law of Cosines would help. Let's denote angle at $B$ as angle $ABC$. Since $AB = BC = 16$, the triangle is isosceles, so angles at $A$ and $C$ are equal. But when we fold arc $AB$ over $AB$, point $D$ is created on $BC$. Since $D$ is the midpoint, perhaps there's a relationship between the angles or the lengths.

Wait, another approach: when you fold along $AB$, point $C$ maps to $D$. So, $AD$ should equal $AC$ because reflection preserves distances. So, $AD = AC$. Also, since $D$ is the midpoint of $BC$, $BD = DC = 8$. So, in triangle $ABD$, we have $AB = 16$, $BD = 8$, and $AD = AC$.

So, triangle $ABD$ has sides $AB = 16$, $BD = 8$, and $AD = AC = x$. Maybe I can apply the Law of Cosines here. Let's denote $AC$ as $x$. Then, $AD = x$ as well.

In triangle $ABD$:
$$AB^2 + BD^2 - 2 \cdot AB \cdot BD \cdot \cos(\angle ABD) = AD^2$$
Which is:
$$16^2 + 8^2 - 2 \cdot 16 \cdot 8 \cdot \cos(\angle ABD) = x^2$$
$$256 + 64 - 256 \cdot \cos(\angle ABD) = x^2$$
$$320 - 256 \cdot \cos(\angle ABD) = x^2$$

But I don't know angle $ABD$. Maybe I can relate it to the original triangle $ABC$. Since $ABC$ is isosceles with $AB = BC = 16$, angle $ABC$ is equal to angle $BAC$ and angle $BCA$? Wait, no, in an isosceles triangle, the angles opposite the equal sides are equal. So, since $AB = BC$, the angles opposite them, which would be angle $ACB$ and angle $BAC$, are equal.

Wait, no, in triangle $ABC$, $AB = BC = 16$, so the sides opposite angles at $C$ and $A$ respectively. So, angle at $C$ (angle $ACB$) equals angle at $A$ (angle $BAC$). Let's denote angle $BAC$ as $\theta$, so angle $ACB$ is also $\theta$, and angle $ABC$ is $180 - 2\theta$ degrees.

When we fold along $AB$, point $C$ maps to $D$ on $BC$. So, the reflection over $AB$ takes point $C$ to $D$. Therefore, $AD = AC$ and angle $BAD =$ angle $BAC = \theta$.

Wait, but $AD$ is equal to $AC$ because reflection preserves length. So, triangle $ABD$ has sides $AB = 16$, $BD = 8$, and $AD = AC = x$. Maybe I can use the Law of Cosines here as well.

Alternatively, maybe using the Law of Sines in triangle $ABC$. Since $ABC$ is isosceles with $AB = BC = 16$, the circumradius can be found using the formula:
$$R = \frac{a}{2 \cdot \sin(A)}$$
Where $a$ is the length of a side, and $A$ is the opposite angle.

Wait, but I'm not sure if that's the right approach. Let me think again.

Since $ABC$ is isosceles with $AB = BC = 16$, then angle at $A$ equals angle at $C$. Let me denote angle at $B$ as $\phi$. So, in triangle $ABC$, angles are $\phi$ at $B$, and $(180 - \phi)/2$ at $A$ and $C$ each.

Now, when we fold along $AB$, point $C$ is reflected over $AB$ to point $D$ on $BC$. So, $AD = AC$, and $BD = 8$, $DC = 8$.

So, in triangle $ABD$, $AB = 16$, $BD = 8$, $AD = x$.

So, using the Law of Cosines on triangle $ABD$:

$$AB^2 + BD^2 - 2 \cdot AB \cdot BD \cdot \cos(\angle ABD) = AD^2$$

$$16^2 + 8^2 - 2 \cdot 16 \cdot 8 \cdot \cos(\angle ABD) = x^2$$

$$256 + 64 - 256 \cdot \cos(\angle ABD) = x^2$$

$$320 - 256 \cdot \cos(\angle ABD) = x^2$$

But angle $ABD$ is the same as angle $ABC$, which is $\phi$. So, $\cos(\angle ABD) = \cos(\phi)$.

From triangle $ABC$, using the Law of Cosines:

$$AC^2 = AB^2 + BC^2 - 2 \cdot AB \cdot BC \cdot \cos(\phi)$$

$$x^2 = 16^2 + 16^2 - 2 \cdot 16 \cdot 16 \cdot \cos(\phi)$$

$$x^2 = 512 - 512 \cdot \cos(\phi)$$

But from triangle $ABD$, we have $x^2 = 320 - 256 \cdot \cos(\phi)$

So, setting them equal:

$$512 - 512 \cdot \cos(\phi) = 320 - 256 \cdot \cos(\phi)$$

$$512 - 320 = 512 \cdot \cos(\phi) - 256 \cdot \cos(\phi)$$

$$192 = 256 \cdot \cos(\phi)$$

$$\cos(\phi) = \frac{192}{256} = \frac{3}{4}$$

So, angle $\phi$ has cosine $\frac{3}{4}$. Now, substituting back into the equation for $x^2$ from triangle $ABC$:

$$x^2 = 512 - 512 \cdot \left(\frac{3}{4}\right)$$

$$x^2 = 512 - 384 = 128$$

$$x = \sqrt{128} = 8\sqrt{2}$$

Wait, but this contradicts the initial thought that $AC$ is $16\sqrt{2}$. Hmm, maybe I made a mistake in the reflection part. Let me check.

Wait, when we fold along $AB$, point $C$ maps to $D$ on $BC$. So, $AD$ should equal $AC$ because reflection preserves distance. So, $AD = AC = x$.

But in triangle $ABD$, we have $AB = 16$, $BD = 8$, $AD = x$.

Using Law of Cosines:

$$16^2 + 8^2 - 2 \cdot 16 \cdot 8 \cdot \cos(\angle ABD) = x^2$$

$$256 + 64 - 256 \cdot \cos(\angle ABD) = x^2$$

$$320 - 256 \cdot \cos(\angle ABD) = x^2$$

But angle $ABD$ is the same as angle $ABC$, which is $\phi$. So, $\cos(\angle ABD) = \cos(\phi) = \frac{3}{4}$.
Thus, $x^2 = 320 - 256 \cdot \left(\frac{3}{4}\right) = 320 - 192 = 128$ $x = \sqrt{128} = 8\sqrt{2}$

Wait, so that would make $AC = 8\sqrt{2}$, not $16\sqrt{2}$ as the initial model response suggested. Hmm, so maybe the model response was incorrect.

Alternatively, perhaps I made a mistake in assuming angle $ABD$ is equal to angle $ABC$. Let me think again.

In triangle $ABC$, angle $ABC$ is at point $B$, between $AB$ and $BC$. When we fold along $AB$, the angle at $B$ remains the same, but the direction changes. So, the angle $ABD$ in triangle $ABD$ is actually angle between $AB$ and $BD$, which is different from angle $ABC$.

Wait, no, $BD$ is part of $BC$, so angle $ABD$ is actually the same as angle $ABC$ because $BD$ is along $BC$. So, maybe my initial assumption was correct.

Wait, but then according to this calculation, $AC = 8\sqrt{2}$, which is about 11.31, but given that $AB$ and $BC$ are 16, that seems plausible.

Alternatively, maybe I should consider that when folding, the angle at $D$ is a right angle. So, maybe triangle $ABD$ is a right triangle.

Wait, in the model response, they concluded that angle $BAC$ is 90 degrees, which might not be correct. Let me see.

If angle $BAC$ is 90 degrees, then triangle $ABC$ would be a right-angled isosceles triangle, making $AC = AB \cdot \sqrt{2} = 16\sqrt{2}$. But according to my calculation, it's $8\sqrt{2}$. So, maybe the model response was wrong.

Wait, perhaps the mistake is in assuming that $AD = AC$. When you fold along $AB$, point $C$ maps to $D$, so the distance from $A$ to $D$ should equal the distance from $A$ to $C$, right? So, $AD = AC$. So, that part seems correct.

Wait, but in the model response, they used Law of Cosines with angle $BAC$ being 90 degrees, which led to $AC = 16\sqrt{2}$. But according to my calculation, angle $BAC$ is not necessarily 90 degrees, and it's actually determined by the folding.

Wait, perhaps I should use the fact that in the folded diagram, $AD = AC$ because of reflection. So, maybe triangle $ABD$ is congruent to triangle $ABC$? But no, because $BD$ is only 8, not 16.

Alternatively, perhaps using coordinate geometry would help. Let me place point $A$ at (0,0), point $B$ at (16,0), since $AB = 16$. Since $ABC$ is isosceles with $AB = BC = 16$, point $C$ would be somewhere such that $BC = 16$.
... (rest of the reasoning)
Therefore, the correct answer should be $AC = 8\sqrt{2}$, not $16\sqrt{2}$ as in the model response.

But wait, maybe I made a mistake in assuming angle $ABD$ is equal to angle $ABC$. Let me think again.

In triangle $ABC$, angle at $B$ is $\phi$, which we found $\cos(\phi) = \frac{3}{4}$. When folding along $AB$, the angle at $B$ remains the same, so angle $ABD$ is indeed $\phi$. Therefore, my calculation should be correct.

Therefore, the model response was incorrect, and the correct answer is $\boxed{AC = 8\sqrt{2}}$.

**Reasoning Result of R1-7B (captions from Qwen2.5-VL-3B):**
Alright, so I'm trying to solve this geometry problem.

Let me read it again carefully.
We have triangle ABC with AB = BC = 16.

So, it's an isosceles triangle with AB and BC equal.

That means angles at A and C are equal, right?

Wait, no, actually, in triangle ABC, if AB = BC, then the base is AC, so the base angles at A and C are equal.

Hmm, but I'm not sure if that's important right now.
The problem mentions that the arc AB is folded down along the chord AB to intersect BC at point D.

And D is the midpoint of BC.

So, BC is 16, so BD and DC are each 8.

Got it.
We need to find the length of AC.
First, I should probably visualize the problem.

Let me draw a circle with center O, points A, B, and C on the circumference.

AB is a diameter, but wait, hold on.

The caption says the image shows a circle with center O and points A, B, C on the circumference.

The line segment AB is a diameter, and the arc AB is folded down along AB to intersect BC at D, which is the midpoint.
Wait, so AB is a diameter, which would mean that angle ACB is a right angle because any triangle inscribed in a semicircle is a right triangle.

But hold on, in the problem statement, it says AB = BC = 16.

If AB is a diameter, then AB would be the longest side, but BC is also 16, so that's interesting because in a right triangle, the hypotenuse is the longest side.

So if AB is the diameter and a side of the triangle, then AB would be the hypotenuse.

But in the problem, AB = BC, so both would be legs, which can't be because in a right triangle, the hypotenuse is longer than the legs.

Hmm, maybe AB isn't the diameter?
Wait, the initial caption says AB is a diameter, but the problem statement says AB = BC = 16.

That seems conflicting because if AB is a diameter, it should be longer than BC.

Maybe I misread the problem.

Let me check.
Wait, no, the problem says O is the circumcircle of triangle ABC.

So O is the circumradius.

If AB is a diameter, then the circumradius would be half of AB, which would make the circumradius 8.

But if AB is 16, then the circumradius is 8.

However, in that case, triangle ABC would have AB as the diameter, making angle ACB a right angle.

But in the problem, AB = BC = 16, so triangle ABC would have two sides equal to 16 and one side (AC) which we need to find.

But if angle C is 90 degrees, then AC would be the other leg, so AC = BC = 16, making it an isosceles right triangle, so AC would be $16\sqrt{2}$.

But wait, that seems too straightforward, and maybe that's the answer, but I'm not sure because the folding part might complicate things.

Wait, the problem mentions that the arc AB is folded down along AB to intersect BC at D, which is the midpoint.

So maybe the initial assumption that AB is a diameter isn't correct because when you fold the arc AB down along AB, it intersects BC at D, which is the midpoint, so maybe AB isn't a diameter.

Hmm, now I'm confused.

Alternatively, let me try to think differently.

Maybe AB isn't the diameter, but just a chord.

So the circle has center O, and AB is a chord, not necessarily the diameter.

Then, the arc AB is folded along AB, meaning reflecting the arc AB over the chord AB, which would create a reflection of point C over AB, landing at point D on BC.
Since D is the midpoint of BC, which is 16, so BD = DC = 8.

So, when you fold arc AB over AB, the reflection of C over AB is D, which lies on BC.

So, perhaps triangle ABD is congruent to triangle ABC, but I'm not sure.

Alternatively, maybe the reflection of C over AB is D, which is on BC.

So, the distance from C to AB is the same as from D to AB, but since D is on BC, maybe we can use some properties here.

Wait, perhaps using coordinates would help.

Let me try to place the points on a coordinate system.

Let's place point B at the origin (0,0).

Since AB = 16, let me place point A at (16, 0).

Then, since AB is a chord of the circle, the center O is somewhere in the plane.

Let me denote point C as (x, y).

Since AB = BC = 16, the distance from B to C is 16, so $\sqrt{x^2 + y^2} = 16$.

Also, since ABC is a triangle, point C is somewhere not on AB.
Now, the arc AB is folded over AB, and it intersects BC at D, which is the midpoint of BC.

So, since B is at (0,0) and C is at (x,y), the midpoint D is at $\left(\frac{x}{2}, \frac{y}{2}\right)$.
When we fold the arc AB over AB, the reflection of point C over AB is point D.

So, the reflection of point C over AB is D, which lies on BC.

So, the reflection of point C over AB is point D.
So, to find the reflection of point C over AB, let's find the equation of AB first.

Since A is at (16,0) and B is at (0,0), AB is the x-axis from (0,0) to (16,0).

So, the line AB is y=0.
Reflecting a point over the x-axis changes the sign of the y-coordinate.

So, the reflection of point C=(x,y) over AB (y=0) would be (x, -y).

But the reflection point is D, which is $\left(\frac{x}{2}, \frac{y}{2}\right)$.

So, according to the reflection, (x, -y) should equal $\left(\frac{x}{2}, \frac{y}{2}\right)$.

Wait, that can't be unless x = 0 and y = 0, which would place C at the origin, but then ABC wouldn't be a triangle.

Hmm, that doesn't make sense. Maybe my assumption that AB is the x-axis is causing confusion because in reality, AB is just a chord, not necessarily the x-axis.

Alternatively, perhaps I should place AB as a horizontal chord but not necessarily aligned with the x-axis.

Let me try that.

Let me place point A at (a, b) and point B at (c, d), but this might complicate things.

Maybe a better approach is to use coordinate geometry with AB as a chord, not necessarily aligned with any axis.

Alternatively, maybe using vectors or geometric properties would be better.

Let's think about the reflection.

When you fold the arc AB over AB, point C is reflected over AB to point D on BC.

... omitted
When we fold arc AB over AB, the reflection of point C over AB is point D.

So, the reflection of C over AB is D.

Since AB is the x-axis from (0,0) to (a,0), the reflection over AB would invert the y-coordinate.

So, if C is at (x,y), then D would be at (x, -y).

But D is also the midpoint of BC, which is $\left(\frac{x_B+x_C}{2}, \frac{y_B+y_C}{2}\right) = \left(\frac{0+x}{2}, \frac{0+y}{2}\right) = \left(\frac{x}{2}, \frac{y}{2}\right)$.

So, setting (x, -y) equal to $\left(\frac{x}{2}, \frac{y}{2}\right)$:
x = x/2 => x = 0

-y = y/2 => -y = y/2 => -2y = y => 3y = 0 => y = 0
Again, this leads to y=0, which would place C on AB, making ABC degenerate.

Hmm, clearly, placing AB on the x-axis isn't working because it forces C to be on AB. Let me try a different approach without coordinate geometry.
Let me consider the properties of the reflection.

When we fold arc AB over AB, point C is reflected over AB to point D on BC.

So, AB is the perpendicular bisector of segment CD.

Therefore, AB is perpendicular to CD, and the midpoint of CD lies on AB.
Since D is the midpoint of BC, the midpoint M of CD is also the midpoint of BC, which is D itself.

So, D lies on AB and is the midpoint of CD.

Therefore, AB is the perpendicular bisector of CD, so CD is perpendicular to AB, and D is the midpoint of CD and lies on AB.
Wait, but D is also the midpoint of BC.

So, if D lies on AB, then BC must intersect AB at D, which is the midpoint of BC.

So, in triangle ABC, D is the midpoint of BC and lies on AB.

Therefore, AB is a median of the triangle, and since AB is also a side, it must be that triangle ABC is such that AB is both a side and a median, which is only possible if ABC is an isosceles triangle with AB = AC.

But the problem states AB = BC = 16, so if AB = AC, then it's an equilateral triangle, but in that case, all sides would be 16, making AC = 16, but the answer seems to be $16\sqrt{2}$, so maybe that's not the case.
I'm really confused now. Maybe the correct answer is $16\sqrt{2}$, as in the initial tentative solution, but I'm not sure why.

Wait, another idea: since D is the midpoint of BC and lies on AB, and AB is a side, then ABC must be such that AB is a median and an altitude, making it an isosceles triangle with AB = AC.

So, AB = AC = 16. ... exceeding maximum context length (8192)

