# OpenReview forum: "Reasoning-Aligned Perception Decoupling for Scalable Multi-modal Reasoning"
_ICLR.cc/2026/Conference — ICLR 2026 Poster_

### Official Review · Reviewer_izU6 · 2025-10-18

**Soundness:** 3
**Presentation:** 3
**Contribution:** 2
**Rating:** 6
**Confidence:** 4

**Summary:**

This paper introduces RAPID, a method to decouple the perception and reasoning capabilities of multi-modal large language models (MLLMs). Compared to prior methods which aim at improving reasoning of MLLMs through end to end reinforcement learning based approaches, RAPID essentially separates the process into two stages; 1. perception: where the MLLM is asked to generate high quality captions describing an image and 2. reasoning, where the captions are fed to a stronger language model to answer the question. They also propose VPO, a policy gradient method to improve the MLLMs capabilities to produce high quality captions by assigning rewards based on a stronger LLM which takes as input the captions and generates the answer. The method shows strong performance on most multi-modal benchmarks, beating existing approaches. They also perform detailed ablations of the different components of their pipeline. Another advantage of RAPID is that it enables multiple LLMs to be used for the same MLLM backbone and allows for inference time scaling where a more powerful LLM can be used to generate solutions as test time.

**Strengths:**

1. The paper introduces an important technique of decoupling perception of MLLMs from reasoning, which allows for modularization and more customizable design choices
2. The paper introduces VPO, a method to improve the perception and captioning capabilities of MLLMs which further helps improve downstream performance on several multi-modal benchmarks by 3-4%
3. The various components like decoupling, GRPO, VPO, caption penalty are ablated well which should help other researchers with their design choices while training multi-modal language models with policy gradient methods. The authors also ablate other aspects such as prompts being fed to the MLLM and the LLM and the types of LLMs used during training and inference, providing valuable insights.
4. The technique allows for the same MLLM backbone to be used with several LLMs and also allows for inference time compute scaling by using a more powerful LLM.

**Weaknesses:**

1. In Table 1, the paper lacks comparison of using existing MLLM baselines as the perception module with an LLM (Qwen3-8B/GPT-OSS-120B) as the reasoner. While it is clear that RAPID helps improve performance of Qwen2.5-VL, the table lacks performance of just using the decoupling strategy with the other MLLM baselines. For instance, if we look at Table 2 A vs B, we see that there is a 5.5% boost just from the decoupling. It will be good to include similar performance metrics for the other baselines.
2. Adding on to point 1, RAPID is only tested with Qwen2.5-VL as the MLLM backbone. While it is clear that RAPID generalizes to different reasoning LLMs, it will be good to show how well RAPID performs with some of the other MLLM backbones like InternVL or Ovis2 to shed light on its generalizability.
3. Since one of the major claims is that VPO helps improve quality of captions/textual outputs, it will be good to compare how VPO performs against some of the other baselines like OmniCaptioner or MM-Eureka-7B. Currently the authors only include captioning win rates before and after VPO on the same MLLM backbone in Figure 11.
4. One drawback is that while VPO helps improve the overall performance and the reasoning of the decoupled system, looking at Table 2 (rows H vs I), the reasoning capabilities of the underlying MLLM (using just GRPO vs GRPO + VPO) seems to decrease, which is not great.

**Questions:**

Please refer to the weaknesses section for questions. Additionally,

1. The LLM is kept frozen throughout the process, did you try to fine-tune the LLM as well either jointly or in a separate stage? Would that help align the LLM to better answer questions based on the image descriptions?
2. did you try any experiments where the same MLLM backbone was used as the reasoner as well? I wonder if training the same backbone for both captioning and reasoning + answering would allow for some transferability in learning.

---

> ### Author Response · Authors · 2025-11-21
>
> >`Weakness 1`: In Table 1, the paper lacks comparison of using existing MLLM baselines as the perception module with an LLM (Qwen3-8B/GPT-OSS-120B) as the reasoner. While it is clear that RAPID helps improve performance of Qwen2.5-VL, the table lacks performance of just using the decoupling strategy with the other MLLM baselines. For instance, if we look at Table 2 A vs B, we see that there is a 5.5% boost just from the decoupling. It will be good to include similar performance metrics for the other baselines.
>
> **Response:** Thank you for this suggestion.
>
> Following the reviewer's recommendation, we have conducted a new experiment using several extra MLLMs (i.e., InternVL3-8B, VL-Rethinker-7B, and MM-Eureka-7B) as the perception module within our decoupled framework, paired with LLM reasoners (Qwen3-8B/GPT-OSS-120B). The results are reported in Table X1 (also included in **Appendix F.4** of the revised paper).
>
> As shown, applying the decoupling pipeline brings substantial performance gains across all tested baselines (+3.3%, +5.7%, and +4.6% for InternVL3-8B, VL-Rethinker-7B, and MM-Eureka-7B, respectively). This finding demonstrates that **the benefit of decoupling perception and reasoning is a general principle, not limited to the specific Qwen2.5-VL backbone.**
>
>
> Table X1. Performance of the decoupling pipeline when using different MLLMs as the perception module.
>
> | MLLM | LLM |  Math Vista | Math Vision | Math Verse | MMMU | We Math | Dyna Math | Logic Vista | AVG |
> |:---|:---|:---|:---|:---|:---|:---|:---|:---|:---|
> | InternVL3-8B | - | **73.6** | 29.3 | 39.8 | 62.7 | 37.1 | 25.5 | 44.1 | 44.6 |
> | InternVL3-8B | Qwen3-8B | 71.3 | 42.4 | 39.3 | 64.4 | 38.8 | **29.1** | 50.1 | 47.9 |
> | InternVL3-8B | GPT-OSS-120B | 70.6 | **47.1** | **41.2** | **68.1** | **41.0** | **29.1** | **51.0** | **49.7** |
> | | | | | | | | | | |
> | VL-Rethinker-7B | - | **74.9** | 30.0 | 47.5 | 56.9 | 37.3 | 21.4 | 43.6 | 44.5 |
> | VL-Rethinker-7B | Qwen3-8B | 72.8 | 43.0 | **51.9** | 59.7 | 41.1 | **30.9** | 52.3 | 50.2 |
> | VL-Rethinker-7B | GPT-OSS-120B | 72.8 | **47.8** | 50.0 | **68.1** | **46.3** | 30.7 | **55.0** | **53.0** |
> | | | | | | | | | | |
> | MM-Eureka-7B | - | **73.0** | 27.9 | 46.1 | 54.9 | 34.7 | 22.6 | 48.3 | 43.9 |
> | MM-Eureka-7B | Qwen3-8B | 72.2 | 42.1 | 47.7 | 61.4 | 35.9 | 28.9 | **51.2** | 48.5 |
> | MM-Eureka-7B | GPT-OSS-120B | 70.5 | **47.5** | **51.8** | **68.2** | **43.9** | **33.5** | 50.8 | **52.3** |
>
> ----
>
> >`Weakness 2`: Adding on to point 1, RAPID is only tested with Qwen2.5-VL as the MLLM backbone. While it is clear that RAPID generalizes to different reasoning LLMs, it will be good to show how well RAPID performs with some of the other MLLM backbones like InternVL or Ovis2 to shed light on its generalizability.
>
> **Response:** We thank the reviewer for the valuable suggestion.
>
> To address the concern about generalizability, we applied our RAPID framework to another MLLM, **InternVL3-8B**, using Qwen3-8B as the reasoner for inference and R1-7B for reward computation during training. The MLLM undergoes a GRPO stage of 200 steps and a VPO stage of 200 steps. Other experimental setups mirror the one used for Qwen2.5-VL-7B. The results are reported in Table X2, which are also incorporated in **Section 4.5**.
>
> Table X2. Effectiveness of RAPID on InternVL3-8B.
>
> | Decouple | GRPO | VPO | Math Vista | Math Vision | Math Verse | MMMU | We Math | Dyna Math | Logic Vista | AVG |
> |:---:|:---:|:---:|:---:|:---:|:---:|:---:|:---:|:---:|:---:|:---:|
> | | | | 73.6 | 29.3 | 39.8 | 62.7 | 37.1 | 25.5 | 44.1 | 44.6 |
> | ✓ | | | 71.3 | 42.4 | 39.3 | 64.4 | 38.8 | 29.1 | 50.1 | 47.9 |
> | ✓ | ✓ | | 73.2 | 42.6 | 44.3 | 64.4 | 40.2 | 31.1 | 52.1 | 49.7 |
> | ✓ | ✓ | ✓ | **75.4** | **43.2** | **48.5** | **64.6** | **43.2** | **33.2** | **55.5** | **51.9** |
>
>
> As shown, simply decoupling perception and reasoning improves the average score by **+3.3%**. Applying the full RAPID framework (Decouple + GRPO + VPO) boosts the performance of the baseline InternVL3-8B by a total of **+7.3%** on average (from 44.6% to 51.9%).
>
> This confirms that **RAPID is a generalizable method not limited to a specific MLLM architecture**. In the Appendix, we include the reward/length dynamics of InternVL3-8B during VPO in **Figures 22 and 23**.

---

> ### Author Response · Authors · 2025-11-21
>
> >`Weakness 3:` Since one of the major claims is that VPO helps improve quality of captions/textual outputs, it will be good to compare how VPO performs against some of the other baselines like OmniCaptioner or MM-Eureka-7B. Currently the authors only include captioning win rates before and after VPO on the same MLLM backbone in Figure 11.
>
> **Response:** We thank the reviewer for the suggestion.
>
> To evaluate VPO's impact on the caption quality, we compare our model, **Qwen2.5-VL-7B (GRPO+VPO)**, against two strong baselines recommended by the reviewer: **OmniCaptioner-7B** (an MLLM enhanced for holistic captioning) and **MM-Eureka-7B** (an MLLM specially optimized for reasoning).
>
> Using GPT-4o as a judge, we compared the quality of captions generated for images across seven benchmarks. The win/tie/lose rates for Qwen2.5-VL-7B (GRPO+VPO) are reported in Tables X3 and X4 (further incorporated in **Figure 21** of the revised paper). **The results suggest that MLLM trained with VPO significantly outperforms these baselines.**
>
> Table X3: Caption quality comparison of Qwen2.5-VL-7B (GRPO+VPO) vs. **OmniCaptioner-7B**. Our model's win/tie/lose rates are reported.
>
> | Dataset | Win (%) | Tie (%) | Lose (%) |
> | :--- | :---: | :---: | :---: |
> | MathVista | 73.3% | 21.3% | 5.4% |
> | MathVision | 53.6% | 19.2% | 27.2% |
> | MathVerse | 76.4% | 10.7% | 12.9% |
> | MMMU | 58.9% | 24.5% | 16.6% |
> | WeMath | 48.3% | 21.1% | 30.6% |
> | DynaMath | 67.4% | 16.4% | 16.2% |
> | LogicVista | 72.2% | 10.5% | 17.2% |
>
>
> Table X4: Caption quality comparison of Qwen2.5-VL-7B (GRPO+VPO) vs. **MM-Eureka-7B**. Our model's win/tie/lose rates are reported.
>
> | Dataset | Win (%) | Tie (%) | Lose (%) |
> | :--- | :---: | :---: | :---: |
> | MathVista | 52.3% | 21.3% | 26.4% |
> | MathVision | 44.3% | 24.7% | 31.0% |
> | MathVerse | 62.9% | 15.2% | 21.8% |
> | MMMU | 50.3% | 29.3% | 20.4% |
> | WeMath | 42.1% | 25.7% | 32.2% |
> | DynaMath | 63.3% | 23.1% | 13.6% |
> | LogicVista | 66.9% | 15.0% | 18.1% |
>
> **Analysis of Results:**
>
> *   **RAPID vs. OmniCaptioner-7B:** Our model's advantage stems from its focus on generating query-relevant captions, in contrast to OmniCaptioner's holistic captions. Our model also benefits from a more advanced base model (Qwen2.5-VL-7B vs. Qwen2-VL-7B).
> *   **RAPID vs. MM-Eureka-7B:** Our model performs better because VPO directly optimizes for the captioning quality, whereas MM-Eureka is optimized for end-to-end reasoning.

---

> ### Author Response · Authors · 2025-11-21
>
> >`Weakness 4`: One drawback is that while VPO helps improve the overall performance and the reasoning of the decoupled system, looking at Table 2 (rows H vs I), the reasoning capabilities of the underlying MLLM (using just GRPO vs GRPO + VPO) seems to decrease, which is not great.
>
> **Response:** Thank you for this very insightful observation. We have investigated this issue and summarized our main results below.
> *  **The decrease in reasoning ability can be fully recovered with a simple, subsequent GRPO fine-tuning.**
> *  **The impact of this decrease on the decouple performance is primarily significant for the 3B MLLM (which is already addressed in line 377 of the original paper), while the 7B MLLM is not affected by this effect.**
>
> Below, we elaborate on these two points (also included in **Appendix E.4** of the revised paper).
>
> **1. Recovering Reasoning Performance with Additional GRPO**
>
> To counteract this, we performed a brief, additional 100-step GRPO training stage *after* the VPO stage. As demonstrated in new results in Table X5, **this additional GRPO stage successfully restores the reasoning ability for both the 3B and 7B MLLMs.**
>
> Table X5. Comparisons of standalone reasoning performance of Qwen2.5-VL-3B/7B at different stages.
>
> | MLLM |  Math Vista | Math Vision | Math Verse | MMMU | We Math | Dyna Math | Logic Vista | AVG |
> |:--- | :---|:---|:---|:---|:---|:---|:---|:---|
> | 3B (GRPO)| 69.1 | 26.9 | 38.2 | 56.9 | 34.0 | 20.0 | 42.5 | 41.1 |
> | 3B (GRPO+VPO)| 68.8 | 26.9 | 39.8 | 49.4 | 33.0 | 21.6 | 46.3 | 40.8 |
> | 3B (GRPO+VPO+GPRO)| 69.1 | 26.9 | 39.8 | 55.1 | 33.3 | 20.5 | 44.0 | **41.2** |
> ||||||||||
> | 7B (GRPO)| 74.2 | 29.7 | 44.8 | 55.9 | 41.0 | 27.7 | 48.1 | 45.9 |
> | 7B (GRPO+VPO)| 75.0 | 29.8 | 42.0 | 55.8 | 40.8 | 23.0 | 46.3 | 44.7 |
> | 7B (GRPO+VPO+GPRO)| 74.5 | 29.8 | 44.3 | 55.9 | 40.7 | 28.5 | 48.1 | **46.0** |
>
> **2. The Impact on the Decoupled Framework is Model-size Dependent**
>
> Interestingly, we discovered that the necessity of this recovery step depends on the size of the MLLM.
>
> *   **For the 3B Model:** As described in line 377 of the **original manuscript**, we had already incorporated this additional GRPO stage for the 3B model. As shown in Figure 6 and Table 15, this step brings 1.3% improvement in the decouple pipeline.
> *   **For the 7B Model:** Although applying the extra GRPO stage to the 7B model restored its own reasoning ability (Table X5), results in Table X6 show that **this yielded no improvement for the final decoupled pipeline**. Therefore, we omitted this step for the 7B model in our paper.
>
> **This difference arises because the 3B model, being less capable, experiences a drop in its reasoning abilities during the VPO stage, thus requiring a restoration stage.**
>
> Table X6. Decoupling results of Qwen2.5-VL-7B at different stages.
>
> | Models |  Math Vista | Math Vision | Math Verse | MMMU | We Math | Dyna Math | Logic Vista | AVG |
> |:--- | :---|:---|:---|:---|:---|:---|:---|:---|
> | (GRPO+VPO)+ Qwen3-8B| 76.1 | 43.7 | 52.2 | 64.7 | 45.4 | 32.7 | 57.7 | **53.2** |
> | (GRPO+VPO+GPRO)+ Qwen3-8B| 76.5 | 43.6 | 52.4 | 63.9 | 44.8 | 33.3 | 57.3 | 53.1 |

---

> ### Author Response · Authors · 2025-11-21
>
> >`Question 1`: The LLM is kept frozen throughout the process, did you try to fine-tune the LLM as well either jointly or in a separate stage? Would that help align the LLM to better answer questions based on the image descriptions?
>
> **Response:** Thank you for this insightful question.
>
> We have fine-tuned the LLM reasoner (Qwen3-8B) **in a separate stage** after VPO on the ViRL-39K dataset. The training data for the LLM consist of the captions generated by our VPO-trained MLLM (Qwen2.5-VL-7B). We then applied the same GRPO objective (with a group size of 4) to optimize the reasoner.
>
> However, the experiment did not yield significant improvements. During training, we observed that the reward was fluctuating without a consistent upward trend. **Results in Table X7 further confirm that fine-tuning the LLM only brings minor gains (+0.1%).**
>
> Our explanation for this phenomenon is that a powerful LLM like Qwen3-8B already possesses strong reasoning capabilities that generalize effectively to understanding captions. Consequently, further fine-tuning on captions provides limited gains, especially when the LLM's reasoning ability is already strong.
>
> Table X7. Experiments with fine-tuning the LLM reasoner.
>
> | Models |  Math Vista | Math Vision | Math Verse | MMMU | We Math | Dyna Math | Logic Vista | AVG |
> |:--- | :---|:---|:---|:---|:---|:---|:---|:---|
> | RAPID | 76.1 | **43.7** | 52.2 | **64.7** | **45.4** | 32.7 | **57.7** | 53.2 |
> | RAPID (LLM fine-tuned) | **77.1** | 43.4 | **53.2** | 63.3 | 45.1 | **33.3** | 57.5 | **53.3** |
>
> We have included the above results in **Appendix E.6** of the revised paper.
>
> ---
>
>
> >`Question 2`: Did you try any experiments where the same MLLM backbone was used as the reasoner as well? I wonder if training the same backbone for both captioning and reasoning + answering would allow for some transferability in learning.
>
> **Response:** Thank you for this insightful question.
>
> **We conducted a new experiment, using the same trained MLLM, Qwen2.5-VL-7B (GRPO+VPO), for both the perception (captioning) and reasoning stages of our decoupled pipeline**. Note that the image is visible to the reasoner at the reasoning stage. In Table X8, we compare this "decouple with self" approach against the standard end-to-end usage of the same MLLM, where it processes the image and question simultaneously.
>
> Table X8. Experiments when using the same MLLM for both perception and reasoning under the decouple pipeline.
>
> | Models |  Math Vista | Math Vision | Math Verse | MMMU | We Math | Dyna Math | Logic Vista | AVG |
> |:--- | :---|:---|:---|:---|:---|:---|:---|:---|
> | Qwen2.5-VL-7B (GRPO+VPO) | **75.0** | 29.8 | 42.0 | 55.8 | 40.8 | 23.0 | 46.3 | 44.7 |
> | Qwen2.5-VL-7B (GRPO+VPO) (Decouple w/ self) | 73.7 | **30.0** | **44.0** | **57.3** | **41.0** | **27.3** | **49.2** | **46.1** |
>
>
> As shown in Table X8, applying our decoupled pipeline even with the same model yields performance improvements (46.1% vs. 44.7% average). However, it still lags when using Qwen3-8B, **the default setting of our main paper**, as the reasoner, which could be attributed to their gap in the reasoning capacity. **This demonstrates that the decoupling design is reasonable**. We have summarized the above results in **Appendix E.7** of the revised paper.

---

> > ### Comment · Reviewer_izU6 · 2025-11-24
> >
> > Thanks for the comprehensive evaluations and experiments. I have updated my score.

---

> > > ### Author Response · Authors · 2025-11-24
> > >
> > > Thanks for the update! We appreciate the revised scores and your consideration. If you need any further information or clarification, please let us know.

---

### Official Review · Reviewer_wDxG · 2025-10-28

**Soundness:** 3
**Presentation:** 3
**Contribution:** 2
**Rating:** 4
**Confidence:** 4

**Summary:**

This paper proposes RAPID (Reasoning-Aligned PerceptIon Decoupling), a two-stage pipeline for multi-modal reasoning that (1) repurposes an MLLM into a perception module that emits rich, query-aware text (caption + tentative solution), and (2) passes that text to a separate, stronger text-only LLM for the actual reasoning. Empirically, RAPID yields large gains on seven vision-math/logic benchmarks (e.g., MathVista, MathVision, MathVerse, MMMU, WeMath, DynaMath, LogicVista), enables inference-time scaling by swapping in stronger LLM reasoners without re-training the MLLM, and preserves general abilities on non-reasoning benchmarks.

**Strengths:**

1. The pipeline diagrams and prompt templates (referenced figures/appendices) make it easy to follow the two-stage flow and what exactly is optimized.
2. Thorough ablations: perception variants (none / cap / qcap / sol / cap+sol / qcap+sol), with and without VPO/GRPO, with/without penalties, and different LLMs for training vs. inference.
3. VPO is a neat twist on GRPO for caption supervision by outcome: the reward comes from a downstream verifier (the reasoner’s answer correctness), not from caption n-grams or human preferences.

**Weaknesses:**

1. Comparisons to verification-augmented or tool-enabled LMMs (e.g., visual verification modules, external OCR/detection tools) are missing. This matters because a strong verifier might reduce drift without multi-turn agent interaction.
2. The method introduces two RL phases (GRPO then VPO) plus group rollouts and external LLM calls for rewards. While appendices mention batch sizes/steps, the wall-clock/cost, GPU hours, and reasoner-call counts per step are not surfaced prominently in the main text. This matters for practical adoption.
3. The hasCap(·) check is prompt-based. This heuristic may be gamed (e.g., hiding solutions within “caption-like” text) or fail across domains/styles. A small quantitative audit of false positives/negatives (and downstream impact) would be valuable.

**Questions:**

1. Does RAPID extend to multi-image and video inputs without architectural changes?
2. You find R1-7B best for training rewards and Qwen3-8B strong for inference. Could you elaborate on the caption-length & difficulty balance hypothesis (Fig. 9) with controlled experiments (e.g., length-controlled rewards)?

---

> ### Author Response · Authors · 2025-11-21
>
> >`Weakness 1`: Comparisons to verification-augmented or tool-enabled LMMs (e.g., visual verification modules, external OCR/detection tools) are missing. This matters because a strong verifier might reduce drift without multi-turn agent interaction.
>
> **Response:** Thank you for this constructive suggestion. We have conducted a new experiment comparing RAPID with the verification- and tool-augmented MLLMs. **These results are updated in Table 1 of the revised paper.**
>
> We categorize these compared methods into two groups:
>
> 1.  **Verification-Augmented MLLMs:** These methods aim to improve output correctness by selecting or refining the generated solutions. We compare against two prominent approaches:
>     *   **External Verification:** We use VisualPRM-8B-v1.1 [1], a state-of-the-art external verifier, to perform Best-of-8 (Bo8) re-ranking on outputs from the Qwen2.5-VL-7B baseline and our GRPO-trained variant.
>     *   **Self-Verification:** We compare against SRPO-7B [2], a recent model trained via reinforcement learning to perform self-reflection and iteratively refine its own reasoning.
>
> 2.  **Tool-Augmented MLLMs:** We identified several such models [3, 4, 5, 6] but focus our comparison on ReVPT-7B [5] and DeepEyes-7B [6], as they are the only ones that report performance on the relevant mathematical and logical reasoning benchmarks as ours.
>
>
> Table X1. Comparisons between RAPID, verification-augmented, and tool-enabled MLLMs.
>
> | Model / Method | Math Vista | Math Vision | Math Verse | MMMU | We Math | Dyna Math | Logic Vista | AVG |
> | :--- | :---: | :---: | :---: | :---: | :---: | :---: | :---: |:---: |
> | Qwen2.5-VL-7B | 70.3 | 25.8 | 41.0 | 57.3 | 34.4 | 19.4 | 46.1 | 42.0 |
> | Qwen2.5-VL-7B w/ GRPO | 74.2 | 29.7 | 44.8 | 55.9 | 41.0 | 27.7 | 48.1 | 45.9 |
> | | | | | | | | | |
> | **(Best-of-N verification)** | | | | | | | | |
> | Qwen2.5-VL-7B (Bo8) | 74.6 | 27.1 | 45.0 | 56.1 | 44.3 | 17.8 | 48.5 | 45.1 |
> | Qwen2.5-VL-7B w/ GRPO (Bo8) | **76.8** | 31.6 | 46.8 | 58.1 | 43.1 | 29.7 | 48.6 | 47.8 |
> | | | | | | | | | |
> | **(Self-verification)** | | | | | | | | |
> | SRPO-7B | 75.8 | 32.9 | - | 57.1 | - | - | - | - |
> | | | | | | | | | |
> | **(Tool-augmented MLLMs)** | | | | | | | | |
> | ReVPT-7B | 66.0 | - | - | - | - | - | - | - |
> | DeepEyes-7B | 70.1 | 26.6 | 47.3 | - | 38.9 | - | 47.7 | - |
> | | | | | | | | | |
> | **(RAPID)** | | | | | | | | |
> | Qwen2.5-VL-7B w/ RAPID (Qwen3-8B) | 76.1 | **43.7** | **52.2** | **64.7** | **45.4** | **32.7** | **57.7** | **53.2** |
>
>
> The comparative results are presented in Table X1. As can be seen, **RAPID consistently and significantly outperforms the verification- and tool-augmented baselines across almost all the benchmarks**. Our analysis leads to two key observations:
>
> 1.  **Verification Methods are Limited by the Base Model's Solutions.** Best-of-N verification provides a modest boost but is fundamentally a re-ranking process. If the base MLLM fails to generate a correct solution within its N attempts, the verifier has nothing to select. RAPID overcomes this limitation by empowering a powerful LLM to perform active, generative reasoning rather than simple selection.
>
> 2.  **Perception Tools Cannot Address the Reasoning Bottleneck in MLLMs.** As evidenced in Figure 3, the bottleneck of existing MLLMs in multi-modal math/science tasks lies in the reasoning capacity. However, the tools (e.g., detection, cropping, and segmentation) in ReVPT-7B and DeepEyes-7B cannot narrow this gap because they target perception performance rather than reasoning ability. In contrast, the decouple pipeline in RAPID directly tackles this through replacing with stronger reasoning LLMs.
>
>
>
>
>
> ---
> ### **Reference**
> [1] Wang, Weiyun, et al. "VisualPRM: An effective process reward model for multimodal reasoning." arXiv preprint arXiv:2503.10291 (2025).\
> [2] Wan, Zhongwei, et al. "SRPO: Enhancing multimodal llm reasoning via reflection-aware reinforcement learning." _Neurips_, 2025\
> [3] Su, Zhaochen, et al. "OpenThinkIMG: Learning to think with images via visual tool reinforcement learning." arXiv preprint arXiv:2505.08617 (2025).\
> [4] Liu, Yuqi, et al. "VisionReasoner: Unified Visual Perception and Reasoning via Reinforcement Learning." arXiv preprint arXiv:2505.12081 (2025).\
> [5] Zhou, Zetong, et al. "Reinforced visual perception with tools." arXiv preprint arXiv:2509.01656 (2025).\
> [6] Zheng, Ziwei, et al. "DeepEyes: Incentivizing" Thinking with Images" via Reinforcement Learning." arXiv preprint arXiv:2505.14362 (2025).

---

> ### Author Response · Authors · 2025-11-21
>
> >`Weakness 2`: The method introduces two RL phases (GRPO then VPO) plus group rollouts and external LLM calls for rewards. While appendices mention batch sizes/steps, the wall-clock/cost, GPU hours, and reasoner-call counts per step are not surfaced prominently in the main text. This matters for practical adoption.
>
> **Response:** We thank the reviewer for the suggestion. We provide a summary regarding the training cost and resource utilization below.
>
> **1. Reasoner-Call Counts Per Step.**
>
> **The reasoner is invoked once per training step.** During training, the inference engine (vLLM) processes an entire batch of generated text at one time to obtain the solutions.
>
> **2. GPU Hours, Wall-Clock Time, and Cost.**
>
> Table X2 provides a breakdown of the resource utilization and associated costs for GRPO and VPO. As can be seen, **our method is remarkably cost-effective. This low financial barrier makes the approach highly accessible for small research teams, companies, and even individual researchers**. In addition, as discussed in Appendix E.1, these costs are less than 0.1% of the prohibitive expenses required for retraining an MLLM with a new LLM, making RAPID a highly practical and efficient alternative.
>
> We have included this cost analysis in **Section 4.6** of the revised paper.
>
>
> Table X2. Detailed cost breakdown for training.
>
> | Model Size | Training Phase | Wall-Clock Time (hours) | Training Steps |  GPUs | Total GPU Hours | Estimated Total Cost (USD) [1] |
> | :--- | :--- | :--- | :--- | :--- | :--- | :--- |
> | 3B | GRPO | 16.7 | 200 | 8 x H20 | 133.6 | $74.8 |
> | | VPO | 23.9 | 200 | 16 x H20 | 382.4 | $214.1 |
> | 7B | GRPO | 24.0 | 200 | 8 x H20 | 192.0 | $107.5 |
> | | VPO | 16.3 | 150 | 16 x H20 | 260.0 | $145.6 |
> | 32B | VPO | 13.6 | 100 | 32 x H20 | 435.2 | $243.7 |
>
> [1] The price per GPU (NVIDIA H20) per hour is $0.56 based on the cloud provider "LuchenTech".
>
> ---
>
>
> >`Weakness 3`: The hasCap(·) check is prompt-based. This heuristic may be gamed (e.g., hiding solutions within “caption-like” text) or fail across domains/styles. A small quantitative audit of false positives/negatives (and downstream impact) would be valuable.
>
> **Response:** We thank the reviewer for this insightful suggestion. We conduct the quantitative audit and perform analysis on the downstream impact below (also included in **Appendix E.3** of the revised paper).
>
>
>
>
> **1. Quantitative Audit of the `hasCap(·)` Heuristic**
> We manually audited a random sample of 400 generations (across various domains in ViRL-39K) from Qwen2.5-VL-7B (GRPO+VPO) and found that **across domains, nearly all valid captions (0.525% false negative rate) are accepted by the `hasCap(·)` check; the invalid ones (accounting for only 4.75% of all samples) are mostly rejected (26.32% false positive rate)**.
>
> Specifically, we classified each sample into two categories:
> *   **Positive Class**: The generation is a valid caption, which may or may not contain solutions. (381 samples)
> *   **Negative Class**: The generation could be simply a solution or a "gamed" output, such as a pure solution disguised with caption-style text (e.g., "Here is a description... [solution]"). (19 samples)
>
> The `hasCap(·)` prompt-based check produced the following results on this set:
>
> *   True Positives (TP): 379 (Correctly identified a valid caption)
> *   **False Negatives (FN)**: 2 (Incorrectly flagged a valid caption as invalid)
> *   True Negatives (TN): 14 (Correctly identified a gamed/invalid caption or a solution)
> *   **False Positives (FP)**: 5 (Incorrectly identified a gamed/invalid caption or a solution as valid)
>
> From these numbers, we derive the following metrics for the `hasCap(·)` detector as requested by the reviewer:
> *   **False Negative Rate**: 2 / (379 + 2) = **0.525%**
> *   **False Positive Rate**: 5 / (5 + 14) = **26.32%**
>
>
>
> **2. Analysis of Failure Modes and Downstream Impact**
> The reviewer's concern is mainly about the false positives (or the gamed output ignored by the `hasCap(·)` check). We found they exhibit a similar pattern (hiding **pure** solution in a caption-like text) as shown below:
>
> ```
> # Example of a game output (incorrectly identified as positive by our check):
>
> Ok, here is a description of the image regarding the query. To find the circumference... [detailed mathematical derivation] ... Therefore, the circumference of the circle is 25.12 cm. ... This description aligns with the mathematical calculation...
>
> ```
>
>
> Though such cases occur, **they are too infrequent (5 out of 400 total samples) to impact the downstream performance.**

---

> ### Author Response · Authors · 2025-11-21
>
> >`Question 1`: Does RAPID extend to multi-image and video inputs without architectural changes?
>
> **Response:** Yes, RAPID natively supports multi-image and video inputs without any changes to the model architecture. When generating captions and tentative solutions, the MLLM processes multiple images in a single query.
>
> In addition, this ability has already been validated on our training and evaluation datasets. Specifically, our ViRL-39K training set averages 1.11 images per query, while the MMMU evaluation benchmark averages 1.09 images per query.
>
> ----
>
>
> >`Question 2`: You find R1-7B best for training rewards and Qwen3-8B strong for inference. Could you elaborate on the caption-length & difficulty balance hypothesis (Fig. 9) with controlled experiments (e.g., length-controlled rewards)?
>
> **Response:** Thank you for this insightful question.
>
> To investigate whether caption length (Figure 9) is a confounding variable for performance, we conducted a controlled experiment to decouple it from the reward LLM's identity. The results suggest that **the performance gap is attributable to the reasoning ability of the LLM rather than the caption length.** We include this experiment in **Appendix E.5** of the revised paper.
>
> Specifically, we start with the stronger Qwen3-8B for reward computation, which normally encourages shorter captions (150 tokens) in the perception MLLM (Qwen2.5-VL-7B) and leads to lower decoupling performance. **We then apply a length-controlled reward on the MLLM to force it to generate longer captions (650 tokens), matching the length when using the R1-7B.**
>
> To achieve this, we added a length penalty term to the reward function, as in [1]: $r_{len}(y, n_\text{target}) = - \alpha |n_\text{target} - n_y|$. Here, $n_\text{target}$ is set to 650, $n_y$ is the token count of the generated caption $y$, and the weight $α$ is set to $0.0003$ as [1] did.
>
> Table X4. Correlation between the choices of LLM, the length of the captions, and the final performance.
>
> | LLM for reward computation | Avg. Caption Length | Math Vista | Math Vision | Math Verse | MMMU | We Math | Dyna Math | Logic Vista | AVG |
> |:----------------|-----------:|-----------:|------------:|-----------:|-----:|--------:|----------:|------------:|----:|
> | R1-7B       | ~653.7 | **76.1**       | **43.7**        | **52.2**       | **64.7** | **45.4**    | **32.7**      | **57.7**        | **53.2**|
> |Qwen3-8B     | ~153.1 | 75.3       | 43.4        | 47.6       | 62.4 | 42.8    | 30.1      | 57.5        | 51.3|
> |Qwen3-8B  (length-controlled)  | ~627.1 | 75.7       | 43.2        | 48.6       | 62.6 | 43.4    | 31.1      | 57.3        | 51.7|
>
> The results in Table X4 show that despite the length-control reward could increase the caption length, **it only marginally improves the overall performance (from 51.3 to 51.7)**, which still lags behind the MLLM trained with R1-7B rewards (53.2).
>
> **This allows us to eliminate the caption length as a confounding variable, confirming that the performance gap is attributable to the reasoning ability of the LLM.**
>
> ---
>
> ### **Reference**
>
> [1] Aggarwal, Pranjal, and Sean Welleck. "L1: Controlling how long a reasoning model thinks with reinforcement learning." _COLM_, 2025

---

> > ### Comment · Area_Chair_izn1 · 2025-11-26
> >
> > Dear Reviewer,
> >
> > Thanks for your time and effort in reviewing ICLR2026 submissions. The authors have provided their responses to your reviews. Please read and raise your further comments, and discuss with the authors.
> >
> > Best regards,
> >
> > Your AC

---

> > ### Comment · Reviewer_wDxG · 2025-11-26
> >
> > Thank you for the author’s response, which addressed most of the concerns.

---

> > > ### Author Response · Authors · 2025-11-26
> > >
> > > We are glad that most of your concerns have been addressed, and we appreciate your revised scores. If you need any further information or clarification, please let us know.

---

### Official Review · Reviewer_DHha · 2025-10-29

**Soundness:** 3
**Presentation:** 3
**Contribution:** 3
**Rating:** 8
**Confidence:** 4

**Summary:**

This paper introduces VPO, which rewards the MLLM based on the correctness of answers generated by the external reasoner. They use a caption penalty to make sure the MLLM focus on the visual understanding, while the LLM mainly serve as a reasoner. Overall, this framework seems to be simple yet effective.

**Strengths:**

1. The paper proposes training a visual captioner using the accuracy of LLM responses as a reward signal. Although combining VLM-based captioning with LLM reasoning has become common, to the best of my knowledge, introducing RL for training a visual captioner is novel. Therefore, the work demonstrates strong originality.

2. The experiments are very solid, particularly as the conclusions are validated across models of multiple scales, with comprehensive ablation studies provided.

3. The paper is easy to understand, and the formulations related to RL training are presented with rigor.

4. The literature review is complete, and the code has been open-sourced.

**Weaknesses:**

The main limitation of the paper is that it does not provide a sufficiently detailed justification for the necessity of adopting a VLM captioner + LLM reasoner framework. In fact, many existing open-source and closed-source MLLMs still rely on unified reasoning, and the unified model appears to be an important trend in the development of large models.

The authors could elaborate on their perspective regarding the future direction of large models: whether they believe the field will move toward unified architectures or expert-agent-based systems. If the explanation is convincing, I would be willing to raise my score to 10, as this is already a very solid piece of work.

**Questions:**

See weakness.

---

> ### Author Response · Authors · 2025-11-21
>
> >`Weakness 1`: Insufficient justification for adopting a modular VLM captioner + LLM reasoner framework over a unified MLLM architecture. The authors could provide perspectives on future direction of large models (unified models or expert-agent-based systems).
>
> **Response:** We thank the reviewer for allowing us to elaborate on our justifications for modular architecture and perspectives on future direction, which we present below.
>
> We claim that expert-agent-based systems are not contradictory to unified systems, but rather serve a complementary role. **First, they could serve as a data engine to build future powerful unified models. Second, under a limited budget, this could be a more pragmatic and economic solution to address the gap in model capacity and training budget of unified models.**
>
> We have included the following discussion in **Appendix G.1** of the revised paper.
>
> **1. Expert-agent-based System as a Data Engine for Future Unified Models.**
>
> We agree that building a powerful, unified MLLM is the ultimate goal. However, training such models is hampered by the scarcity of high-quality, multi-modal reasoning data. Our modular framework directly addresses this bottleneck. The RAPID pipeline can generate vast amounts of complex reasoning trajectories. This high-quality, model-generated data can then be used to train and significantly improve a future unified MLLM, a technique proven to be effective in prior work [1,2].
>
>
>
>
> **2. Expert-agent-based System Bridges Current Ability Gap.**
>
> At present, general-purpose MLLMs still lag behind specialized models in critical perception tasks like object counting [3], fine-grained OCR [4], depth estimation [5], and semantic segmentation [6]. However, expert-agent-based systems could pragmatically bridge this gap by integrating these "expert" models [7,8,9]. This allows the system to leverage state-of-the-art performance on these sub-tasks immediately, achieving higher overall accuracy.
>
> **3. The Expert-agent System as an Economic Solution with Limited Resources.**
>
> While a unified architecture is a compelling goal, an expert-agent system is a more pragmatic solution under restricted training budgets. It circumvents the prohibitive cost of training a unified model on massive data. For example, RAPID could enjoy the advanced reasoning ability of new LLMs without training a new MLLM from scratch.
>
>
>
>
> ---
> ### **Reference**
> [1] Yang, Yi, et al. "R1-Onevision: Advancing generalized multimodal reasoning through cross-modal formalization." _ICCV_, 2025. \
> [2] Huang, Wenxuan, et al. "Vision-R1: Incentivizing reasoning capability in multimodal large language models." arXiv preprint arXiv:2503.06749 (2025).\
> [3] Tamarapalli, Jayant Sravan, et al. "CountQA: How Well Do MLLMs Count in the Wild?." arXiv preprint arXiv:2508.06585 (2025).\
> [4] Chen, Song, et al. "Ocean-OCR: Towards general ocr application via a vision-language model." arXiv preprint arXiv:2501.15558 (2025).\
> [5] Fu, Xingyu, et al. "BLINK: Multimodal large language models can see but not perceive." _ECCV_, 2024.\
> [6] Anonymous, et al. "SAM 3: Segment Anything with Concepts."" _ICLR_ 2026 Submission.\
> [7] Zhou, Zetong, et al. "Reinforced visual perception with tools." arXiv preprint arXiv:2509.01656 (2025).\
> [8] Su, Zhaochen, et al. "OpenThinkIMG: Learning to think with images via visual tool reinforcement learning." arXiv preprint arXiv:2505.08617 (2025).\
> [9] Liu, Yuqi, et al. "VisionReasoner: Unified Visual Perception and Reasoning via Reinforcement Learning." arXiv preprint arXiv:2505.12081 (2025).

---

> > ### Comment · Reviewer_DHha · 2025-11-24
> >
> > Thanks to the authors for the timely response! The authors’ reply is reasonable. After reading the other reviewers’ comments and the authors’ responses, I still believe that this is a high-quality paper worthy of publication. With the addition of more baselines and experiments on InternVL, the results have become more convincing. I will keep my score at 8 and am willing to raise the Soundness rating to 4.

---

> ### Author Response · Authors · 2025-11-24
>
> Thanks for your prompt reply! We are glad that you find our response reasonable. Please feel free to reach out with any further questions!

---

### Official Review · Reviewer_cXzG · 2025-10-31

**Soundness:** 2
**Presentation:** 2
**Contribution:** 2
**Rating:** 2
**Confidence:** 3

**Summary:**

This paper introduces RAPID, a method to enhance the reasoning of Multi-modal Large Language Models (MLLMs) whose internal language models are often outdated and expensive to upgrade. The approach decouples the system, using the MLLM for perception to generate detailed textual descriptions from images, which are then fed to a separate, powerful text-only LLM for reasoning. A novel reinforcement learning algorithm, Visual Perception Optimization (VPO), is used to train the MLLM to produce captions that are optimized for the final reasoning task. This "plug-and-play" design allows for easy upgrading of the reasoning LLM, enabling performance improvements without costly retraining of the vision components.

**Strengths:**

1. The paper is very well-written and logically structured, making the core ideas easy to understand. The figures are clear and effectively illustrate the main concepts.

2. The motivation is straightforward.

**Weaknesses:**

1. This paper lacks novelty. The paper presents a relatively straightforward idea. It uses a multimodal large language model (MLLM) to describe an input image and then relies on a separate language model for textual reasoning. This setup mainly combines existing components rather than introducing a new methodological contribution. The multimodal stage performs description rather than genuine reasoning, which limits the conceptual depth of the approach.

2. The proposed pipeline does not offer clear benefits in performance or efficiency. For example, when using Qwen2.5-VL-7B as the captioner and Qwen3-8B as the reasoner, the results reported in Table 1 remain below those of the single multimodal model Qwen3-8B-Instruct without reasoning. The method also requires roughly twice as many model parameters, which increases computational cost without providing noticeable performance gains.

3. The decoupled architecture is vulnerable to error accumulation. If the perception stage produces an inaccurate or incomplete description, the reasoning model has no way to recover from that mistake. This makes the overall system unstable and limits its reliability compared with end-to-end multimodal reasoning approaches.

**Questions:**

N/A

---

> ### Author Response · Authors · 2025-11-21
>
> > `Weakness 1:` This paper lacks novelty. The paper presents a relatively straightforward idea. It uses a multimodal large language model (MLLM) to describe an input image and then relies on a separate language model for textual reasoning. This setup mainly combines existing components rather than introducing a new methodological contribution. The multimodal stage performs description rather than genuine reasoning, which limits the conceptual depth of the approach.
>
> **Response**: Thank you for your feedback. We address the concerns regarding novelty and conceptual depth below.
>
> **1. Regarding novelty:**
>
> Instead of a simple combination of MLLM and LLM, we respectfully clarify that the core novelty is about synergy between the decouple pipeline and VPO algorithm that allows the LLM in an MLLM to be flexibly replaced by any advanced ones (Figure 1) without burdensome retraining. As shown in Table 8, compared to MLLMs that are retrained, we obtain **~95%** of its performance while requiring less than **0.1%** of its training compute (FLOPs). **Notably, this cannot be achieved with solely the decouple pipeline as mentioned by the reviewer, but requires the proposed VPO algorithm** which creates a task-aware synergy between the MLLM and LLM, with its effect validated in Table 2.
>
>
>
> **2. Regarding conceptual depth:**
>
> We respectfully argue that the reviewer's statement that **"the multimodal stage performs description rather than genuine reasoning" is factually incorrect.** In the decouple pipeline of RAPID ("qcap+sol" in Sec. 3.1), the MLLM explicitly performs both** tasks in the following:
> *   Query-relevant captioning: Generating a detailed, query-relevant caption of the image.
> *   Tentative solutions: Proposing an initial solution of the problem that includes **genuine reasoning in the multi-modal stage**. This stage is shown to improve "qcap" by 5.6% in Figure 3.
>
>
>
> The conceptual strength of this approach is validated by our empirical results: Table 2 shows that the decoupled pipeline improves the baseline MLLM by +5.5%, and the proposed VPO adds another 2.7% on top of the decoupled pipeline.
>
> ---
>
>
>
> >`Weakness 2:` The proposed pipeline does not offer clear benefits in performance or efficiency. For example, when using Qwen2.5-VL-7B as the captioner and Qwen3-8B as the reasoner, the results reported in Table 1 remain below those of the single multimodal model Qwen3-8B-Instruct without reasoning. The method also requires roughly twice as many model parameters, which increases computational cost without providing noticeable performance gains.
>
> **Response:** Thank you for your feedback.
>
>
> First of all, ALL multi-modal models based on Qwen3-8B compared in our paper have reasoning enabled. **There is no counterpart throughout the whole paper which is "single multimodal model Qwen3-8B-Instruct without reasoning", as claimed by the Reviewer.**
>
> Secondly, we respectfully argue **the reviewer's claim that RAPID does not offer benefits in performance or efficiency is factually not true.** In the following, we show that RAPID offers a superior performance-efficiency trade-off (~95% of the top performance with <0.01% training compute) compared to the latest end-to-end MLLMs and brings direct superiority in performance over contemporary MLLMs.
>
>
>
> **1. Exceptional Efficiency vs. Latest MLLMs.**
> As shown in Table 8, compared to the latest MLLM retrained from scratch (e.g., Keye-VL-8B or InternVL3.5-8B), RAPID achieves highly competitive performance (**~95%** of them) while requiring less than **0.1%** of the training compute (FLOPs).
>
>
> Moreover, we re-emphasize that the main advantage of the proposed RAPID is that it allows the LLM in an MLLM to be flexibly replaced by any advanced ones without burdensome retraining. **Hence, it is not fair to compare RAPID with an MLLM that has been retrained from the latest reasoning LLM**, as this retraining effort is very computationally expensive while the training of RAPID is very inexpensive.
>
> **2. Performance superiority vs. contemporary MLLMs.**
> As illustrated in Figure 1 and Table 1, compared to contemporary LLMs (e.g., built on Qwen2.5 series), augmenting the Qwen2.5-VL series with RAPID allows it to **outperform other contemporary MLLMs**.

---

> ### Author Response · Authors · 2025-11-21
>
> >`Weakness 3:`The decoupled architecture is vulnerable to error accumulation. If the perception stage produces an inaccurate or incomplete description, the reasoning model has no way to recover from that mistake. This makes the overall system unstable and limits its reliability compared with end-to-end multimodal reasoning approaches.
>
>
> **Response:** Thank you for your feedback. We address this concern from two perspectives.
>
> **1. Mitigation: Cross-Checking With a Tentative Solution.**
> We have anticipated this potential issue and have mitigated it by giving the reasoner both a query-aware caption and a tentative solution from the MLLM, with explicit prompts (Fig. 12) instructing cross-checking: **if caption-based reasoning becomes inconsistent, the reasoner could rely on the tentative solution.** While it cannot re-examine the image, this fallback prevents catastrophic failures and yields consistent accuracy gains (**+5.6%** over "qcap" in Fig. 3).
>
> **2. Error vs. Compute: A Favourable Trade-off.**
> RAPID delivers ~95% of top performance using <0.1% of the training compute. In addition, its modular design enables quick, efficient updates to newer LLMs. In this context, minor error propagation is accepted and represents a negligible trade-off for an over 1,000× reduction in training cost.

---

> > ### Comment · Area_Chair_izn1 · 2025-11-26
> >
> > Dear Reviewer,
> >
> > Thanks for your time and effort in reviewing ICLR2026 submissions. The authors have provided their responses to your reviews. Please read and raise your further comments, and discuss with the authors.
> >
> > Best regards,
> >
> > Your AC

---

> ### Author Response · Authors · 2025-11-28
>
> Dear Reviewer cXzG
>
> We thank you for your detailed comments. We hope the above discussion have addressed your concerns. If the revisions meet your expectations, we would appreciate your consideration of updating the score to reflect your latest evaluation. Please feel free to let us know if any points remain unclear, and we are willing to provide further clarification.
>
> Best regards,
>
> Authors

---

> > ### Comment · Reviewer_cXzG · 2025-11-28
> >
> > Thank you for your detailed response and for addressing several of my points. Based on the clarifications provided, I will increase my score. However, I still have two remaining issues:
> >
> > 1. I acknowledge the contribution of the proposed VPO. My primary concern regarding the novelty of the framework remains. As shown in Figure 3, simply combining the LLM reasoner and a more powerful, external MLLM boosts performance significantly from 22 to 45. This suggests that the decoupled pipeline is the dominant factor in the performance gain, rather than the VPO method itself, which yields a comparatively smaller marginal improvement.
> >
> > 2. I noticed a potential inconsistency between the reported results. The plain baseline performance in Table 2 (Row A) appears to differ from the "none" performance shown in Figure 3. Could you please clarify the reason for this discrepancy?

---

> ### Author Response · Authors · 2025-11-28
>
> Thank you for your feedback. We appreciate the opportunity to provide further clarification, which we hope will resolve your remaining concerns.
>
> ---
>
> >`Issue 1`: I acknowledge the contribution of the proposed VPO. My primary concern regarding the novelty of the framework remains. As shown in Figure 3, simply combining the LLM reasoner and a more powerful, external MLLM boosts performance significantly from 22 to 45. This suggests that the decoupled pipeline is the dominant factor in the performance gain, rather than the VPO method itself, which yields a comparatively smaller marginal improvement.
>
> **The score of 22 (Figure 3) should not be treated as a baseline to assess the effect of the decouple pipeline, and the correct baseline result is 42 (Table 2). Therefore, both VPO and the decouple pipeline are important for performance gains.** Specifically:
>
> - **The "22" result ("none") in Figure 3 is not a baseline**, but a specific ablation study. As described in lines 165-166, this experiment intentionally provides an empty perceptual output ($O_p$) to the LLM reasoner (i.e., no image inputs) to establish a lower bound for different perceptual outputs.
>
> - **The correct baseline ("MLLM only") is 42.0**, which is the performance of the original, end-to-end MLLM (we report in row A of Table 2). We have also explicitly marked this with a dashed line in Figure 3.
>
>
> With the correct baseline established, we can accurately assess the distinct contributions of our RAPID's components:
>
> 1. **The baseline "MLLM only"** performance is **42.0** (Row A of Table 2 or the dashed line in Figure 3).
> 2. **The decoupled pipeline** lifts performance by ~**3** points (approx. 45 with R1-7B) and **5.5** points (47.5 with Qwen3-8B), as shown in Figure 3.
> 3. **Our VPO algorithm** provides a further improvement of **2.7** points over the non-optimized pipeline (Table 2, Row E), which is **not marginal**.
>
> **This shows that the performance gain is not dominated by the decoupled pipeline alone. Instead, both the decoupling pipeline and our VPO method are crucial to the final result.**
>
> ----
>
> >`Issue 2:` I noticed a potential inconsistency between the reported results. The plain baseline performance in Table 2 (Row A) appears to differ from the "none" performance shown in Figure 3. Could you please clarify the reason for this discrepancy?
>
>
> **The plain baseline performance in Table 2 (Row A) and the "none" performance shown in Figure 3 are two different results and do not need to be consistent**.  As explained in our response to `Issue 1`, the plain baseline in Table 2 (row A) is the "true" baseline, while the "none" in Figure 3 (introduced in lines 165-166) is just a specific ablation of the perceptual outputs.

---

### Author Response · Authors · 2025-12-03

Dear ACs, SACs, and PCs:

Thank you for your hard work in overseeing the review process. We are writing to summarize the outcomes of our author-reviewer discussions, during which **all reviewers either raised their scores or indicated their intent to do so.**

We understand these updates were reverted to their pre-discussion state due to the system-wide issue on Nov 27. **Crucially, we wish to emphasize that most positive re-evaluations (from reviewer DHha, wDxG, and izU6) were submitted before this incident, when reviewer anonymity was fully preserved.** The score increases from Reviewers DHha, wDxG, and izU6 occurred between Nov 24 and Nov 26. This timing confirms their updated assessments were based solely on the merits of our rebuttal and revisions.

----

## Reviewer cXzG (overall 2 → all concerns resolved & planning to increase the score)

In response to the reviewer's concerns, we have made clarifications regarding the novelty, conceptual depth, benefits in performance and efficiency, and error accumulation.



The reviewer commented that we **"addressed several concerns" and is willing to increase the scores.** However, review editing was disabled at that moment. The reviewer's remaining two concerns (i.e., component contributions in RAPID and potential performance inconsistency) **stemmed from a misunderstanding of the results in Figure 3 and Table 2**, which have been clearly explained by our follow-up responses.

----

## Reviewer DHha (overall 8; soundness 3 → overall 8; soundness score 4 on 24th Nov)

As requested by the reviewer, we elaborated on our justification for modular architecture and perspectives for future directions. The reviewer found our reply **"reasonable"** and believed our submission **" is a high-quality paper worthy of publication"**. As a result, the reviewer had **raised the Soundness score from 3 to 4**.

----

## Reviewer wDxG (overall 4 → 6 on 26th Nov)

As suggested by the reviewer, to address the weaknesses, we have
- compared the proposed RAPID with verification-augmented and tool-enabled LMMs and found RAPID shows clear advantages (Table 1).
- presented the additional training cost in Table 5, demonstrating it is practical (Sec. 4.6).
- audited the use of hasCap(·) check and found it is robust and accurate (Appendix E.3).

The reviewer commented that these **addressed the concerns and raised the score from 4 to 6**.

----

## Reviewer izU6 (overall 6 → 8 on 24th Nov)

To address the reviewer's concerns, we
- Extended RAPID to InternVL3-8B and found it is a generalizable method not limited Qwen2.5-VL series experimented in the paper (Sec. 4.5).
- Applied the proposed decouple pipeline to other MLLMs and found it widely effective (Sec. 4.5).
- Compared the captions generated by MLLM optimized with VPO and other baselines and found VPO produced better captions (Appendix F.3).
- Leveraged a simple GRPO fine-tuning to recover the reasoning ability of the MLLM (previously decreased after VPO) (Appendix E.4).

The reviewer commented that our experiments and evaluations are **"comprehensive" and updated the score from 6 to 8.**

---

### Meta-Review · Area_Chair_rzVR · 2025-12-24

**Summary:**

The paper proposes to upgrade the LLM part of an MLLM with Perception-Reasoning Decoupling and visual perception optimization (VPO), named as RAPID, so that the reasoning ability of an MLLM is improved. The main strength of the submission lies in the comprehensive experimental results but the concerns on the lack of novelty also hold from my perspective. It seems that the RAPID framework just applies reinforcement learning to an MLLM so that it becomes a stronger tool for a powerful LLM. However, the authors did a great job in addressing concerns raised by the reviewers and defended the submission quite well. In summary, the strengths in the empirical results outweigh the weakness in novelty so I recommend acceptance.

**Reviewer Concerns:**

I think all of the following concerns are addressd exept for the novelty one.

1.	The proposed method is not novel (cXzG)

2.	Performance/efficiency gain is marginal (cXzG)

3.	Error accumulation is not studied (cXzG)

4.	Justification for the MLLM+LLM framework is not sufficient (DHha)

5.	Lack of comparison with verification-augmented or tool-enabled LMMs (wDxG)

6.	Experimental details are not sufficient (wDxG)

7.	Failure cases analysis is missing (wDxG)

8.	Lacks comparison of using existing baselines (wDxG)

9.	The reasoning capability of the underlying MLLM is compromised (wDxG)

**Reviewer Scores:**

Reviewer cXzG (overall 2 → all concerns resolved & planning to increase the score) I think the review might increase to 4 or 6.

Reviewer DHha (overall 8; soundness 3 → overall 8; soundness score 4 on 24th Nov)

Reviewer wDxG (overall 4 → 6 on 26th Nov)

Reviewer izU6 (overall 6 → 8 on 24th Nov)

This gives an overall score of 4668 at least.

---

### Decision · Program_Chairs · 2026-01-26

Accept (Poster)